# Vulnerability of power distribution networks to local temperature changes induced by global climate change

Kishan Prudhvi Guddanti[1,4], Lin Chen ®[2,4], Yang Weng ®[1] ✉ & Yang Yu ®[3] ✉

Global climate change (GCC) triggers a chain effect, converting temperature pattern changes into variations in blackout risk for power distribution grids (DGs). This occurs through GCC's impacts on electricity supply, demand, and infrastructure, which shift the DG's safe-operation boundary and power flow. This study presents a model integration framework to assess the associated blackout risk, showing that GCC raises blackout risks during peak hours by 4–6%, depending on Gross Domestic Product growth. Kirchhoff's laws amplify these effects, creating nonlinear risk trajectories. Analysis of the chain effect suggests adaptation strategies, including reshaping grid topology and pairing temperature-sensitive users with robust buses. Index-based analysis reveals that over 20% of the U.S. requires at least a 10% DG capacity increase before 2050, with six states exceeding 20%. Europe faces a more moderate impact. These findings highlight the need for policymakers to prioritize peak-load management and address nonlinear risks across regions.

Global climate change (GCC) has been deeply impacting every component of the electricity sector, such as resizing the service capacity of the grid facilities and reshaping the profile of electricity demand[1,2]. For instance, GCC is projected to increase the number of cooling-degree days[3] and exacerbate the peak electricity demand for adapting to the high temperature in hot hours[3–5]. Meanwhile, the anticipated rise in local temperatures in cooling-degree days due to GCC will reduce the capacity of distribution grid (DG) lines and increase the electricity demand.

GCC impacts various power system components, affecting DG reliability. DG, which delivers electricity from high-voltage transmission to users, has less redundancy and protection[6,7], making it more vulnerable. Studies show that 90% of U.S. blackouts stem from DG failures[8]. As a result, DG blackout risk is highly sensitive to changes in infrastructure and electricity demand driven by GCC. Specifically, GCC affects DG reliability through cascading effects, influencing demand, grid topology, operational capacity, and overall reliability. These changes reshape DG power flow and its capacity, changing blackout probabilities. Kirchhoff's laws govern how GCC-driven changes

interact, resulting in a nonlinear blackout risk. To assess these risks, modeling must capture both chain effects and the intermediate process.

It is crucial to understand how the combined impacts of GCC influence the DG reliability through a chain effect. On one hand, these interconnected impacts can make DG more vulnerable. Even if each individual GCC's impact is insufficient to cause a blackout event, their interfered effects can push DG to the brink of failure. Furthermore, the nonlinear response of the power flow can amplify these cumulative effects throughout the cascading process, exacerbating the DG vulnerability and increasing the risk of system failure. Additionally, the reliability of the electricity distribution service plays a crucial role in shaping regional economic advantages and quality of life. For instance, the reliability of electricity during hot and cold hours influences regional public health. Every DG blackout causes a severe economic loss[9]. Finally, the progress of the energy transition drives a growing penetration of electric vehicles, distributed energy generation technologies, and energy storage facilities, all connected to DG[10]. Therefore, a reliable electricity

[1]School of Electrical, Computer and Energy Engineering, Arizona State University, Tempe, USA. [2]Institute for Interdisciplinary Information Sciences, Tsinghua University, Beijing, China. [3]School of Economics and Management, China University of Petroleum, Beijing, China. [4]These authors contributed equally: Kishan Prudhvi Guddanti, Lin Chen. ✉e-mail: yang.weng@asu.edu; yangyu@cup.edu.cn

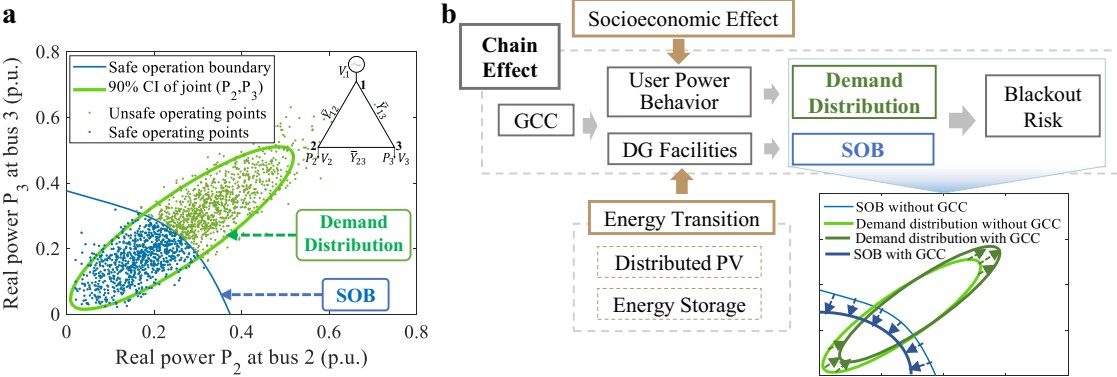

**Fig. 1 | Framework for assessing global climate change impact on distribution grid blackout risk. a** The left part of the figure shows a sample distribution grid (DG) and the demand-temperature sensitivity of two groups of real-world consumers at 30 °C. The green ellipse represents the 90% quantile of the joint distribution of real power consumption at nodes $P_1$ and $P_2$. If the two groups of consumers are located at Nodes 2 and 3 of the triangular DG, the blue points depict a safe operating boundary (SOB), while the green points represent demand profiles that could trigger a blackout. This allows for calculating the blackout probability based on the joint distribution of nodal demands. **b** This diagram illustrates the chain effect of Global climate change (GCC)'s combined impacts and associated processes. It depicts how GCC-induced temperature changes influence DG blackout risk, with intermediate processes shaped by energy transitions, as well as demographic, socio-cultural, and economic factors. A small picture in the lower right corner shows GCC's impact on the SOB of the DG and demand distribution. Due to GCC, the SOB shrinks, and the joint distribution of power demand shifts upwards, indicating increased demand. The movement of the SOB and demand ellipse away from each other results in an increased risk of DG collapse, which is the nonlinear effect.

distribution service is essential for the stability and advancement of human society[11].

To understand the combined impacts of GCC through the chain effect, it is essential to integrate models that capture both the GCC's influence on various components of the electricity sector and the way these combined impacts translate into the DG's blackout risk through the complex DG power flow dynamics. Existing studies have primarily examined the impacts of GCC on various components of DG, including shifts in demand profiles and changes to the grid infrastructure. Data-driven methods have been developed to estimate how electricity demand responds to local temperature fluctuations in the transmission grid, providing insights into load variations[12–14]. Discussions on the GCC's impact on the DG's infrastructure can be categorized into two main approaches. The first focuses on physical damage to infrastructure caused by extreme weather events. For instance, studies have shown that the DG's components, such as the grid lines and transformers, are highly susceptible to damage from extreme weather conditions[15,16]. Other forms of structural damage have also been explored in this context[17,18]. The second approach examines capacity constraints, which arise when electricity demand surpasses the substation capacity, leading to operational inefficiencies and reliability concerns[5,11,12].

However, a gap remains in frameworks that integrate GCC's impact on nodal load and infrastructure while modeling power flow dynamics to assess DG blackout risk. The challenge lies in converting load fluctuations and infrastructure changes into DG power flow dynamics. Consequently, how DG reliability is transformed by GCC remains unclear. The relationship between GCC and blackout risk follows a complex climate-economic-engineering process[19], making direct temperature-based vulnerability estimates unreliable.

This work develops a model integration framework to capture the climate-economic-engineering chain effect through which GCC influences DG's blackout risk. Our framework integrates GCC, temperature-demand, and temperature-facility models, with an intermediate-process model linking them. The model of intermediate process captures the nonlinear process that the DG power-flow dynamic converts changes in nodal loads and grid infrastructure into the probability of the DG's blackout risk. This enables the analysis of GCC's impact on DG reliability to account for micro-topology and technical factors, which reveals micro-dynamics of the chain effect and informs the non-conventional adaptation strategies, such as topological redesign and matching temperature-sensitive users with non-vulnerable nodes. Applied to real-world data, we quantify climate change impacts at both micro and macro scales. We demonstrate that GCC's impact, amplified through power-flow dynamics, has been influencing DG reliability in reality. This effect was previously offset by grid investments for economic growth before 2020. However, it is projected to become too substantial to be counterbalanced by economically driven DG expansion after 2030.

## Results
### Model integration framework for analyzing the combined impacts of GCC through chain effect
A DG blackout occurs when the profile of nodal electricity load leads the power flow through any line to exceed its capacity. The safe-operation region is the set of all nodal load profiles that prevent such overflows, whose boundary is known as the safe-operation boundary (SOB). The risk of a DG blackout in any given hour $h$ is determined by the probability that the demand profile exceeds the SOB, which depends on two factors: (1) the probability of temperature $t$ occurring $p(t)$, and (2) the probability that the demand profile **P** exceeds the SOB at temperature $t$.

The blackout risk can be quantified in equation (1):

$$Risk_h = p(t) \cdot p(\Phi[\mathbf{P}|t, h] \le 0), \qquad (1)$$

where $\Phi[\mathbf{P}|t, h]$ measures the margin between the demand profile **P** and the SOB under temperature $t$ at hour $h$. If $\Phi \le 0$, the demand profile exceeds the SOB, leading to a blackout. Figure 1a presents the SOB of a three-node DG and conceptually explains the blackout process. Note that the SOB is a physical characteristic of the grid and independent of specific load profiles. The SOB is determined by the physical characteristics of the DG facilities.

Figure 1b illustrates the climate-economic-engineering chain effect driving DG blackout risk changes. GCC-induced temperature shifts change DG facility capacity and electricity demand. The intermediate process links these effects, reshaping DG blackout risk via two pathways: facility capacity changes resize DG's maximum load-handling capacity, while facility and demand changes modify DG

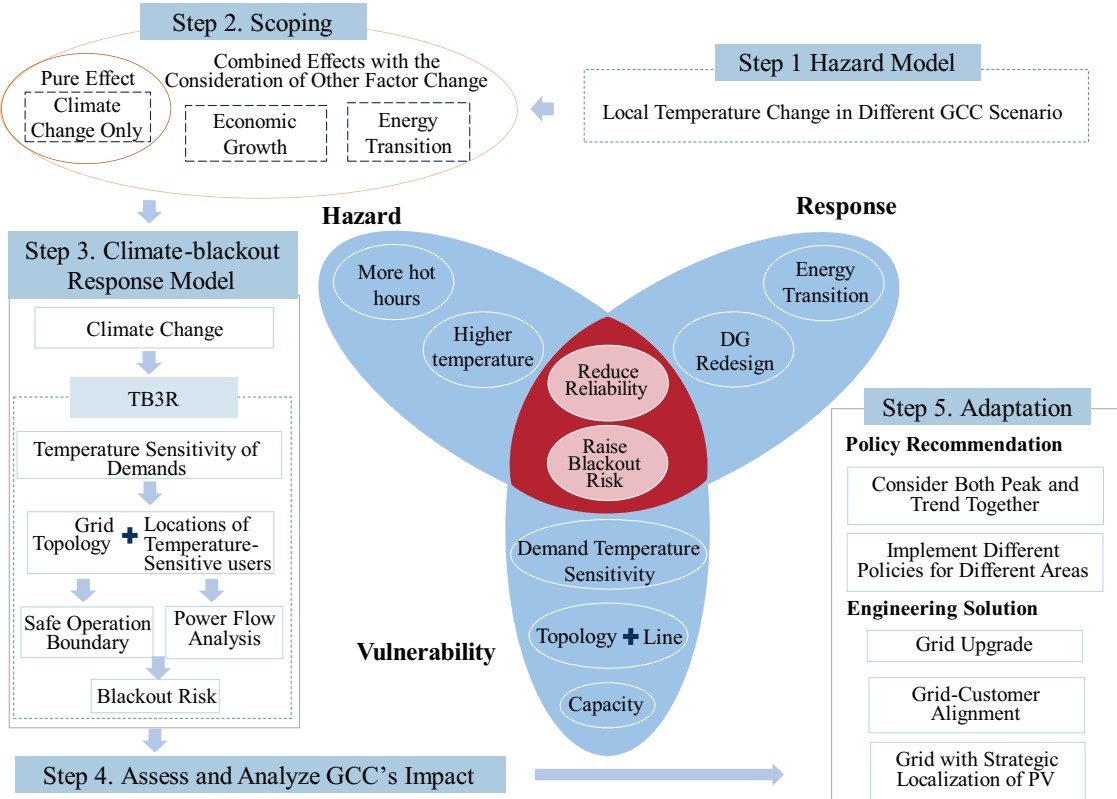

**Fig. 2 | The analysis diagram based on model integration for assessing the impact of climate change on electricity distribution services.** This analysis diagram is visualized using a centered flower-style plot that summarizes the hazard-vulnerability-response model[21]. Hazard refers to the potential physical events or phenomena that could cause harm or disrupt the power distribution system. Vulnerability is the degree to which a power distribution system and its consumers are susceptible to the adverse impacts of climate change. Response refers to how the power distribution grid reacts to climate-induced hazards. Step 1: Modeling the hazard of the global climate change (GCC)-driven local temperature changes. Step 2: Scope the Environment for Assessment. The assessment begins by identifying the pure effect of climate change, isolated from other factors such as economic growth

and energy transition. Step 3: Climate-Blackout Response Model. This step involves modeling the response of the grid to the interfered effects of temperature sensitivity of demand and climate change under various GCC scenarios, which can be analyzed by Temperature-Blackout Risk Response (TB3R). Step 4: Assessing and comparing the GCC's impact. Step 5: Analyze how the GCC-driven temperature change raises the blackout risk and design adaptation strategies. The analysis diagram captures the complex interactions between demand temperature sensitivity, grid topology, capacity, and climate change, providing some adaptations to address the vulnerabilities of electricity distribution grids to climate-related stresses.

power flow together. These combined variations jointly decide blackout probability.

The intermediate process shapes the GCC's combined impacts on DG's blackout risk. During the intermediate process, rising temperatures shrink grid capacity via the Temperature Coefficient Formula, while the line-capacity change and demand shifts alter power flow per Kirchhoff's laws, both leading to nonlinear blackout risk variations.

Engineering advancements and socioeconomic developments influence the intermediate process, allowing energy transitions and Gross Domestic Product (GDP) growth to interfere with GCC's impact. The deployment of distributed Photovoltaics (PV) and storage modifies grid capacity, leading to GCC's influence being altered by energy transition progress. The temperature-demand relationship is characterized by demographic, socio-cultural, and economic factors. The interference of engineering advancement and socioeconomic developments also makes DG adaptation region-specific. For instance, although Colorado and Kansas share similar climates, their demand-temperature relations differ due to socioeconomic factors, necessitating distinct adaptation strategies.

Due to the chain effect, assessing the GCC's impact on DG reliability requires an integrated modeling that encompasses the GCC-driven local meteorological changes, electricity demand response, grid infrastructure adaptation, variations in the SOB, and DG power flow dynamics. We combined our DG reliability assessment method[20]

and the GCC analysis framework[21] to develop the framework of model integration, referred to as the Temperature-Blackout Risk Response (TB3R) framework. The TB3R framework combines the GCC model, temperature-demand sensitivity model, temperature-grid SOB model, and power flow model. The TB3R framework is illustrated in Fig. 2, which involves five key steps. The first step models GCC-driven local temperature changes. The second step determines the scope of assessment, which identifies whether and what other processes are included when assessing the GCC's impact. When the intervention of all other processes, such as GDP growth, is excluded, the assessment calibrates the size of the GCC's pure effect. Otherwise, the assessment verifies the GCC's effect intervened by the change of other factors. The third step develops the climate-blackout response model by integrating temperature-demand, temperature-SOB, and power flow relationships. The fourth step quantifies GCC's impact on blackout risk and compares regional vulnerabilities. The final step proposes the adaptation strategies to address GCC-induced reliability risks.

The TB3R framework is able to analyze the vulnerability and adaptation of the real-world DG within specific regions. To gain a deeper understanding of the chain effect and the influences of technological advancements and socioeconomic development, we employ a two-step analysis. The first step is to analyze the pure effect, excluding the interferences of other natural and human activities on DG blackout risk. The pure effect clarifies whether and when the GCC-

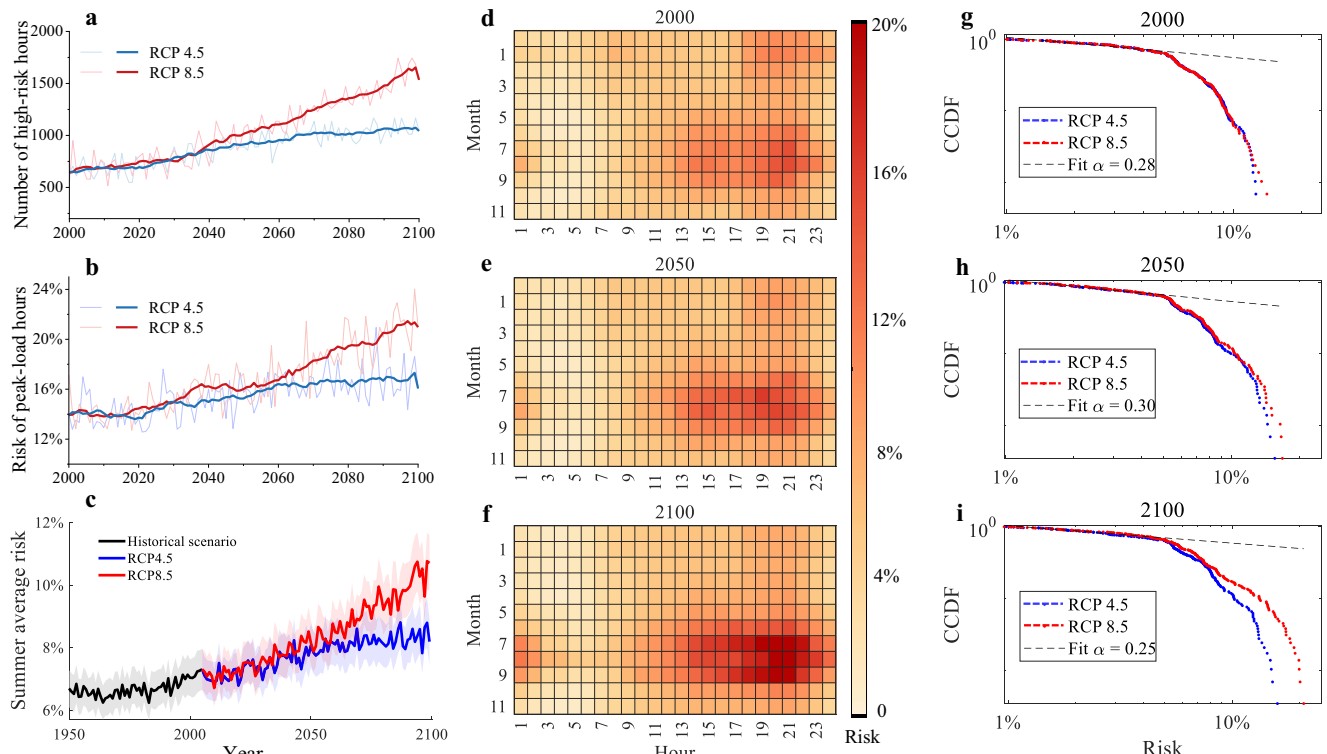

**Fig. 3 | Assessment of blackout risk under Representative Concentration Pathway (RCP) 4.5 and RCP 8.5. a** High-risk hours (risk > 10%) from 2000 to 2100 under RCP 4.5 (light blue) and RCP 8.5 (light red), with smoothed trends (dark blue and dark red). **b** Risk trends of peak-load hours (annual highest load hour, typically in summer) under RCP 4.5 (light blue) and RCP 8.5 (light red), with smoothed trends (dark blue and dark red). **c** Average summer risk from 1950 to 2100 under RCP 4.5 (blue), RCP 8.5 (red), and historical records until 2005 (black). The thick line shows the average of 20 Coupled Model Intercomparison Project Phase 5 models, with 95% confidence intervals (shaded envelopes). Uncertainty includes climate change and consumer distribution effects. **d**–**f** Hourly risks under RCP 8.5 in 2000, 2050, and 2100. The color bar indicates blackout risk for Feeder 1, with darker colors representing higher risk. A typical day represents each month. **g**–**i** Log-log plots of Complementary Cumulative Distribution Function (CCDF) for hourly risk under RCP 4.5 (blue) and RCP 8.5 (red) in 2000, 2050, and 2100. The x-axis shows risk, and the y-axis shows CCDF, both on logarithmic scales. Dashed lines indicate fitted power-law distributions. Downward tail curvature suggests fewer extreme events than a pure power-law. RCP 8.5 shows increasingly heavier tails over time, indicating more extreme risk events compared to RCP 4.5. Source data are provided as a Source Data file.

driven temperature changes have begun to manifest in the studied area, as well as to understand the characteristics of the GCC's combined impacts through the chain effect. The second step examines the interfered effect in various future technological and socioeconomic scenarios. The interfered effect reveals how GCC interacts with other factors, such as GDP growth, to influence DG reliability.

The TB3R framework compares GCC's effect on maintaining reliable DG service, highlighting geographic variability. As rising temperatures increase peak electricity demand, distribution reliability becomes more challenging and costly. To quantify this, we developed an index measuring the minimal DG size needed for reliable distribution. Standardized via the IEEE DG model, this index (measured in *p.s.m.*) captures how GCC-driven changes nonlinearly alter DG service. Analyzing it over time and regions reveals how power system physics shapes GCC's evolving impact on distributed electricity.

## Micro perspective of analyzing combined impacts of GCC through chain effect

Using the TB3R framework of model integration, we analyze the blackout risk of real-world DGs in California, which has seven feeders covering both urban and suburban areas (Supplementary Notes 7 and 8). We evaluate risks under two GCC scenarios derived from greenhouse gas concentration trajectories used in climate models: Representative Concentration Pathway (RCP) 4.5 and RCP 8.5 (detailed method is provided in the "Methods" section).

The pure effect of GCC's combined impacts has been increasing for decades and will intensify further this century. The pure effect would increase the studied DG's average blackout risk in summer by approximately 0.84% from the average level observed in the 1950s to that of the 2020s if everything but the global climate remained the same. This effect is projected to intensify over the coming decades: summer blackout risk is projected to climb 8.15–11.89% by the 2050s. GCC's impact concentrates on vulnerable hours but also extends its duration. Blackout risks exceeding 10% will rise 30% from 2000 to 2050 (650 to 950 h, Fig. 3a). Extreme peak-load blackout risk will increase 5% over the next two decades under both RCP scenarios (Fig. 3b). Even with no other changes, GCC's pure effect under RCP 8.5 is projected to double blackout risk by 2100 compared to the 1950s (Fig. 3c).

The TB3R framework reveals the nonlinear impact of GCC on DG blackout risk shaped through the chain effect, with implications for investment and management decisions. By comparing the plots in Fig. 3a–c, we observed the changes in slope of the blackout risk's growth trajectory in the rest of this century. For example, in Fig. 3c, under the RCP 8.5 scenario, the impact of GCC on summer-average blackout risk reaches a tipping point around 2050. After this point, the average blackout risk grows at a faster rate compared to earlier years. This nonlinearity is evident in different aspects of DG blackout risk, as seen in the varying curves for different scenarios.

In contrast, the risk trajectory for peak-load hours shown in Fig. 3b reveals multiple tipping points. In the RCP 8.5 scenario, the GCC-driven

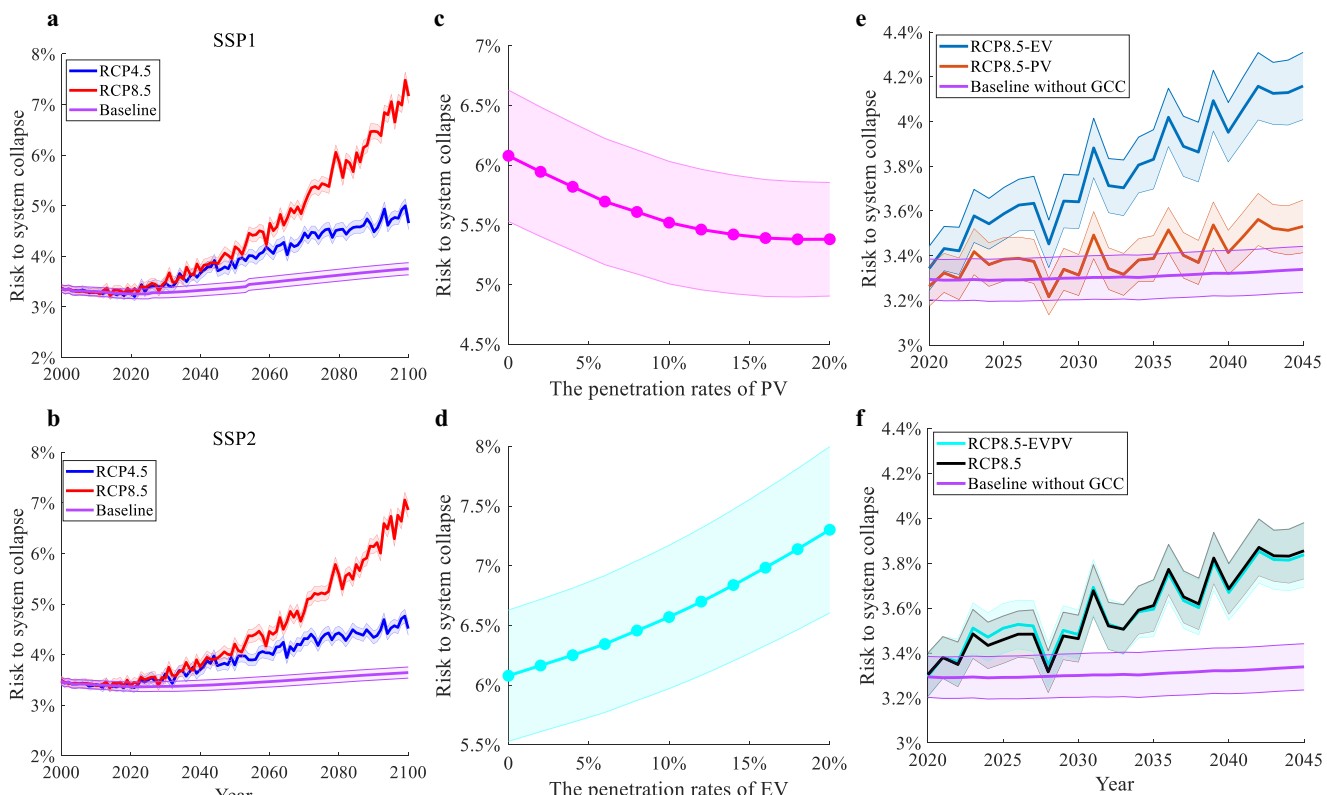

**Fig. 4 | Impact of photovoltaic (PV) and electric vehicle (EV) penetration on system collapse risk under representative concentration pathway (RCP) 4.5 and RCP 8.5 scenarios. a** Average summer risk of system collapse under Shared Socioeconomic Pathway (SSP) 1 for baseline, RCP 4.5, and RCP 8.5 scenarios, considering grid capacity expansion proportional to Gross Domestic Product (GDP) growth. **b** Average summer risk under SSP2 for baseline, RCP 4.5, and RCP 8.5 scenarios, with grid capacity expansion proportional to GDP growth. The risk increase is slightly lower compared to SSP1. **c** System collapse risk in 2050 under RCP 8.5 as a function of PV penetration rates. Increased PV penetration reduces risk, indicating renewable energy integration mitigates climate change impacts.

**d** System collapse risk in 2050 under RCP 8.5 as a function of EV penetration rates. Increased EV penetration slightly raises risk, suggesting higher EV demand strains the grid. In all panels, the thick line shows the average of 20 Coupled Model Intercomparison Project Phase 5 (CMIP5) models, with 95% confidence intervals (shaded envelopes). **e** Average risk under SSP2 and RCP 8.5, with separate growth rates for PV and EV penetration, showing varied impacts. **f** Average risk under SSP2 and RCP 8.5, with simultaneous growth of PV and EV penetration, showing a compounded risk effect and higher blackout risk compared to baseline. Source data are provided as a Source Data file.

risk increase accelerates in the 2030s, flattens in the 2040s, and then accelerates again in the 2050s. This nonlinear behavior arises from complex interactions between demand and DG engineering factors. On the demand side, electricity consumption responds nonlinearly to temperature changes. As local temperatures exceed certain thresholds due to GCC, electricity demand surges sharply because of increased use of air conditioning and other weather-responsive equipment, creating tipping points in risk growth.

Because of the chain effect governed by Kirchhoff's laws and the DG topology, the same demand increases at different nodes lead to heterogeneous changes in power flow and, consequently, different degrees of blackout risk changes. For example, an increase in electricity demand at nodes farther from the substation results in a greater rise in DG blackout risk compared to nodes closer to it. Thus, even if nodal demand exhibits a uniform response to temperature increases, the blackout risk will grow nonlinearly, with multiple tipping points emerging along the way. Additionally, consumers at different nodes often have varying threshold temperatures that trigger changes in energy consumption. As a result, the same temperature increase leads to an uneven rise in nodal loads, introducing a secondary effect that further amplifies the nonlinear relationship between the temperature changes and the blackout risk.

We argue that our findings on peak-load hours are applicable beyond California. In California DG, GCC has a stronger impact in the peak-load hours rather than off-peak-load hours because the demand

is more sensitive to the temperature in hotter scenarios. Literature has demonstrated that the relationship between DG demand and temperature in most areas in the world is represented by a U-shaped curve, where demand spikes during both hotter summer and colder winter hours, which are also peak-load periods[13,22]. Therefore, we conclude that GCC will similarly increase DG blackout risks during peak-load hours in most other regions in the world. Additionally, the power flow's effect, causing the highly nonlinear TB3R, also exists worldwide. Therefore, the GCC's polarization effect on demand will be further amplified when converting to the impact on the DG's blackout risk.

We further investigated the interfered effect of GCC's combined impact in various scenarios that have different GDP growth conditions or various energy transition progress. To examine the GDP-growth interfered effect, we set various scenarios according to the Shared Socioeconomic Pathways (SSP) framework[23]. We compared the interfered effect in the sustainability scenario(SSP1) and the Middle-of-the-Road scenario(SSP2). For each GDP growth scenario, we consider three scenarios involving varying demand, SOB, and GCC: (1) a benchmark scenario without GCC, (2) the RCP 4.5 scenario with moderate GCC, and (3) the RCP 8.5 scenario with significant climate change (detailed in "Methods").

The TB3R framework reveals how GCC and GDP growth interact, shifting the dominant factor in DG blackout risk over time. As shown in Fig. 4a, b, the GDP growth is the primary driving force before 2030. Consequently, while the pure effect analysis in the previous section

confirms the presence of the GCC's impact on power flow, it does not significantly exacerbate the average DG blackout risk beyond the GDP-driven effects. Correspondingly, the grid upgrades spurred by GDP growth effectively mitigate the blackout risk increase caused by temperature changes during this period. However, the GCC's impact on power flow introduces a substantial additional risk beyond that caused by GDP. These GDP-driven grid upgrades are insufficient to counteract the risk increases caused by temperature changes after 2030, as the GCC's influence on DG blackout risk remains relatively moderate before this point.

A further examination of the mixed effects of GCC and GDP growth on the DG power flow clarifies the conditions when the GCC's effect contributes significant extra blackout risk in addition to the GDP growth's effect. According to the pure effect analysis, there exists a threshold temperature beyond which the GCC's effect on DG's blackout risk will be accelerated when the nonlinear effect of the power flow dynamic converts the temperature change to the blackout risk increase. In the RCP 8.5 scenario, the local temperature will reach the threshold level around 2030. Consequently, the tipping point where the GCC's impact on blackout risk starts to escalate-occurs around 2020–2030, as indicated by the pure effect shown in Fig. 3c. After 2030, the rate of increase in blackout risk during peak-load hours and the number of vulnerable hours accelerates more rapidly in the GCC scenarios compared to the benchmark scenario. Correspondingly, DG expansion to accommodate GDP-driven demand growth will no longer counterbalance the effects of GCC. This divergence results in distinctly different risk trajectories for DG blackout across the three GCC scenarios for the remainder of the century.

GDP growth nonlinearly amplifies GCC's impact on DG blackout risk, varying across GDP scenarios. In the SSP 1 scenario, the summer-average blackout risk of Fig. 4a by 2100 is projected to more than double compared to 2000, increasing from 3.36 to 7.16% in the RCP 8.5 scenario. In the SSP 2 scenario, which assumes slower GDP growth, the same RCP 8.5 scenario results in a near doubling of blackout risk from 3.36 to 6.86% by 2100. The GCC's impact caused the blackout risk in the SSP 1 scenario to grow by an additional 0.3% compared to the SSP 2 scenario, which is nearly 9% higher. Therefore, the faster growth of GDP will amplify the GCC's impact. The additional 0.3% risk increase in the SSP 1 scenario is attributed to both a higher hourly blackout risk during peak-load hours and an increase in the number of vulnerable hours. The GDP growth will further amplify the GCC's impact in peak-load hours. Compared to SSP 2, the GCC impact in SSP 1 will lead the blackout risk growth 13.6% higher by the end of this century.

We discovered a complex mechanism by which GDP growth influences the impact of GCC by analyzing blackout risk differences between the SSP 1 and SSP 2 scenarios. Due to differing GDP growth trajectories, the growth rates of DG nodal demands diverge in these scenarios. As GCC causes local temperatures to rise, the joint distribution of DG nodal demands shifts in different directions under the two SSP scenarios. Furthermore, the relationship between demand and temperature is dependent on GDP levels, meaning that the same rate of temperature increase results in varied demand growth across SSP 1 and SSP 2. Additionally, GDP-driven changes in demand affect the extent of DG expansion. The interplay of these factors leads to different impacts on the power flow and thus causes divergent blackout risk change trajectories.

We also examined how energy transitions, via new technologies, interfere GCC's impact on DG reliability by altering nodal loads and power flow. For example, the Paris Agreement has catalyzed the development of PV and Electric Vehicles (EVs)[24]. Increased PV penetration enhances DG reliability by mitigating the power distracting from the nodes, which helps mitigate the risk of blackouts. However, the growth of EVs introduces higher loads and thus can amplify the GCC's impact. Therefore, the pace and mix of different technologies will critically determine how energy transitions affect DG blackout risk under the GCC process.

We assess how different technologies affect the impact of climate change on the blackout risk of California's primary distribution Feeder 1. A 20% PV penetration reduces DG blackout risk by 0.72% in 2050, fully offsetting RCP 4.5 impacts and significantly mitigating RCP 8.5 effects. In contrast, Fig. 4d indicates that a 20% penetration of electric vehicles (EVs) in the same area will increase the DG blackout risk by approximately 1.19% in 2050. This increase corresponds to a higher load on the distribution network nodes as EV penetration rises. Moreover, this effect is compounded by typical EV charging behaviors, as shown by a statistical analysis of EV usage patterns[25].

In the real world, GCC, GDP growth, and energy transition will simultaneously impact the DG's blackout risk. We examined the impact of the energy transition on the GCC's effects in the SSP 1 scenario. The penetration growth rates are set according to the Los Angeles 100% Renewable Energy (LA100) study conducted by the National Renewable Energy Laboratory (NREL)[26]. To analyze and compare the effect of various energy-transition technologies, we generated three energy-transition scenarios for the next two decades: the PV-only scenario, the EV-only scenario, and the scenario where the penetrations of PV and EV simultaneously grow. Our results in Fig. 4e suggest that the PV penetration rate, as projected in the LA100 study, is roughly sufficient to offset the GCC's impact if EV penetration is absent. However, if EV penetration grows according to the LA100 projections, Fig. 4f shows that the mitigating effect of PV will be offset. Consequently, the GCC's effect will still drive a substantial increase in DG blackout risk.

## Efficient adaptation by managing the chain effect

The TB3R framework identifies overlooked DG blackout risk factors, such as temperature-sensitive user locations and network topology. Figure 5 illustrates our experimental setup and results for evaluating the impact of network topology. For instance, we tested two types of network structures: a tree structure (Fig. 5d) and a mesh structure (Fig. 5h). For each structure, we analyzed vulnerable nodes under three scenarios: a benchmark scenario, a climate change scenario, and a climate change scenario with consumer reallocation. The third scenario is specifically designed to assess locational vulnerability. Figure 5a shows the joint distribution of loads $P_2$ and $P_3$, while Fig. 5b depicts the temperature-induced load increase for $P_2$.

In Fig. 5c, we swap the relationship in Fig. 5b to swap the locations. In Fig. 5c, we swap the locations of the loads from Fig. 5b to examine the effects of location changes. By applying these load conditions to the tree and mesh topologies in Fig. 5d and h, respectively, we generated six risk plots shown in Fig. 5e–g, i–k. For the benchmark cases in Fig. 5e and i, we observe that the mesh DG exhibits a blackout risk 12 times lower under the same conditions compared to the tree DG. For more detailed information about the experimental setup, please refer to the Supplementary Information. When comparing Fig. 5f and e, we see that increasing the load on Bus 2, which is near the reference bus, causes the blackout risk to rise from 41.48 to 51.22% in the tree topology. If the same load increase occurs at a leaf node (bus 3), the risk jumps to 73.55% (Fig. 5g). In contrast, under the same conditions in the mesh topology, the blackout risk remains significantly lower—3.39%, 9.15%, and 9.21% respectively (Fig. 5i–k). These findings demonstrate that a mesh DG topology is less sensitive to the locations of temperature-sensitive users, underscoring their resilience to changing conditions. Conversely, the blackout risk for tree distribution systems remains considerably higher, especially when temperature-sensitive users are connected to remote nodes.

These comparative analyses highlight the need to rethink DG planning to mitigate climate change risks. We strongly recommend adopting a mesh structure to enhance the reliability of distribution

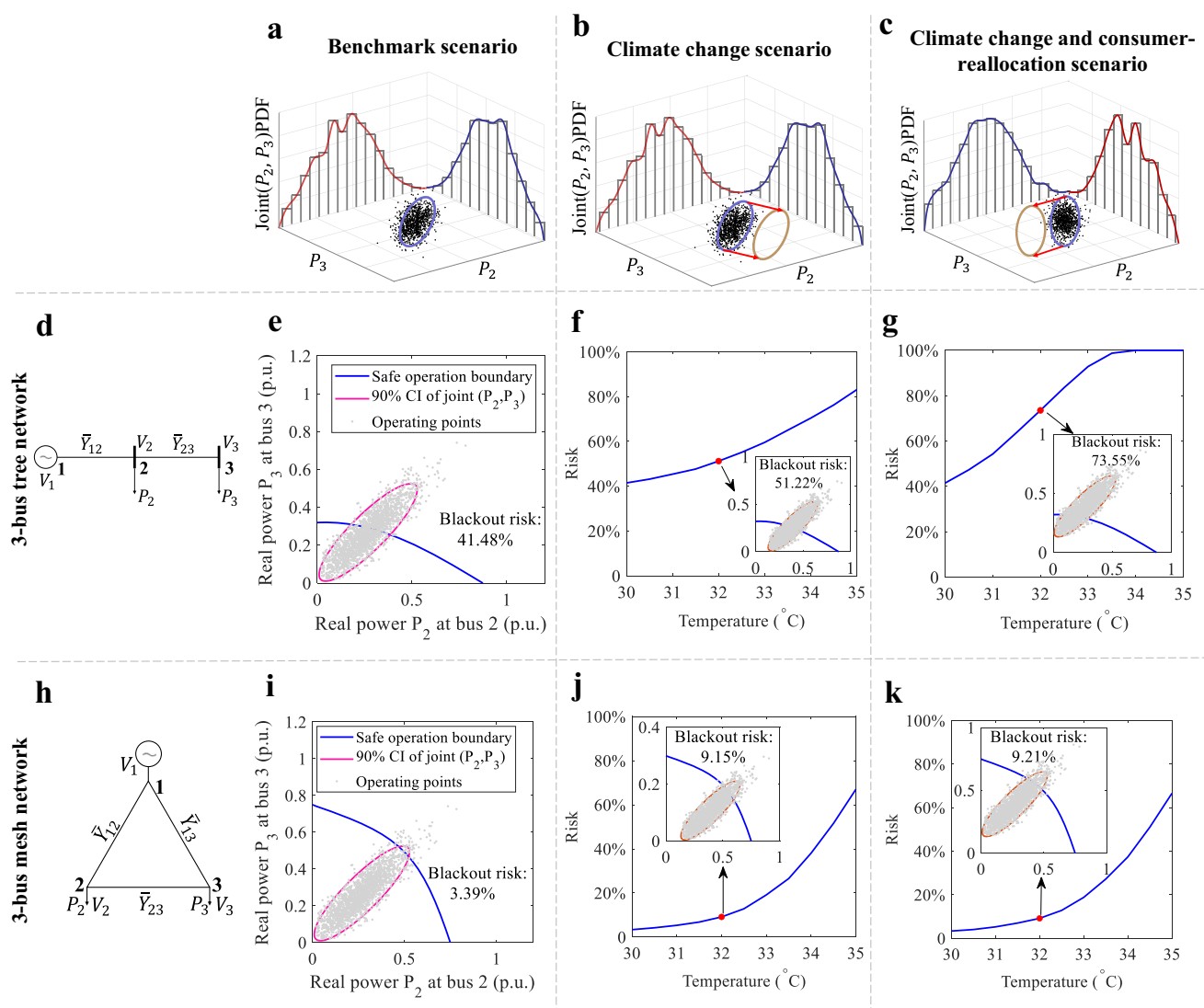

**Fig. 5 | Dependence of distribution grid vulnerability on the type of consumers' electricity demand and their location. a–c** Load distribution under different scenarios. **a** Benchmark scenario: Consumer on Node 2 is more temperature-sensitive than on Node 3 at low temperature (15 °C). **b** Climate change scenario: Consumers' locations are the same as in the Benchmark Scenario, but the temperature is high (30–35 °C). Load on Node 2 increases while load on Node 3 remains unchanged, causing the operating point to move along the $P_2$ axis. **c** Climate change and consumer-reallocation scenario: Consumers' locations are swapped from the climate change scenario under the same temperature (30–35 °C). Load on Node 3 increases while load on Node 2 remains unchanged, causing the operating point to move along the $P_3$ axis. **d, h** Network structures: $V_1$, $V_2$, and $V_3$ represent voltages at Nodes 1, 2, and 3, respectively. $\overline{Y}_{12}$, $\overline{Y}_{23}$, and $\overline{Y}_{13}$ represent purely resistive distribution lines. **d** A three-bus tree network with a substation at Node 1. **h** A three-bus

mesh network with a substation at Node 1. **e–g, i–k** Risk analysis under tree network and mesh network, respectively: The blue curve indicates the safe-operation boundary (SOB) of the power grid. Scatter points represent operation points, and a 90% confidence ellipse shows data distribution characteristics. Quantities are expressed in per unit (*p.u.*), defined as fractions of a base unit. **e, i** Benchmark scenario: Overlap of the joint distribution of the benchmark scenario and the SOB of the tree network (**e**) and mesh network (**i**). **f, j** Climate change scenario: Risk values for the tree network (**f**) and mesh network (**j**) under the climate change scenario for different temperatures. **g, k** Climate change and consumer-reallocation scenario: Risk values for the tree network (**g**) and mesh network (**k**) under the climate change and consumer-reallocation scenario for different temperatures. Source data are provided as a Source Data file.

systems in the face of climate change. Currently, a large proportion of distribution systems worldwide are designed as tree networks, with mesh-structured distribution systems primarily found in critical areas such as metropolitan or industrial zones. However, upgrading to a mesh network in a distribution grid may be more feasible than expected due to the relatively short node distances at low voltage levels. We also suggest a vital strategy for DG adaptation, which involves strategically deploying distributed energy technologies at vulnerable nodes with sensitive users to enhance overall system resilience. Considering technology features in relation to user sensitivity and nodal vulnerability is crucial for effectively planning the implementation of distributed energy technologies.

## Macro perspective of geographically heterogeneous stresses from GCC to regional distribution services

Beyond micro-level impacts useful for utility planning, large-scale assessments are crucial for legislative bodies to target vulnerable areas and develop adaptation strategies. Therefore, we compared the minimum reliable standard indices for distributed generation across 47 American states and 33 European countries at different scales. Such a comparison uses U-curve and RCP 8.5 to include factors such as demographic, socio-cultural, and economic characteristics. The results, presented in Fig. 6, highlight the geographic heterogeneity in the GCC-induced stress on electricity distribution reliability. Recall that we considered both RCP 4.5 and RCP 8.5 scenarios in our micro-

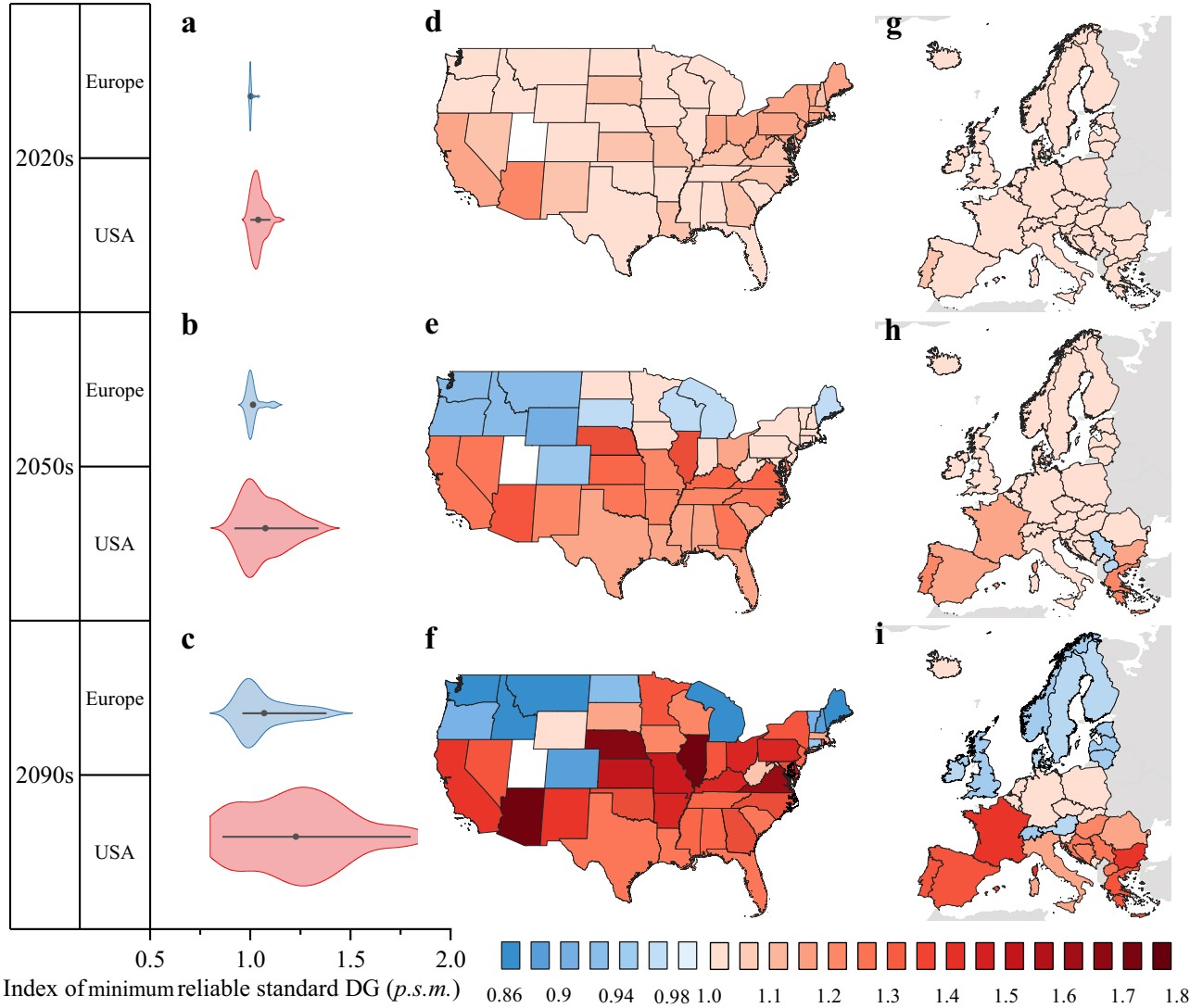

**Fig. 6 | The change of the minimum reliable standard distribution grid index in 2020–2029, 2050–2059, and 2091–2100 for projected maximum load under the Representative Concentration Pathway (RCP) 8.5 scenario in Europe and the United States.** Each row represents changes in the distribution grid (DG) index for the 2020s, 2050s, and 2090s in the United States and Europe. In the first column (**a**–**c**), violin plots depict the distribution of index values for the United States (red) and Europe (blue) in each decade, with solid points indicating mean index values. The mean index in the United States is higher than in Europe, and this difference intensifies with climate change. The second column (**d**–**f**) and third column (**g**–**i**), respectively, present the distribution of index values for the United States and Europe in each decade using maps. In the map, the blue color on the graph represents countries or states with a standard-DG-model-based index of less than 1, indicating that climate change leads to a decrease in their load, thereby reducing pressure on the DG. On the other hand, the red color represents countries or states with $M_{c,y,m}^{min}$ values greater than 1, indicating that climate change exerts more pressure on their distribution network. Under the RCP 8.5 scenario, the projected changes in the distribution network affected by climate change are expected to vary geographically, with the largest increases observed in southern Europe and the southern United States. This trend becomes most apparent in the 2090s but is already emerging in the 2050s. The maps in this study were generated using QGIS (version 3.32). The base map is a global vector map downloaded from Natural Earth (version 5.1.1), which is in the public domain and freely available for use. For more information, see the Natural Earth terms of use[43]. Source data are provided as a Source Data file.

scale study. The two scenarios demonstrated similar trends, with the results under RCP 8.5 being more pronounced. As the purpose of macro-scale research is to identify damaged and benefited areas, RCP 8.5 can better reflect the damaged and benefited areas[27,28]. Therefore, we focus on the RCP 8.5 for macro analysis in this section[29].

GCC's impact on electricity distribution varies widely. Some regions face severe stress, while others benefit from reduced strain. DG-benefit areas experience lower demand or conditions that ease grid stress, enhancing reliability. In contrast, DG-vulnerable areas face higher demand, extreme weather, or other climate-related factors that heighten grid strain and blackout risks.

GCC will lead to significant divergence between DG-benefit and DG-vulnerable areas. However, the pace of this divergence will differ between the U.S. and Europe as we look toward the future. In the U.S., GCC is expected to cause substantial divergence among states. This is evident when comparing the color changes between 2020 and 2050 for the U.S. map in Fig. 6d and e. The changes in colors show that 55% of the states need to upgrade DG to ensure reliable distribution services for users, whose demand-weather response relationship remains the same.

Meanwhile, 45% of states are expected to benefit from the GCC-driven temperature changes. When examining extreme cases, the gap between the highest and lowest minimum reliable standard DG index values grows from 0.14 *p.s.m.* in the 2020s to 0.42 *p.s.m.* by 2050. By mid-century, 13 states will see their minimum reliable standard DG index increase by over 10%, with 6 of them experiencing expansions of

20% or more. Conversely, 5 DG-benefit states will see their reliability index decrease by 5%. After 2050, the divergence between states is projected to accelerate further; the largest gap could reach 0.94 *p.s.m.* in the 2090s-nearly seven times higher than in 2020. During this period, 19 DG-vulnerable states could see their indices increase by over 30%, while 7 DG-benefit states might experience a reduction of more than 10%.

Similarly, Fig. 6g–i shows a slower and more moderate divergence in Europe compared to the U.S. In southern European countries, the need for upgraded distribution systems will increase due to climate change, but the impact remains moderate. For instance, changes in reliability indices across all 33 European countries remained consistently below 5% before 2050. Even in the 2090s, the gap between the highest and lowest indices in Europe only reaches the level of divergence seen in the U.S. in the 2050s.

The divergence of DG-benefit from DG-vulnerable areas is driven by both geographic and socioeconomic reasons. A large number of DG-benefit states are concentrated in North America, where peak loads are primarily driven by low temperatures. As GCC raises local temperatures, power demand decreases, thereby improving DG reliability between 2020 and 2050, as shown in Fig. 6d and e. However, due to the chain effect, temperature is not the only factor causing the geographic difference; demand-temperature sensitivity also determines how severely GCC affects a region's ability to provide reliable electricity distribution. For example, despite being one of the most northern states, Minnesota is still vulnerable to GCC. This vulnerability is due to the state's high sensitivity of electricity demand to temperature changes, causing Minnesota's minimum reliable standard DG index to increase by 30% by the end of the century. Such a result demonstrates Minnesota's vulnerability to GCC, driven by temperature-sensitive electricity demand based on demographic, socio-cultural, and economic factors in this region.

Because TB3R captures power flow dynamics of the chain effect, we can examine and compare the tipping points when the DG blackout risk of various regions will be rapidly exacerbated by the GCC-driven temperature change. Some regions may become more vulnerable than others at different times due to changes in the nonlinear relationship between risks and years in the TB3R model. For example, the temperature-demand relation has a faster growth rate in South Carolina than in Minnesota as local temperatures rise from current levels. As a result, by 2050, the minimum reliable standard DG index is projected to increase to 1.1 *p.s.m.* in South Carolina while remaining around 1 *p.s.m.* in Minnesota. However, if GCC continues to push temperatures higher, Minnesota's temperature-demand relation will have a faster growth rate than South Carolina's after 2050. Consequently, by the 2090s, the index will reach 1.18 *p.s.m.* in South Carolina but surge to 1.3 *p.s.m.* in Minnesota, indicating that Minnesota will be more vulnerable at the end of the century. These findings suggest that the timing of investments in the GCC's adaptation should be tailored to the unique circumstances of each region.

## Discussion

This study reveals that GCC's combined impacts on DG's blackout risk through the chain effect have unique features, differing from extreme weather events that cause immediate infrastructure damage. Even if a weather event is insufficient to destroy any single component of a DG system, the combination of impacts on all components together can still lead to a DG blackout. The nonlinearity of the chain effect's intermediate process further amplifies the blackout risk's occurrence probability. Therefore, DG can be more vulnerable to GCC than previously expected, as many components can be influenced.

The blackout risk's growth trajectory with multiple tipping points further emphasizes the necessity of focusing on the nonlinear change of infrastructure reliability driven by GCC. Many studies have focused on the threats to power grids and other infrastructure from the nonlinear progress of meteorological changes during GCC. Our discoveries show that the steady GCC progress can still lead to a cliff-like drop in DG reliability, which reminds policymakers and infrastructure managers to pay attention to how the infrastructure's physical dynamics interact with the GCC progress.

Our findings further indicate the necessity of a broader perspective for rethinking DG planning for climate change adaptation. For instance, we found that mismatches between temperature-sensitive electricity consumers and vulnerable grid nodes increase DG susceptibility to GCC-driven temperature fluctuations. This suggests that adaptation strategies should prioritize reinforcing weak grid nodes where critical loads are concentrated. Strategic deployment of distributed PV systems and energy storage in these locations could mitigate supply-demand imbalances, enhancing DG resilience in a cost-effective manner. The structure of DG networks further influences vulnerability, with variations in grid topology significantly shaping the degree of risk exposure. Optimizing DG network design based on regional climate conditions and socioeconomic factors presents a promising yet underexplored avenue for increasing resilience[30].

The results also demonstrate the importance of region-specific adaptation strategies. Our macro analysis reveals that two regions experiencing similar temperature changes can exhibit significant differences in electricity distribution vulnerability due to differences in infrastructure investment, energy policy, and socioeconomic conditions[31]. Furthermore, our spatial and temporal analysis highlights the polarizing effect of GCC on electricity distribution reliability, with distinct trends emerging between the United States and Europe[32]. These observations emphasize the need for localized mitigation strategies that integrate climate risks with existing infrastructure limitations.

Accurately capturing the above mechanisms requires an integrated modeling framework that reflects the interdependencies between climate, infrastructure, and electricity demand. The TB3R framework shows the importance of interdisciplinary model integration for analyzing GCC's impact on large-scale and complex infrastructure systems in human society.

This study has certain limitations. First, the dynamic relationship between electricity demand and temperature under evolving socioeconomic conditions remains difficult to model with precision. Behavioral shifts, energy efficiency improvements, and technological advances could alter the demand response to GCC beyond the projections used in this study[14,24]. Second, while the widely used SSP scenarios form the basis of our analysis, alternative economic trajectories, urbanization trends, and policy innovations could lead to different outcomes[23]. Expanding the range of modeled scenarios could provide a more comprehensive understanding of potential DG vulnerabilities. Finally, uncertainties inherent in climate projection models, particularly under extreme conditions, remain a challenge for long-term impact assessments. The incorporation of probabilistic approaches and ensemble modeling techniques could improve the robustness of future analyses.

These findings highlight the critical role of integrated climate-energy modeling in designing effective adaptation strategies for DG systems. By bridging power system analysis with climate science, this study offers a framework for evaluating long-term risks and informing resilient infrastructure planning. Future research should explore how grid topology modifications and adaptive control strategies can enhance DG resilience, ensuring the continued reliability of decentralized electricity generation in the face of GCC-driven uncertainty.

## Methods

This paper presents a framework of model integration to assess the pressure and risk on controllable power distribution services. This section provides a detailed introduction to the framework. The first four subsections form the core of the integrated model: (1) "Climate

change model," which predicts hourly temperatures from 1950 to 2100; (2) "Learning power consumption behavior of consumers" subsection, which forecasts hourly load consumption based on historical temperature and time-of-day patterns; (3) "Calculating the SOB of DG system method," which determines the operational limits of DG systems under predicted load conditions; and (4) the "Risk assessment method," which evaluates the risk of system collapse at each bus using the predicted load and SOB data. These interconnected subsections provide a comprehensive foundation for analyzing the impacts of climate change on power distribution systems.

Building on the integrated model, the subsequent subsections focus on analysis and experimental design. "Characterizing the climate change's impact and defining the index" quantifies the pressure exerted by climate change on stable distribution, introducing a standardized index to compare vulnerabilities across regions. "Modeling the impact of economic growth and climate on energy demand and DG system expansion" subsection extends the analysis to incorporate economic factors, exploring how energy demand and DG system expansion interact with climate change. The "Experiment design" subsection outlines the setup for testing the framework under various scenarios. Together, these subsections provide a robust methodology for evaluating and mitigating the risks posed by climate change to power distribution systems.

## Climate change model

To understand the impact of climate change in different areas, there is a sequence of studies and projects that develop the models to calibrate the local temperature change in different areas on the earth[33,34]. Here, we use the American data from the U.S. Geological Survey (USGS) National Climate Change Viewer (NCCV) and the European data from the Climate Change Knowledge Portal (CCKP)[35]. The USGS NCCV dataset includes the historical and future climate projections from 20 downscaled models for two RCP emission scenarios[36]. The dataset consists of the temperature trajectory of all counties in the U.S. Two climate change scenarios proposed by the IPCC are considered: the RCP 4.5 and RCP 8.5 scenarios. The specific 20 Coupled Model Intercomparison Project Phase 5 (CMIP5) models are shown in Supplementary Table 1. The mean values of downscaled 20 CMIP5 models are shown in Supplementary Fig. 8 of Supplementary Information.

## Learning power consumption behavior of consumers

For the case study in California, we analyze the temperature versus load consumption response curves for all the consumers in the available dataset to estimate consumer behavior. This is made possible by fusing the power consumption data of unique users from real-world consumer databases and the temperature data from the weather API of the National Oceanic and Atmospheric Administration (NOAA)[37], using their hourly resolution timestamps. We observe that these temperature response curves are U-shaped and roughly symmetric around an equilibrium temperature value, as shown in Supplementary Fig. 9a of the Supplementary Information.

For Europe, we utilized regression curves of country-level-aggregated peak load versus daily average temperature, as disclosed by Leonie Wenz[14]. Meanwhile, for the United States, we employed the response functions of electricity peak load versus daily average temperature, as disclosed by Maximilian Auffhammer[13]. Given that counties within each state in the U.S. are served by different electric utilities, we selected representative U-shaped curves of typical power companies within each state as the load temperature response functions. The correspondence between the power companies and states is presented in the Supplementary Information. We retrieved the corresponding data using the *getdata* tool and performed curve fitting to obtain the U-shaped curves for each state and each country.

## Calculating the SOB of DG system

The SOB of a DG system is defined as the margin to blackout at which the grid can operate safely without collapsing. In our analysis, the geometric circles of the power flow have nice properties for deriving the loading capacity bounds[20]. We adopt the rectangular coordinates for complex voltages. Then, the power flow equations are shown in equations (2)–(5).

$$P_i = d_{i,1} \cdot v_{i,r}^2 + d_{i,2} \cdot v_{i,r} + d_{i,1} \cdot v_{i,m}^2 + d_{i,3} \cdot v_{i,m}, \qquad (2)$$

$$Q_i = d_{i,4} \cdot v_{i,r}^2 - d_{i,3} \cdot v_{i,r} + d_{i,4} \cdot v_{i,m}^2 + d_{i,2} \cdot v_{i,m}, \qquad (3)$$

where

$$d_{i,1} = -\sum_{k \in \mathcal{N}(i)} g_{ki}, \qquad d_{i,2} = \sum_{k \in \mathcal{N}(i)} (v_{k,r} g_{ki} - v_{k,m} b_{ki}), \qquad (4)$$

$$d_{i,3} = \sum_{k \in \mathcal{N}(i)} (v_{k,r} b_{ki} + v_{k,m} g_{ki}), \qquad d_{i,4} = \sum_{k \in \mathcal{N}(i)} b_{ki}, \qquad (5)$$

where $\mathcal{N}(i)$ represents the neighbors of bus $i$, and $P_i$ and $Q_i$ are the active and reactive power injections at bus $i$. The voltage at bus $i$ is represented by the complex phasor $v_i$. In rectangular coordinates, $v_{i,r} = |v_i| \cos \theta_i$ and $v_{i,m} = |v_i| \sin \theta_i$ represent the real and imaginary parts of the complex voltage at bus $i$, respectively. $\theta_{ik} = \theta_k - \theta_i$ denotes the phase angle difference between buses $k$ and $i$, and $g_{ki}$ and $b_{ki}$ are the electrical conductance and susceptance between bus $i$ and bus $k$. Together, $y_{ki} = g_{ki} + j \cdot b_{ki}$ forms the admittance, where $j$ is the imaginary unit.

For fixed constants $d_{i,1}, d_{i,2}, d_{i,3}, d_{i,4}$, equations (2) and (3) describe two circles in the $v_{i,r}$ and $v_{i,m}$ space. The coordinates of the circle center $E$ for the active power flow are $\left( -\frac{d_{i,2}}{2d_{i,1}}, -\frac{d_{i,3}}{2d_{i,1}} \right)$ for bus $i$, and its radius decreases as $P_i(t)$ increases. Similarly, the center $D$ for the reactive power flow is located at $\left( \frac{d_{i,3}}{2d_{i,4}}, -\frac{d_{i,2}}{2d_{i,4}} \right)$, and its radius decreases as $Q_i(t)$ increases.

Thus, if the active power circle and the reactive power circle do not intersect, the DG system will experience a blackout. This condition implies that an operating point on the SOB is tangent to both circles at a single point. Geometrically, a point lies on the boundary if there is no other point that can consume more power.

**Theorem 1.** Checking whether an operating point is on the boundary of the feasible power flow region is equivalent to solving a linear programming problem, as shown in equations (6)–(8)

$$\min_{\mathbf{z}} 1 \qquad (6)$$

$$\text{s.t. } \mathbf{z}^T \mathbf{h}_i \geq 0, \text{ for all } i = 1, \ldots, n, \qquad (7)$$

$$\sum_{i=1}^{n} \mathbf{z}^T \mathbf{h}_i = 1, \mathbf{h}_i = \left[ \frac{\partial P_i}{\partial v_{1,r}} \quad \frac{\partial P_i}{\partial v_{1,m}} \quad \frac{\partial P_i}{\partial v_{2,r}} \quad \cdots \quad \frac{\partial P_i}{\partial v_{n,m}} \right] \in \mathbf{R}^{2n}, \qquad (8)$$

where $\mathbf{h}_i$ is the gradient of $P_i$ with respect to all state variables. Therefore, $\mathbf{h}_i$ is the transpose of the $i$-th row of the Jacobian matrix. Let $\mathbf{z} \in \mathbf{R}^{2n}$ be a direction in which the real and imaginary parts of the voltages are moved.

A point is on the boundary if no direction exists in which the consumption at one bus can increase without decreasing consumption at others. Therefore, we can check if there is a direction $\mathbf{z}$ that makes the optimization problem feasible. The objective function is irrelevant

as we are only concerned with feasibility. Finally, an operating point is on the boundary if and only if the problem in equations (6)–(8) is infeasible. Similarly, if we find a unit vector $\mathbf{z}$ such that the sum of the active power is maximized, the value of the optimization problem, denoted by $\Phi$, represents the margin to the SOB. Therefore, solving the following optimization problem yields the margin to the SOB, with the optimal value of the objective function being the margin itself, as shown in equations (9)–(11).

$$\Phi = \max_{\mathbf{z}} \sum_{i=1}^{n} \mathbf{z}^T \mathbf{h}_i \tag{9}$$

$$\text{s.t. } \mathbf{z}^T \mathbf{h}_i \geq 0, \qquad \text{for all } i = 1, \ldots, n, \tag{10}$$

$$\| \mathbf{z} \|_2 \leq 1. \tag{11}$$

The above mathematical correctness is supported by the works of refs. [38],[39]. As a result, our method is theoretically robust and provides high accuracy in analyzing the DG system's responses to climate change. For a DG system serving consumers withdrawing power from $N$ nodes, the margin to the SOB at a specific temperature $t$ can be captured by a function $\Phi[\mathbf{P}]$, where $\mathbf{P} = [P_1, P_2, \cdots, P_n]$, $P_i(t)$ represents the net demand at Node $i$ during temperature $t$[40]. This net demand is defined as the user load at a specific node, adjusted by subtracting the generated power from PV panels and adding the charging load of EVs. When $\Phi[\mathbf{P}] > 0$, the demand profile $\mathbf{P}$ can be safely supported by the DG system. When $\Phi[\mathbf{P}] < 0$, the grid collapses, resulting in a blackout. Therefore, $\Phi[\mathbf{P}] = 0$ represents the **SOB**.

## Risk assessment method

Now, the Monte Carlo simulation is proposed to assess the probability of a DG's operating condition violating the SOB due to climate change impact. The overall step-by-step procedure is as follows. First, given that the temperature model ($T$) due to climate change[41], a specific temperature ($t = T_j$) from a climate change scenario set. Second, the conditional distribution of the total demand on DG at hour ($h$) of the day $\mathcal{D}(\mathbf{P}|T_j, h)$ is learned. Then we can draw samples $(\mathbf{P}|T_j, h)$ of the user's conditional power demand distribution. For every sample drawn from $(\mathbf{P}|T_j, h)$, the method to calculate the margin to SOB ($\Phi$) above is used to estimate the distance between the sample and the SOB. Finally, by using Monte Carlo simulations, we can count the number of samples that drive the DG to operate in an unsafe region ($\mathcal{N}(\Phi[\mathbf{P}|T_j, h] \leq 0)$) and finally compute the risk as described in equation (12):

$$Risk_h = \sum_{j=1}^{S} p(T_j) \cdot p(\Phi[(P_1, P_2, \cdots, P_n)|T_j, h] \leq 0)$$
$$= \sum_{j=1}^{S} p(T_j) \cdot \frac{\mathcal{N}(\Phi[\mathbf{P}|T_j, h] \leq 0)}{\mathcal{N}(\mathbf{P}|T_j, h)}, \tag{12}$$

where $n$ represents the total number of nodes in the DG system. $S$ represents the total number of climate change scenarios. $p(T_j)$ denotes the probability of climate change scenario $j$. We use $Risk_h$ to develop three indices assessing the probability of a blackout occurring and the degree of exposure to the risk. The three indices is: (1) Risk in the peak-load hours, $\max(Risk_h, h \in year)$; (2) Number of exposure hours to high risk, $\sum_h I(Risk_h > \text{threshold})$; (3) the annual/seasonal average risk.

## Characterizing the climate change's impact and defining the index

For the macro impact, the climate-blackout response model calculates the minimal DG size to provide the same group of users with a reliable

electricity distribution service in different climate change scenarios to assess climate change's impact on reliable DG size. The design of the index must effectively reflect climate change's influence on the reliable electricity distribution service and facilitate the identification of vulnerable regions. Therefore, the index has to enable comparisons over years and regions. Here, we propose using the standardized IEEE distribution network model (IEEE 33-bus) for index design. The *index of minimal reliable standard* $DGM_{c,y,m}^{\min}$ in country $c$ during month $m$ of year $y$ is determined by minimizing the function $f$ in equation (13):

$$M_{c,y,m}^{\min} = \arg\min_{M_{c,y,m}} f\left( \frac{L_{\text{IEEE-33}}}{\max(L_{c,2020})} \cdot L_{c,y,m}, M_{c,y,m} \right) \tag{13}$$

Here, $M_{c,y,m}^{\min}$ represents index of minimal reliable standard DG for country $c$ in year $y$ and month $m$ to prevent blackouts, $L_{\text{IEEE-33}}$ is the baseline load of the IEEE-33 bus standard distribution model, $\max(L_{c,2020})$ denotes the maximum load of country $c$ in the year 2020, $L_{c,y,m}$ is the actual load of country $c$ in year $y$ and month $m$, and $M_{c,y,m}$ stands for the distribution network expansion factor. The function $f$ simulates the actual distribution network operation based on the scaled load and expanded distribution network. To assess the impact of climate change on different regions, we consider a group of users whose total maximum demand in 2020 equals the total maximum load of the IEEE-33 bus model and is distributed over nodes according to the standard model setting. Therefore, we assess $M_{c,y,m}^{\min}$ in units of *per standard model* (*p.s.m*). We employ the standardized IEEE distribution network model (IEEE-33 buses) to construct comparable indicators of climate change's effects in each area. These indicators represent the minimum multiples by which the standard IEEE-33 node model needs to be expanded to avoid blackout risks under given demand scenarios.

To evaluate the proportional increase in investment needed in the power grid from 2020 to 2100 (over 80 years) for different countries to achieve 100% reliable distribution services, we assess $M_{c,y,m}^{\min}$ in units of *p.s.m*. This evaluation is achieved by scaling the load of different countries to the baseline load level of IEEE-33 in 2020. When the load is equal to the standard load of IEEE-33, the minimum expansion factor of the distribution network is 1. Thus, we need to determine this scaling factor. We calculate the scaling factor by dividing the maximum load value in 2020 by the baseline load of IEEE-33. We choose the maximum load value in 2020, $\max(L_{c,2020})$, as the baseline because it represents the peak power demand experienced at present. By comparing the actual load ($L_{c,y,m}$) in future years with the 2020 baseline, we can evaluate how much the distribution network needs to be expanded or contracted (represented by $M_{c,y,m}$) to accommodate the increasing or decreasing load due to climate change and ensure uninterrupted power supply.

Therefore, $M_{c,y,m}^{\min}$ values less than 1 indicate that climate change leads to a reduction in the load in the country $m$, thus reducing the pressure on the distribution network. Conversely, larger values of $M_{c,y,m}^{\min}$ indicate greater pressure on the distribution network caused by climate change.

## Modeling the impact of economic growth and climate on energy demand and DG system expansion

In this model, the impact of economic growth on electricity demand is incorporated through the per-capita GDP adjusted growth factor. As the economy grows, it typically drives higher energy demand due to increased industrial activity, commercial growth, and residential usage. To capture this effect, we introduce GDP growth as a key variable in the calculation of the annual demand mean. By combining GDP with heating degree days (HDD, $T < 12.5\,°C$) and cooling-degree days (CDD, $T > 27.5\,°C$), the model comprehensively accounts for both economic and climate influences on the energy system. The coefficients used for each factor are based on the values provided in the studies[22].

The annual demand mean for each year is calculated in equation (14):

$$\mu_y = gdp_b \cdot GDP(y) + hdd_b \cdot HDD(y) + cdd_b \cdot CDD(y), \qquad (14)$$

where the terms $GDP(y)$, $HDD(y)$ and $CDD(y)$ represent the values of GDP, HDD, and CDD in year $y$, respectively. The coefficients are $gdp_b = 0.3763$, reflecting the impact of economic growth (GDP) on energy demand. $cdd_b = 0.0103$, representing the influence of HDD on energy consumption. $hdd_b = 0$, indicating no significant effect from CDD in the reference model.

Next, based on the historical data (2015–2019), we derive the U-curve, which captures how consumers' electricity demand responds to temperature variations. The U-curve illustrates how demand changes with temperature, with higher demand during extreme heat or cold, and lower demand at moderate temperatures.

The hourly demand for each future year $y$ is calculated by adjusting the 2019 baseline demand with a scaling factor based on the U-curve and the annual demand gap in equation (15):

$$D_{y,h} = \alpha \cdot (\mu_y - \mu_{2019}) + U(t_h), \qquad (15)$$

where $\alpha$ is calculated as the ratio of the demand mean derived from the U-curve in 2019 to the baseline demand mean for 2019, $\mu_{2019}$. The DG system expansion for year $y$ is proportional to the growth in annual demand relative to 1950, ensuring the system scales with increasing electricity demand.

## Experiment design

The experimental design for the 3-node discussion includes the following steps: (1) determining the benchmark case and treatments, (2) setting the experimental environment, (3) setting the observed variables, and (4) analyzing the experiment results. The experiments are implemented in the blackout-response model to verify how various features shape the magnitude of climate change's impact through complex power-flow dynamics. For analytical examination, we chose a 3-node DG system serving real-world sampled consumers as the experimental environment. The risk of blackout is selected as the observed variable. In each experiment, we observe how the blackout risk changes when the temperature increases from 30 to 35 °C.

To identify key influences, we outline potential features affecting the magnitude of climate change's impact. According to the climate-blackout response model, blackout risk increases as climate change drives demand to exceed the DG's SOB. There are two types of features that influence the magnitude of this impact. The first type relates to the factors driving the speed of demand growth due to climate change. The second type involves technical characteristics affecting the DG's SOB, such as grid capacity and network topology.

Given that most current DG systems are tree-structured, we use a tree network as the benchmark case. We considered features like demand temperature sensitivity, grid line capacity, network topology, and the location of temperature-sensitive users. Changing these features constitutes the treatment.

In the log-log plot of Fig. 3g–i, the data were first sorted in ascending order. Then, the Empirical Cumulative Distribution Function (ECDF) was calculated. The CCDF, defined as 1 − ECDF, was then calculated to represent the proportion of data points greater than a particular value. The downward curvature of the tails suggests lighter tails than a pure power-law, meaning fewer extreme events. Over time, the RCP 8.5 scenario shows increasingly heavier tails, indicating a higher frequency of extreme risk events compared to RCP 4.5.

## Limitation and potential future work on method

We show the potential limitations of current modeling. The first limitation comes from the simplified representation of the electricity demand model for the DG. Due to the lack of demographic details of household-level data, most electricity demand models in the current literature are regional rather than household. Consequently, there are few models about how the household electricity demand varies by household-level demographic features. Therefore, the electricity demand model integrated into the TB3R framework in this paper is regional and makes it difficult to clarify how the results are influenced by different demographic and social scenarios that have various distributions of race, income, and marital status. Further, the lack of a household-level model prevents us from applying the TB3R to analyze DGs directly connected to households. Future studies can integrate an improved household-level demand-side model into the TB3R framework for further research.

The second limitation comes from the simplified discussion of energy transition. Energy transition in DG can be self-motivated by the households or independent investors, who are influenced by the energy transition policies and individual environmental attitudes. Because this paper focuses on the chain effect of the GCC's combined impacts, it does not include related discussions to avoid distraction. Consequently, the conclusions in this work cannot reflect how users and investors respond rationally to GCC. In future studies, the TB3R framework can be combined with energy transition models to further discuss the influence of policy design and public attitudes on DG's vulnerability and adaptation to GCC.

The third limitation is the simplification between socioeconomic change and GCC's combined impacts. In this work, we addressed whether the blackout risk increase driven by GCC's combined impacts can be mitigated by the expansion of DG responding to the demand increase induced by GDP growth. However, the decisions on DG expansion and renewal can also be influenced by other factors such as price designs for energy storage. The absence of those details in the model disables the TB3R framework from comprehensively understanding socioeconomic changes and the DG blackout risk change due to GCC's combined impacts. For instance, before improving the DG expansion model, the TB3R framework makes it hard to discuss the effect of the market designs for energy transition.

The above method limitations restrict this paper's estimation on the size of GCC's combined impacts on DG's blackout risk in the discussed scenarios. However, these limitations do not affect the main conclusion of this paper, such as the tipping points of the nonlinear trajectory of DG blackout risk and the insights for DG's vulnerability and adaptations to the GCC's combined impacts through the chain effect. Future studies may benefit from the TB3R framework's detailed modeling of the chain effect's intermediate processes. The improved socioeconomic models, policy models, and energy transition models can be integrated into TB3R to examine the corresponding change of SOB and power flow.

## Data availability

The historical and future climate projections from 20 downscaled models for two RCP emission scenarios were sourced from https://doi.org/10.5066/F7W9575Thttps://doi.org/10.5066/F7W9575T. Hourly load consumption data for Feeder 1 were collected from 2015 to 2019, tagged with geolocation and timestamps. European load data were sourced from ENTSO-E (https://www.entsoe.eu/data/power-stats/https://www.entsoe.eu/data/power-stats/). For the U.S., we extracted the corresponding minimum and maximum loads for each company from the original databases, the Federal Energy Regulatory Commission Form 714 (https://www.ferc.gov/industries-data/electric/general-information/electric-industry-forms/form-no-714-annual-electric/datahttps://www.ferc.gov/industries-data/electric/general-information/electric-industry-forms/form-no-714-annual-electric/data) and Energy Information Administration Form 861 (EIA-861) (https://www.eia.gov/electricity/data/eia861/https://www.eia.gov/electricity/data/eia861/). Historical temperature data were retrieved

using NOAA's weather API (https://www.weather.gov/documentation/services-web-apihttps://www.weather.gov/documentation/services-web-api). The source data underlying all the figures in the main article and supplementary information are provided as a Source Data file and deposited in a public repository[42]. Source data are provided with this paper.

## Code availability

The code used in this work is freely available at https://github.com/ChenFLD/Distribution-grid-blackout-riskhttps://github.com/ChenFLD/Distribution-grid-blackout-risk.

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

## Acknowledgements

This work was supported by the National Natural Science Foundation of China (No. 71934006 and 72140005, awarded to Y.Y.).

## Author contributions

Conceptualization, K.P.G., L.C., Y.W. and Y.Y.; Methodology, K.P.G., L.C., Y.W. and Y.Y.; Investigation, L.C. and K.P.G.; Writing—Original Draft, K.P.G., L.C., Y.W. and Y.Y.; Writing—Review & Editing, L.C., Y.W. and Y.Y.; Supervision, Y.W. and Y.Y.

## Competing interests

The authors declare no competing interests.
