## [Transparent Peer Review file · Nature Communications]

Vulnerability of Power Distribution Networks to Local Temperature Changes Induced by Global Climate Change

Corresponding Author: Professor Yang Weng

Version 0:

Reviewer comments:

Reviewer #1

(Remarks to the Author)

I'm delighted to review the paper in greater depth because the subject is interesting. Moreover, the submission is worth publication. Following are some minor comments:

This paper proposes a new, valuable systematic framework for evaluating the impact of global climate change (GCC) on electricity distribution (ED) services. Notably, the study overcomes challenges in estimating the safe operation boundary of distribution grid (DG) systems, providing insights into the macro and micro impacts of GCC. Moreover, the proposal to integrate more mesh structures for climate adaptation and the strategic matching of temperature-sensitive users with robust nodes offer innovative approaches to enhance the resilience of DG systems.

Some problems must be solved before it is considered for publication. If the following problems are well-addressed, this reviewer believes that the essential contribution of this paper is vital for electricity distribution research.

1. Relevant research background needs to be supplemented in INTRODUCTION. And it would help if you introduced the work of your predecessors properly.
2. On page 4, Climate change's impact on the size of reliable DG system part. There are a few suggestions, replacing 'minimal' with 'minimum' for more commonly accepted usage.
3. On page 9, discussion, authors are suggested to replace the color of the changing the color of the figure (e and i)
4. The conclusion section presents potential avenues for future research rather than a detailed summary of the current study's results.

In conclusion, this paper introduces a valuable systematic framework for evaluating the impact of global climate change (GCC) on electricity distribution (ED) services, demonstrating a certain level of novelty. While the study's overall content is engaging, structural improvements are warranted based on the feedback received during the review process.

(Remarks on code availability)

Reviewer #2

(Remarks to the Author)

The manuscript presents an interesting analysis of the climate change vulnerabilities of electricity distribution based on case studies of 47 states in the U.S. and 33 countries in Europe to analyze distribution grid reliability (macro impact) and a case study of California, U.S. to analyze blackout risk (micro impact). It uses a data-driven modeling approach to report macro and micro impacts of temperature rises on electricity distribution.

The study concludes that climate change poses a significant threat to electricity distribution systems, demanding immediate and strategic adaptations to mitigate risks and ensure reliable distribution of electricity.

This is based on a thought experiment analyzing the vulnerability of the current distribution grid, based on current electricity demand profiles, and current electricity demand-temperature relationship, to future increases in temperatures based on select representative concentration pathways (RCPs). However, the manuscript has significant areas of improvement in terms of the conceptual design, supporting key conclusions with evidence, improving clarity and readability, and relaxing strong assumptions. Details provided below:

1. There are significant issues with the conceptual design:

- The manuscript claims to innovate a “comprehensive and systematic framework to assess and analyze the profound effects of climate change on electricity distribution services.” However, both the conceptual design and related analysis the manuscript offers, insufficiently incorporates climate change impacts i.e., beyond temperature extremes, changes along the human dimensions (e.g., norms, behavior, and attitudes towards energy consumption), energy infrastructure (e.g., broad shifts in energy supply, smart grids, decentralized energy, etc.), and the inter-dependencies.
 - o The study takes an energy-focused approach but does little to conceptually and analytically incorporate the complex dynamics involving demographic, socio-cultural, economic, and political dimensions, all of which can significantly shape energy demands and thus will be of critical interest to scientists, planners, and users (the manuscript corroborates this assertion in L. 92). It remains unclear how these dimensions can be meaningfully incorporated and how the framework allows for incorporating associated uncertainties, both towards incorporating more realistic scenarios.
 - o Similarly, climate change impacts are studied only based on temperature, which severely limits insights on climate change-related vulnerabilities.
- The utility and implications of the control-experiments, considering insufficient details incorporated in the model (described above), remains unclear.
- Using control-experiments towards identifying unique effects of macro-scale drivers, e.g., climate change, is unclear and problematic. The effects of climate change will be shaped by the future prevailing conditions pertaining to the energy system and multiple human dimensions. Using the present conditions as controls is deeply unsound. Moreover, the manuscript offers limited insights and rationale for estimating unique climate change effects and frequently make contradictory statements (see for example, L.90-91 compared to L. 169 - 174).
- The manuscript compares the effects of climate change on distribution reliability in the US and several European countries. The rationale and importance of this comparison is unclear and remains unwarranted. Similarly, the manuscript analyzes blackout risk (micro impact) only in California, presented as a case study. Yet, it makes bold, general claims, some of which may not hold due to significant regional heterogeneities and others that are previously known or are intuitive, e.g., higher blackout risk in peak-load hours, higher frequency of blackout risk, and seasonal changes.

2. Conclusions remain unsupported or limited by the evidence provided:

- The key finding of the impact of climate change on the size of a reliable distribution grid system is unfounded for at least three reasons. First, the analysis selectively considers the most extreme (most pessimistic) climate change scenario to arrive at the sizeable and spatially differentiated impact conclusion (L. 140-143), which causes a circular reasoning. Second, owing to a ceteris paribus analysis, it assumes no change in other important aspects of the energy infrastructure and use, which can act as a conduit to shaping climate change impacts (L. 93). Here, identifying the unique effect of climate change is unwarranted. Third, the effect of climate change is only analyzed with a focus on demand response profile, whereas (distribution) grid reliability depends on several other variables, particularly extreme events, that cannot be accounted for while focusing only on temperature.
- The key findings on the future impact of climate change on electricity distribution reliability again assumes a constant demand response profile and no future changes in the grid, both of which are problematic. For example, the utilities are currently undertaking several programs like demand-response, Volt/Var optimization, dynamic Distributed energy resources (DER - inverter) control for improving power factor, evaluating hosting capacity for introducing more DERs etc for enabling reliable electricity distribution now and for meeting future demands (through integrated distribution planning).
- Overall, key conclusions are based on strong, unreasonable assumptions:
 - o Constant user’s temperature sensitivity
 - o Constant distribution grid infrastructure
- The manuscript offers no primary evidence, in terms of an economic analysis, to assert the following claim: “Climate change redefines the economic competitive advantages from the electricity perspective.”

3. The manuscript is very dense, uses a lot of jargons, and contains several normative assertions, which altogether will limit broader scientific readership. Particularly concerning are the normative statements about climate change, its impacts on the grid infrastructure, and the challenging nature of modeling the associations, which remains prevalent throughout the manuscript. Additionally, the manuscript casually uses key terms (climate change, macro and micro impact, risk, vulnerability, hazard, size assessment), while introducing them later in-text, deterring readability.

4. The manuscript distances itself with the rich existing literature on the topic. Incorporating the existing literature, e.g., (Bartos & Chester, 2015; Cox et al., 2017; Dumas et al., 2019; Jacobson et al., 2017; Jaglom et al., 2014; Stephens et al., 2013; Ward, 2013), can help clarify the critical knowledge gap(s), minimize normative assertions, and better inform the conceptual design.

Overall, the manuscript will benefit from improving the conceptual design, strengthening the evidence base for, and more accurately capturing, the conclusions, explaining the concepts upfront, relaxing the stringent assumptions regarding constant demand profiles and grid infrastructure, and incorporating relevant existing literature.

Bartos, M. D., & Chester, M. V. (2015). Impacts of climate change on electric power supply in the Western United States. *Nature Climate Change*, 5(8), 748–752.

Cox, S. L., Hotchkiss, E. L., Bilello, D. E., Watson, A. C., & Holm, A. (2017). Bridging Climate Change Resilience and Mitigation in the Electricity Sector Through Renewable Energy and Energy Efficiency: Emerging Climate Change and Development Topics for Energy Sector Transformation (NREL/TP-6A20-67040). National Renewable Energy Lab. (NREL), Golden, CO (United States). <https://doi.org/10.2172/1411521>

Dumas, M., Kc, B., & Cunliff, C. I. (2019). Extreme Weather and Climate Vulnerabilities of the Electric Grid: A Summary of Environmental Sensitivity Quantification Methods (ORNL/TM-2019/1252). Oak Ridge National Lab. (ORNL), Oak Ridge, TN (United States). <https://doi.org/10.2172/1558514>

Jacobson, M. Z., Delucchi, M. A., Cameron, M. A., & Frew, B. A. (2017). The United States can keep the grid stable at low cost with 100% clean, renewable energy in all sectors despite inaccurate claims. *Proceedings of the National Academy of Sciences*, 114(26), E5021–E5023. <https://doi.org/10.1073/pnas.1708069114>

Jaglom, W. S., McFarland, J. R., Colley, M. F., Mack, C. B., Venkatesh, B., Miller, R. L., Haydel, J., Schultz, P. A., Perkins, B.,

Casola, J. H., Martinich, J. A., Cross, P., Kolian, M. J., & Kayin, S. (2014). Assessment of projected temperature impacts from climate change on the U.S. electric power sector using the Integrated Planning Model®. *Energy Policy*, 73, 524–539. <https://doi.org/10.1016/j.enpol.2014.04.032>

Stephens, J. C., Wilson, E. J., Peterson, T. R., & Meadowcroft, J. (2013). Getting Smart? Climate Change and the Electric Grid. *Challenges*, 4(2), Article 2. <https://doi.org/10.3390/challe4020201>

Ward, D. M. (2013). The effect of weather on grid systems and the reliability of electricity supply. *Climatic Change*, 121(1), 103–113. <https://doi.org/10.1007/s10584-013-0916-z>

(Remarks on code availability)

Reviewer #3

(Remarks to the Author)

Thank you very much for your submission.

I found the work to be extremely interesting and closely aligned with current research themes.

Regarding the abstract, I would suggest further emphasizing the real innovation of the presented work.

In the introduction, it is essential to focus on the gap identified in the literature, explaining the need to address this issue and outlining how your work deviates from existing research.

I suggest implementing the literature review.

I noticed that Figure 1, although well-executed, is content-dense and sometimes challenging to read. I believe it would be appropriate to provide a more detailed description of the steps presented in Figure 1 to enhance accessibility.

Figure 4 is not clearly visible, and the interpretation of the graphs is very difficult. In my opinion, reconsidering the distribution of the graphs may improve overall readability.

Furthermore, more explanations are needed regarding the model used.

An exploration of how the limitations presented in your study could be overcome might add value to the overall research.

(Remarks on code availability)

Version 1:

Reviewer comments:

Reviewer #1

(Remarks to the Author)

Thank you for taking my advice seriously. The article is complete and clearly explained, contains a lot of detail and methodological notes, and I think the logic and ideas are very clear and intuitive throughout the article. However, there are still some minor issues with the article that can be revised. Here are my revisions that I hope will help you get your article published:

1. The formatting of the literature cited in the article is not consistent, with some citations using the [16] format and others being superscripted in the top right-hand corner of the text. Please confirm the formatting problem.
2. On page 8, in the sentence “As demand rises, a growing GDP may encourage grid upgrades.”, “a growing GDP” can be changed to “GDP growth” for consistency.
3. Consistency in Tense: Ensure that the tense is consistent throughout the passage. For example, if you are discussing future projections, use future tense consistently.
4. On page 12, “The changes in colors show that 55% of the states need to upgrade DG to ensure reliable distribution services for users, whose demand-weather response relation remains the same.” “relation” can be changed to “relationship.”
5. Add “the” for nouns present in some sentences.

In conclusion, this paper has analyzed the impact of global climate change (GCC) on electricity distribution in the United States and Europe and the mitigating or exacerbating effects of various technologies, such as photovoltaic (PV) systems and electric vehicles (EVs), on the impact of GCC. The paper details the methodology used to assess system risk, including climate modeling, load consumption data, historical temperature data, and distribution network data.

(Remarks on code availability)

Reviewer #2

(Remarks to the Author)

The revised manuscript addresses several of the previously raised concerns. In its current form, it proposes a framework to analyze the vulnerability of distribution grids to climate change (temperature impacts) and applies it to various regions to derive vulnerability insights. However, the revised manuscript still demonstrates three key areas of improvement.

First, the framework introduced is more modeling-oriented than conceptually oriented, in that it accounts for limited drivers and interactions particularly considering the socioeconomic impacts on energy supply (including distribution network) and energy demand, as well as on various broad-based climate adaptation strategies with energy implications. I will strongly recommend that the manuscript clarifies the “systematic framework” as a “modeling and analysis” framework and reframes the study accordingly.

Second, the discussion section needs to discuss the key findings, in the context of the existing literature and the key contributions the study makes recognizing the inherent limitations and key areas of improvement. Results on assessing adaptation options can be moved to preceding sections.

Third, the methodological framework has inherent limitations, besides the three identified under the “Methods” section, most of which were pointed out in the previous review report. The manuscript needs to be explicit about these limitations and cautious when interpreting the findings. These and other key areas of improvement need to be highlighted in the discussion section.

(Remarks on code availability)

Response to the Reviewers

NCOMMS-23-63838

Vulnerability of Electricity Distribution Service for Climate Change

Kishan Prudhvi Guddanti,¹ Lin Chen,¹ Yang Weng,^{*} and Yang Yu^{*}

We thank the editor and reviewers for their thoughtful comments which helped us to improve the paper. Based on the comments, we have carefully revised the paper. For example, the reviewers' comments are in *italics teal* color text and our answers are in regular font text. Changes to the original paper are in regular *blue* color text. Please see our point-to-point response to each of the reviewers' comments below. The references are attached at the end of the response document.

Response to Reviewer 1

Comment R1.1: *I'm delighted to review the paper in greater depth because the subject is interesting. Moreover, the submission is worth publication.*

Response to R1.1: Thanks for the feedback that our paper has an interesting topic and the analysis is with good depth. We also appreciate the reviewer's comment that our paper worth publication.

The paper is based on several existing projects from Department of Energy, National Science Foundation, and diversified utility data. From the industrial feedback and sponsor's comments, we found great values for the topic. When writing the paper, we discovered more foundational ideas. We will use this revision to answer your comments.

Comment R1.2: *Following are some minor comments: This paper proposes a new, valuable systematic framework for evaluating the impact of global climate change (GCC) on electricity distribution (ED) services. Notably, the study overcomes challenges in estimating the safe operation boundary of distribution grid (DG) systems, providing insights into the macro and micro impacts of GCC. Moreover, the proposal to integrate more mesh structures for climate adaptation and the strategic matching of temperature-sensitive users with robust nodes offer innovative approaches to enhance the resilience of DG systems.*

Response to R1.2: Thanks for summarizing our paper and its contribution. The reviewer had a good point that a uniqueness of this paper comes from coupling of data from global climate change and rigorous derivations in mathematical function for power systems. The macro and micro analysis of impacts on GCC let readers have a contrast for the evaluation at different scales. We discussed with the utility managers the recommendation for more mesh structures for climate adaptation and the node adaptation for resilience. The recommendation is feasible based on the comment from the managers. This is because the buses in distribution grid are much closer when compared to the nodes in transmission grids. Therefore, adding a line for mesh structure is not that costly.

At the same time, we want to highlight that the recommendation is not limited to upgrading the grid with mesh structures. In this paper, we have recommendations to both the policy makers and the power engineers. For policy makers, we suggest them to look into the peak load with the trend when deciding the policy for DG. This is because we notice that the slope of risk increases is not monotonic. So, the policy can have a delay or adjustment if needed. We also recommend the policy maker to consider geographical differences. This is because different factors are impacting the risks on DG blackout in different areas based on our macro analysis. For engineers, we recommend to have weakly meshed upgrades and put sensitive users to strategic locations, and put deploying distributed energy technologies at vulnerable nodes with sensitive user. Such outcome shows the power of the conceptual concepts, the theoretically sound foundation in power, and the flexibility in the tool evaluation.

Comment R1.3: *Some problems must be solved before it is considered for publication. If the following problems are well-addressed, this reviewer believes that the essential contribution of this paper is vital for electricity distribution research.*

Response to R1.3: Thanks for the comments that the paper is vital for research on electricity distribution. We also agree with the reviewer that we need to address the comments well before publication. In the next few pages, we will address your comments one by one.

Comment R1.4: *Relevant research background needs to be supplemented in INTRODUCTION. And it would help if you introduced the work of your predecessors properly.*

Response to R1.4: Thank you for your suggestion regarding the need to provide more relevant research background and properly introduce the work of predecessors in the Introduction. We have revised the Introduction to address this concern by expanding the discussion of prior research and positioning our work within the broader context of existing studies.

In the revised version, we have included a more detailed overview of key studies that examine the vulnerabilities of distribution grids (DGs) to global climate change (GCC), particularly in the areas of grid reliability, extreme weather events, and shifts in electricity demand. We now properly reference the foundational work of researchers like [R4, R8], who have explored the impacts of climate change on electricity systems, and [R18, R21], who have analyzed the challenges associated with upgrading DG systems to cope with increasing demand and climate-related stresses.

Additionally, we have clarified how our work builds on these studies by introducing a novel framework that integrates climate, economic, and engineering perspectives to assess the impact of GCC on DGs. This framework extends prior research by offering a more comprehensive analysis of how regional DG vulnerabilities evolve under different climate scenarios and renewable energy penetration levels. By properly contextualizing our work in relation to these key studies, we ensure that our contribution is clear and well-positioned within the existing body of research.

The expanded literature review is shown in Figure R1, and we believe this revision enhances the depth and comprehensiveness of our study.

However, DG does not have the redundancy and protection that the transmission grid has^{7,8}. Such a fact makes DG
more vulnerable to temperature fluctuations and changes in demand profiles. For example, [9] pointed out that 90% of the
blackouts in the U.S. are caused by the breakdown of the DGs. Therefore, GCC will fundamentally alter consumer demand
patterns, exacerbating the vulnerability of the DG⁶. For instance, climate change is projected to increase the frequency of
cooling-degree days, intensifying peak demand hours¹⁰. Several studies predict a more frequent surge of power demand
during the peak-demand hours in summers^{10–12}. One simple solution is to upgrade the distribution grids. However, it is
costly to upgrade all the distribution systems worldwide, giving the scale of the distribution systems for business, industrial,
and residential users¹³. Moreover, the higher demand in peak hours and the increasing cooling-degree days will elevate the
probability of overloading, which is the primary source of electricity blackout risk¹⁴. The anticipated rise in temperatures
due to climate change will reduce the capacity of distribution grids, further elevating the risk of overloading¹⁵. Consequently,
climate change can exacerbate the blackout risk of the existing DGs.

To address this, a rigorous approach is needed to analyze how GCC amplifies the vulnerability of distribution grids,
mainly through temperature changes. Existing studies have primarily examined GCC threats to DGs in two ways. The first
focuses on the damage to infrastructure caused by extreme weather events. For instance, [16, 17] have highlighted that
current DG facilities, such as grid lines and transformers, are highly vulnerable to damage from extreme weather events.
Other forms of physical damage have also been considered in this context^{16–19}. The second approach addresses the issue of
insufficient capacity, which occurs when the total electricity demand exceeds substation capacity^{6,12,20}. To analyze changes
in load, [20–22] have proposed data-driven methods to estimate how electricity demand responds to local temperature
changes in the transmission grid.

While increased load is an indicator of grid vulnerability, existing studies lack analytical tools to evaluate this vulner-
ability across different parts of a grid^{6,23}. For instance, previous research has struggled to determine the safe operational
boundaries of distribution grids due to the highly non-linear variable coupling and non-convex nature of the optimization
problem¹⁵. Additionally, the relationship between GCC and blackout risk in distributed generation systems involves a com-
plex, long-chain climate-economic-engineering process²⁴, making it challenging to estimate real-world DG vulnerability
based solely on temperature changes. Consequently, it is difficult to conduct scenario studies for GCC with DG at different
nodes within a feeder or across different distribution grids over a large geographic area. This limitation prevents the proper
modeling of a dose-response function that captures the relationship between climate change and DG blackout risk, which
is essential for effective risk assessment and management^{21,25}.

Figure R1: The expanded literature review in Introduction.

Comment R1.5: *On page 4, Climate change’s impact on the size of reliable DG system part. There are a few suggestions, replacing ‘minimal’ with ‘minimum’ for more commonly accepted usage.*

Response to R1.5: Thank you for this suggestion. In the revised paper, we use “minimum” instead of “minimal” in all the places. Please see the changes below in Figure R2. The updated term is highlighted in blue for your review.

166 comparing GCC-induced stress across regions. To standardize this comparison, we introduce the *Minimum Reliable Stan-*
321 *adaptation strategies*. Therefore, we compared the **minimum** reliable standard indices for distributed generation across 47
347 cases, the gap between the highest and lowest **minimum** reliable standard DG index values grows from 0.14 *p.s.m.* in the
348 2020s to 0.42 *p.s.m.* by 2050. By mid-century, 13 states will see their **minimum** reliable standard DG index increase by
366 electricity demand to temperature changes, causing Minnesota’s **minimum** reliable standard DG index to increase by 30%
372 Carolina is faster than in Minnesota as local temperatures rise from current levels. As a result, by 2050, the **minimum**

Figure R2: The changes for minimum.

Comment R1.6: *On page 9, discussion, authors are suggested to replace the color of the changing the color of the figure (e and i)*

Response to R1.6: Thank you for your suggestion to change the color of figures (e) and (i) in our manuscript to improve clarity and visual distinction. We have changed the color from green to pink as recommended. For example, the following two figures show the pictures before and after.

Figure R3: Original Figure 4

Figure R4: Revised Figure 4

Comment R1.7: *The conclusion section presents potential avenues for future research rather than a detailed summary of the current study’s results.*

Response to R1.7: Thank you for your suggestion. We recognize the importance of providing a detailed summary of the current study’s results, as well as presenting future research directions. Both aspects are critical: the summary helps readers fully understand the significance and implications of our research findings. It also outlines potential future avenues.

- **Strengthening the Summary:** In the revised conclusion, we have enhanced the summary by emphasizing the key findings of our study. Specifically, we focus on how our assessment framework demonstrates the significant impact of climate change on the reliability of distribution grids (DGs). We summarize the main results of our spatial and temporal analysis, showing how blackout risks are polarized across regions and how climate change interacts with economic growth and energy transitions. These insights are crucial for guiding policy decisions and engineering strategies for enhancing DG resilience. The revised conclusion is shown in Figure R5.
- **Reinforcing Future Research Directions:** In addition to the summary, we retained and reinforced the discussion of future research directions. We further emphasized how our framework can be extended to assess climate change impacts in other global regions. Additionally, we highlighted how future studies can explore how global climate change (GCC)-induced challenges in electricity distribution may affect regional electricity prices and potentially redefine economic comparative advantages. These points align with the broader goals of encouraging further development and refinement of our framework in new contexts. The revised conclusion is shown in Figure R5.
- **Enhancing Limitations and Future Solutions:** We have also added a discussion on potential solutions to the limitations identified in our study, presented in the newly revised Limitation section, as shown in Figure R6. In this section, we explore how advanced grid management techniques, such as Volt/Var optimization and dynamic DER control, could address the simplifications in our model. We also suggest future studies incorporate supply reliability uncertainties and political dimensions through scenario-based approaches. By addressing these limitations, we provide actionable directions for future research, which enhances the overall contribution of our study.

We believe these revisions provide a balanced and robust conclusion, offering a clear

summary of the current study’s contributions while encouraging further research based on the identified limitations and new opportunities for improvement.

**Conclusion**

Our assessment framework demonstrates that climate change significantly impacts the reliability of distribution grids, neces-
sitating innovative approaches to enhance resilience. The framework relies on a rigorous foundation using power flow equa-
tions, coordinate transformations, and techniques like the Pareto front to define safe-operation boundaries. These boundaries
enable scenario analysis of temperature-blackout risk responses, providing insights into risk sensitivity to temperature-
induced factors such as pure effects, economic characteristics, and energy transition dynamics. For example, our spatial
and temporal analysis of DG risks reveals polarization in blackout risks across different regions, states, and countries in
the US and EU. This implies that policymakers should use similar tools to tailor policies according to local DG needs.
Additionally, our findings show that regional effects of climate change on distribution systems vary based on economic
growth and energy transition scenarios, unveiling a new interaction mechanism between climate change and GDP growth.
This suggests that future evaluations of blackout risks should consider both peak hours and growth trends simultaneously
and focus on the stresses from gradual temperature changes, not just extreme weather events.

Regarding engineering solutions for climate adaptation, our results indicate that merely expanding grid capacity may not
be the most cost-efficient approach. Instead, more strategic solutions are needed, such as matching temperature-sensitive
users with more robust nodes and placing photovoltaic systems and temperature-insensitive users in more vulnerable grid
locations. For areas undergoing energy transitions, flexible energy installations should be strategically planned to miti-
gate climate change impacts and potentially avoid costly grid upgrades. If a DG upgrade is necessary, adding a line for
weakly meshed robustness can be an economical solution, given the proximity of nodes in distribution systems compared
to transmission grids. These findings broaden the scope for future studies on infrastructure vulnerability to climate change.
For future work, a comprehensive assessment of the overall impact of climate change on the entire power system—from
generation to end-users, through transmission and distribution networks—is essential. Furthermore, future research could
extend this framework to examine the impacts in other global regions, and explore how GCC-induced challenges in reliable
electricity distribution may redefine regional electricity prices, indirectly influencing economic comparative advantages
across different regions.

Figure R5: The revised conclusion

**Limitation**

We show potential limitations of current study and potential solutions.

- • The first limitation comes from the simplified representation of the distribution grid, assuming balanced loading
and aggregating demands. To overcome this, we propose integrating advanced grid management techniques such as
Volt/Var optimization and dynamic Distributed Energy Resources (DER) inverter control in future research.
- • The second limitation comes from the assumption of reliable and sufficient electricity supply in our current frame-
work. This limited could be refined by incorporating supply-side uncertainties, such as variable renewable energy
generation or disruptions in supply.
- • The third limitation is the lack of political dimensions. Although political factors were not explicitly included in our
current framework, future work could explore scenario-based approaches or qualitative methods to account for the
impact of regulatory policies, geopolitical risks, and international agreements.

Figure R6: The revised limitation

Comment R1.8: *In conclusion, this paper introduces a valuable systematic framework for evaluating the impact of global climate change (GCC) on electricity distribution (ED) services, demonstrating a certain level of novelty. While the study’s overall content is engaging, structural improvements are warranted based on the feedback received during the review process.*

Response to R1.8: Thank you for your comment. In this revision, we have made several structural improvements in line with your suggestions to enhance the clarity and coherence of the manuscript.

- **Rewriting the Abstract:** We rewrote the abstract to better highlight the core innovations of our work. By emphasizing the novelty of our approach, we ensured that the key contributions of the paper are clear from the outset. This new version more clearly conveys the practical and theoretical advancements made by our framework, as shown in Figure R7.

**Abstract**
Global climate change (GCC) changes temperatures, reshaping blackout risks in power distribution grids by gradually po-
larizing electricity demand. Existing studies lack a quantitative and analytical framework to assess blackout risks and devise
effective adaptation strategies to mitigate GCC’s adverse impacts, due to the absence of model converting the temperature
change into variations in blackout risk. To address this gap, we propose a theoretical framework for assessing the vulnera-
bility of arbitrary distribution grids to temperature-induced changes from GCC and for comparing the GCC-driven stress on
reliable electricity service. We observed existing GCC threats and intensifying trend on the reliability of real-world distribu-
tion grids, potentially increasing blackout risks during peak-load hours by 4% to 6%, depending on GDP growth rates. Our
findings reveal that over 20% of U.S. will require a minimum increase of over 10% in distributed generation (DG) capacity
before 2050, with six states needing more than a 20% increase. This suggests that DG capacity in vulnerable regions may
need to double by the end of the century. In contrast, the impact of GCC on power distribution in Europe is significantly
moderate. Our results recommend that policymakers focus on managing peak loads and addressing nonlinear risk trends
across different geographic areas. Furthermore, our study provides new insights for cost-effective grid upgrades, strategic
placement of distributed energy resources, and matching temperature-sensitive users with robust buses to enhance resilience.

Figure R7: Revised abstract highlighting innovations.

- **Reorganizing the Introduction:** The introduction has been reorganized to incorporate a broader review of the relevant literature. We added a comparative analysis of past methods versus our proposed framework. This adjustment helps position our work in the context of existing studies and highlights how our approach advances the state of the art in analyzing the impact of global climate change (GCC) on electricity distribution grids (DG), as shown in Figure R8.
- **Adjusting the Structure of the Main Sections:**

To address these challenges, we present a framework that examines hazards, vulnerabilities, and responses through
 a systematic four-step process. This framework is built on our research, which analytically derives the relationship be-
 tween power allocations and associated risk regions^{12,23,24}. Consequently, our approach integrates power flow analysis into
 climate change assessments, providing performance guarantees. When applied to realistic data, our framework can quanti-
 tatively assess the impact of global climate change on various parts of a grid (micro-scale) and across different geographic
 areas (macro-scale). Its scalable design ensures computational efficiency. For instance, in the U.S. and Europe, our tool
 indicates the necessity of considering peak hours, temperature trends, and economic growth together when evaluating grid
 performance. The impacts of climate change are both immediate and severe. Our findings suggest that strategically plac-
 ing renewable energy sources can postpone the need for grid upgrades in response to climate change. Moreover, if a grid
 upgrade is required, adding a single line to a distributed generation network can reduce vulnerability by up to 40%. This so-
 lution is particularly effective due to the relatively short distances between DG nodes compared to the broader transmission
 grid.

Figure R8: Reorganized introduction with literature review and comparison.

INTRODUCTION	3
RESULTS	4
Framework for assessing the GCC's impacts on ED service	4
Macro impact: energy advantage reshuffle in the U.S. and Europe	5
Climate change's impact on the size of reliable DG system	5
Climate change redefines the economic competitive advantages from the electricity perspective	7
Micro impact: notably threat on ED's reliability	7
DISCUSSION	9

Introduction	3
A Framework for Accessing GCC Impact onto Distribution Grids	4
Theoretical Foundation	4
A Framework	6
Micro perspective: Magnificent Threat from GCC to Distribution Grids	7
GCC's Pure Effect	7
GCC's Combined Effect Intervened by Changes of Other Factors	9
Macro perspective: Geographically-Heterogeneous Stresses from GCC to Regional Distribution Services	11
Distinct GCC Outcomes Across Diversified Regions	11
U.S. Results	11
EU Results	12
Why is There a Divergence?	13
Discussion: Efficiently Adapt the GCC	13
Conclusion	15

Figure R9: Restructured sections.

- **Revised Titles and Definitions:** To ensure that the titles reflect the revised structure and focus, we revised the section titles. This ensures a consistent tone and helps guide the reader through the logical progression of the paper, as shown in Figure R10.
- **Clarifying Key Terms and Keywords:** We define show the meaning of TB3R, which is the core conceptual design. At the same time, we add reference that is the basis of our five steps. For the five steps, we outline each step in details when compared to the short description in the last version, which required the readers to analyze the

INTRODUCTION	3
RESULTS	4
Framework for assessing the GCC's impacts on ED service	4
Macro impact: energy advantage reshuffle in the U.S. and Europe	5
Climate change's impact on the size of reliable DG system	5
Climate change redefines the economic competitive advantages from the electricity perspective	7
Micro impact: notably threat on ED's reliability	7
DISCUSSION	9

Introduction	3
A Framework for Accessing GCC Impact onto Distribution Grids	4
Theoretical Foundation	4
A Framework	6
Micro perspective: Magnificent Threat from GCC to Distribution Grids	7
GCC's Pure Effect	7
GCC's Combined Effect Intervened by Changes of Other Factors	9
Macro perspective: Geographically-Heterogeneous Stresses from GCC to Regional Distribution Services	11
Distinct GCC Outcomes Across Diversified Regions	11
U.S. Results	11
EU Results	12
Why is There a Divergence?	13
Discussion: Efficiently Adapt the GCC	13
Conclusion	15

Figure R10: Revised section titles reflecting the adjusted structure.

figure more without referred discussion. Finally, the hazard, vulnerability, and response as three pillars, so the reader will understand the basis when researcher shall look into the risk analysis. These clarification are shown in Figure R11.

(d) The Systematic Framework

(a) The reorganized framework.

risk in distribution systems. The framework, inspired by [32] and illustrated in Figure 1(d), involves five steps and is built
 on three foundational pillars: hazard, vulnerability, and response. The hazard pillar focuses on understanding the type and
 magnitude of the threat induced by the GCC. By incorporating the scenario of GCC-induced hazard, we apply the TB3R
 to examine clarify the climate-economic-engineering process and assess the degree of the DG's vulnerability to the GCC's
 threat. The vulnerability analysis further provides valuable insights into potential engineering solutions for adaptation. Our
 framework is capable of addressing various impacts of GCC on DG reliability. Below, we outline the detailed steps for
 assessing the impact caused by GCC-driven temperature changes:

- • Modeling the Hazard: Model the magnitude of GCC-driven local temperature changes.
- • Scoping: Identify whether and what other progress is included when assessing the GCC's impact. When the inter-
- vention of all other progresses such as GDP growth is excluded, the assessment calibrates the size of the GCC's pure
- effect. Otherwise, the assessment verifies the GCC's effect intervened by the change of other factors.
- • Calculating the climate-blackout response: Apply the TB3R to various GCC scenarios to generate the climate-
- blackout response, which manifests the DG's vulnerability to the GCC.
- • Assessing and comparing the GCC's impact: Calibrate the increase of the blackout risk driven by the GCC, and
- compare the GCC-induced stress on reliable-electricity distribution over regions.
- • Designing the adaptation strategy: Analyze how GCC-driven temperature change raises the DG's blackout risk and
- design adaptation strategies

(b) The illustration of framework.

Figure R11: Clarified key terms and process for better understanding.

Response to Reviewer 2

Comment R2.1: *The manuscript presents an interesting analysis of the climate change vulnerabilities of electricity distribution based on case studies of 47 states in the U.S. and 33 countries in Europe to analyze distribution grid reliability (macro impact) and a case study of California, U.S. to analyze blackout risk (micro impact). It uses a data-driven modeling approach to report macro and micro impacts of temperature rises on electricity distribution.*

Response to R2.1: Thank you for summarizing our contributions, and we are pleased that the reviewer found our analysis of climate change vulnerabilities to be of interest. This paper builds upon several existing projects supported by the Department of Energy, the National Science Foundation, and diverse utility data. Feedback from industry partners and sponsors has highlighted the importance and relevance of this topic. Through the process of writing the paper and making this revision, we uncovered additional foundational ideas.

In response to your suggestions, we have revised nearly all components of the paper to address your concerns. We have also engaged in discussions with experts from both academia, such as those at the IEEE Power and Energy Society General Meeting at Seattle in July, and industry, including Schweitzer Engineering Laboratories and Resource Innovation in August. These discussions, along with further literature reviews and simulations, have helped us significantly refine several core ideas. Additionally, we enlisted a professional English editor to improve the quality of the writing. In the following pages, we provide detailed responses to each of your comments, supported by justifications.

Comment R2.2: *The study concludes that climate change poses a significant threat to electricity distribution systems, demanding immediate and strategic adaptation to mitigate risks and ensure reliable distribution of electricity. This is based on a thought experiment analyzing the vulnerability of the current distribution grid, based on current electricity demand profiles, and current electricity demand-temperature relationship, to future increases in temperatures based on select representative concentration pathways (RCPs).*

Response to R2.2: Thanks for the comment. We agree with the reviewer that more justification on the experiment shall be provided.

In the last version our focus was on pure effect, as “temperature is a key indicator of climate change” [R20]. However, there are other factors that shall be analyzed together. Therefore, we show how to include an economic growth factor and energy transition into the consideration in this revision. At the same time, we want to point out that we had demographic information included when analyzing the GCC pure effect. In this version, we also explicitly analyze it and discussed the outcome of using our model to capture engineering solutions to GCC. We have also revised the framework to account for all these relationships, as shown in Figure R12. Furthermore, we validate the framework with specific case studies in the following sections.

Figure R12: The revised framework can capture the complex interactions between climate change and other factors (i.e. economic growth and energy transition), providing some adaptation to addressing the vulnerabilities of electricity distribution grids to climate-related stresses.

Specifically, we defined two scopes for pure effects, and combined effects with the consideration of other factor change, such as GDP impacts, and energy transition. Even more important, we show how to use our model to analyze the interdependency among these factors. As our model can flexibly separate and combine pure effect with other scopes, we can look into the interdependence between demand and topology of infrastructures. The revisions include clarification of the demographic information in the pure effect and discussion in Figure R13 and analysis of the other two influencing factors of economic growth in Figure R14 and energy transition in Figure R15.

In the second path, the demographic, socio-cultural, and economic effects are used to estimate the demand-temperature
 relation. Such a calculation can use data-driven methods^{21,22}. Although our result proposed in this paper focuses more
 on the temperature-related factors as a general trend for GCC, the demand-temperature relation can still capture demo-
 graphic characteristics. For instance, although the local temperatures in Colorado and Kansas are similar in the US, the
 demand-temperature relations are different due to different demographic, socio-cultural, and economic characteristics.

Figure R13: The clarification of demographic information in the pure effect.

**GCC's Combined Effect Intervened by Changes of Other Factors**

We further investigated how other factors, such as GDP growth and energy transitions, could amplify or mitigate the impact
 of GCC on DG blackout risk. These factors can indirectly influence DG's TB3R by altering demand-temperature sensitivity,
 grid capacity, and SOB. For instance, [36] shows that GDP growth can affect electricity demand. As demand rises, a growing
 GDP may encourage grid upgrades. To model GDP growth scenarios, we use the Shared Socioeconomic Pathways (SSP)
 framework³⁷. For each GDP growth scenario, we consider three scenarios involving varying demand, SOB, and GCC:
 (1) a benchmark scenario without GCC, (2) the RCP 4.5 scenario with moderate GCC, and (3) the RCP 8.5 scenario with
 significant climate change. Using these scenarios, we apply the U-shaped curve to model load changes and infrastructure
 upgrades, resulting in specific SOB outcomes in the TB3R framework.

We can now benchmark our results against the pure effect presented in the previous subsection for deeper insights.
 Specifically, Figure 3(a) and (b) indicate that the pure effect of temperature increase leads to a rise in risk before 2030.
 However, in the same figures from this section, the risk increases occur after 2030 when multiple factors are considered.
 This suggests that the grid upgrades driven by GDP growth effectively mitigate the risk increase caused by temperature
 changes before 2030. However, these GDP-driven grid upgrades are insufficient to counteract the risk increases caused
 by temperature changes after 2030, as the GCC's influence on DG blackout risk remains relatively moderate before this
 point. This observation is made possible by our two-step approach: analyzing the pure effect in the previous subsection and
 incorporating additional factors in this section.

While GDP is the dominant factor driving electricity demand growth by 2030, the DG expansion responding to the
 GDP-driven demand change is sufficient to keep the DG blackout risk relatively stable, even though the underlying effects
 of GCC have been present for decades. However, our analysis shows that the tipping point—where the GCC impact on
 blackout risk starts to escalate—occurs around 2020–2030, as indicated by the pure effect shown in Figure 2(c). After 2030,
 the rate of increase in blackout risk during peak-load hours and the number of vulnerable hours accelerates more rapidly in
 GCC scenarios compared to the benchmark scenario. Consequently, DG expansion to accommodate GDP-driven demand
 growth will no longer counterbalance the effects of GCC. This divergence results in distinctly different risk trajectories for
 DG blackout across the three GCC scenarios for the remainder of the century.

We found that the GCC's impact on the DG's blackout risk varies significantly across different GDP scenarios. In
 the SSP 1 scenario, the summer-average blackout risk of Figure 3(a) by 2100 is projected to more than double compared
 to 2000, increasing from 3.36% to 7.16% in the RCP 8.5 scenario. In the SSP 2 scenario, which assumes slower GDP
 growth, the same RCP 8.5 scenario results in a near doubling of blackout risk from 3.36% to 6.86% by 2100. The GCC's
 impact caused the blackout risk in the SSP 1 scenario grows additional 0.3% than in the SSP 2 scenario, which is nearly 9%
 higher. Therefore, the faster growth of GDP will amplify the GCC's impact larger. The additional 0.3% risk increase in the
 SSP 1 scenario is attributed to both a higher hourly blackout risk during peak-load hours and an increase in the number of
 vulnerable hours. The GDP growth will more largely amplify the GCC's impact in peak-load hours. Compared to SSP 2,
 the GCC impact in SSP 1 will lead the blackout risk growth 13.6% higher by the end of this century.

Figure R14: The GDP growth scenario.

Additionally, our analysis is based on solid foundations. For example, the flow chart is based on the hazard-vulnerability-response according to the the following papers.

[R2.2.1] Simpson, Nicholas P., Katharine J. Mach, Andrew Constable, Jeremy Hess, Ryan Hogarth, Mark Howden, Judy Lawrence et al. "A framework for complex climate change risk assessment." *One Earth* 4, no. 4 (2021): 489-501.

[R2.2.2] United States Environmental Protection Agency. *Conducting a Human Health Risk Assessment*. 2024. Available at: <https://www.epa.gov/risk/conducting-human-health-risk-assessment>.

Beyond GDP growth, we also examined the GCC impact considering the energy transition, which involves the adoption
 of various distribution-side technologies that fundamentally reshape DG. These technologies have varying effects on DG
 reliability. For example, the Paris Agreement has catalyzed the development of Photovoltaics (PV) and Electric Vehicles
 (EVs)³⁸. Increased PV penetration enhances DG reliability by mitigating the power distracting from the nodes, which helps
 mitigate the risk of blackouts. However, the growth of EVs introduces higher loads and thus can amplify the GCC's impact.
 Therefore, the pace and mix of different technologies will critically determine how energy transitions affect DG blackout
 risk under GCC.

We assess how different technologies affect the impact of climate change on the blackout risk of California's primary
 distribution Feeder 1. As shown in Figure 3(e), a 20% penetration of photovoltaic (PV) systems can reduce the blackout risk
 of distributed generation by 0.72% in 2050. This reduction is sufficient to fully offset the climate change impact under the
 RCP 4.5 scenario. Even in the more severe RCP 8.5 scenario, a 20% PV penetration still significantly mitigates the increased
 blackout risk caused by climate change. In contrast, Figure 3(f) indicates that a 20% penetration of electric vehicles (EVs)
 in the same area will increase the DG blackout risk by approximately 1.19% in 2050. This increase corresponds to a higher
 load on the distribution network nodes as EV penetration rises. Moreover, this effect is compounded by typical EV charging
 behaviors, as shown by a statistical analysis of EV usage patterns³⁹.

In the real world, the GCC, GDP growth, and energy transition will simultaneously impact the DG's blackout risk. We
 examined the impact of the energy transition on GCC's effects in the SSP 1 scenario. We generated three energy transition
 scenarios for the next two decades: the PV-only scenario, the EV-only scenario, and the scenario where the penetrations
 of PV and EV simultaneously grow. The penetration growth rates are set according to the Los Angeles 100% Renewable
 Energy(LA100) study conducted by the National Renewable Energy Laboratory (NREL)⁴⁰. Our results in Figure 3(c)
 suggest that the PV penetration rate, as projected in the LA100 study, is roughly sufficient to offset GCC's impact if EV
 penetration is absent. However, if EV penetration grows according to the LA100 projections, Figure 3(d) shows that the
 mitigating effect of PV will be offset. Consequently, the GCC's effect will still drive the DG blackout risk to substantially
 grow.

Figure R15: Energy transition.

[R2.2.3] National Research Council, Division on Earth, Life Studies, et al. Applications of toxicogenomic technologies to predictive toxicology and risk assessment[J]. 2007. Available at: <https://www.ncbi.nlm.nih.gov/books/NBK10201/>.

[R2.2.4] Federal Emergency Management Agency (FEMA). Risk Assessment Overview. 2024. Available at: <https://www.fema.gov/oet-tools/chemical-incident-consequence-management/4/2>.

Based on the foundational concepts, we developed three steps to capture the hazard, using power flow equation to find accurate blackout risks according to various factors with TB3R model. To better show our rigorous ideas, we add the theorem that analytically correlate the temperature-induced factors and the risks of blackout. We add Figure 1(a) and Figure 1(b) to explain how such a theorem leads to the risks in this paper. We add the TB3R model in Figure 1(c) and a paragraph on page 4 to show how to include other factors. We also redraw the flow chart, so that the hazard-vulnerability-response relationship and the three steps for engineering insights and solutions can be easily identified. These changes can

be seen in the following PrintScreens in Figure R16.

Figure R16: The revised Figure 1 with added theorem illustration and TB3R.

Finally, our focus is similar to other papers that look into the temperature impacts. The main contribution is not an inclusive consideration for all possible factors, but the towards using our theoretical work to analyze the climate-blackout risk relationship for engineering insights and solutions. For example, we refined the position of the paper, e.g., a study that uncovers specific vulnerabilities in the grid, providing actionable insights for future planning, resilience building, and adaptation strategies under climate change. For making this clear, we modified the abstract and introduction. The PrintScreens are in Figure R17.

Abstract

Global climate change (GCC) changes temperatures, reshaping blackout risks in power distribution grids by gradually po-
 larizing electricity demand. Existing studies lack a quantitative and analytical framework to assess blackout risks and devise
 effective adaptation strategies to mitigate GCC's adverse impacts, due to the absence of model converting the temperature
 change into variations in blackout risk. To address this gap, we propose a theoretical framework for assessing the vulnera-
 bility of arbitrary distribution grids to temperature-induced changes from GCC and for comparing the GCC-driven stress on
 reliable electricity service. We observed existing GCC threats and intensifying trend on the reliability of real-world distribu-
 tion grids, potentially increasing blackout risks during peak-load hours by 4% to 6%, depending on GDP growth rates. Our
 findings reveal that over 20% of U.S. will require a minimum increase of over 10% in distributed generation (DG) capacity
 before 2050, with six states needing more than a 20% increase. This suggests that DG capacity in vulnerable regions may
 need to double by the end of the century. In contrast, the impact of GCC on power distribution in Europe is significantly
 moderate. Our results recommend that policymakers focus on managing peak loads and addressing nonlinear risk trends
 across different geographic areas. Furthermore, our study provides new insights for cost-effective grid upgrades, strategic
 placement of distributed energy resources, and matching temperature-sensitive users with robust buses to enhance resilience.

(a) Abstract

To address these challenges, we present a framework that examines hazards, vulnerabilities, and responses through
 a systematic four-step process. This framework is built on our research, which analytically derives the relationship be-
 tween power allocations and associated risk regions^{12,23,24}. Consequently, our approach integrates power flow analysis into
 climate change assessments, providing performance guarantees. When applied to realistic data, our framework can quanti-
 tatively assess the impact of global climate change on various parts of a grid (micro-scale) and across different geographic
 areas (macro-scale). Its scalable design ensures computational efficiency. For instance, in the U.S. and Europe, our tool
 indicates the necessity of considering peak hours, temperature trends, and economic growth together when evaluating grid
 performance. The impacts of climate change are both immediate and severe. Our findings suggest that strategically plac-
 ing renewable energy sources can postpone the need for grid upgrades in response to climate change. Moreover, if a grid
 upgrade is required, adding a single line to a distributed generation network can reduce vulnerability by up to 40%. This so-
 lution is particularly effective due to the relatively short distances between DG nodes compared to the broader transmission
 grid.

(b) Introduction

Figure R17: In abstract and introduction section, we refined the position of the paper.

Comment R2.3: *However, the manuscript has significant areas of improvement in terms of the conceptual design, supporting key conclusions with evidence, improving clarity and readability, and relaxing strong assumptions.*

Response to R2.3: Thanks for pointing out the directions to improve. In this revision, we strictly follow the reviewer’s comments to improve the conceptual design, supporting key conclusions with evidence, improving clarity and readability, and relaxing strong assumptions.

1. Conceptual Design:

The conceptual design in the last submission was based on the pure effect. For example, we build a chain from climate change to local temperature changes, causing the demand change. Such demand variation cause current changes and redistribution, leading to the outage of distribution grid. Such a climate-economic-physical mechanism is not only based on Kirchoff’s law for theoretical foundation, but also consider probabilistic uncertainties in a complex model. We build such idea into the hazard-vulnerability-response framework, leading to three steps for insights and solutions.

However, the reviewer is right that we shall make clear the novelty of our conceptual design. Therefore, we added a theoretical section to explain the complex dynamics in detail, especially emphasizing that the core is the blackout-risk relation, and pointed out that based on the theoretical analysis, the concept of dose-response relationship of climate change on DG was proposed. The theoretical analysis in this section clearly pointed out that climate change directly affects the vulnerability risk of DG by changing the sensitivity of demand and grid security operating boundary (SOB). Other factors such as demography, social customs, and economic conditions indirectly affect these two direct factors and ultimately affect the vulnerability of DG. Therefore, we proposed the importance of introducing the concept of scope when evaluating the impact of climate change on DG to help understand these complex relationships more systematically. In Theoretical Foundation, we revised the conceptual design as Figure R18:

2. Supporting Key Conclusions with Evidences:

For supporting evidence and conclusions, we added three aspects of analysis in the micro analysis to support the main conclusions.

- First, we add more citations to this revision to support our conclusion and evidences. The followings in Figure R19 and R20 are some examples.

A distribution grid has a maximum capacity for meeting nodal demands, constrained by the power flow in grid lines as
 dictated by Kirchhoff's laws. An outage occurs when the power flow through any line exceeds its capacity. Consequently,
 we define the safe-operation region as the set of all nodal load profiles that prevent such overflows. The boundary of this
 region is known as the SOB. The SOB is determined by the physical characteristic of the DG, including the capacities of grid

(a) Core conceptual design focusing on grid capacity under normal conditions and its relationship to climate change.

The theorem formalizes the process of determining the SOB, which represents the maximum load the distribution grid
 can handle without collapsing. The SOB is a physical characteristic of the grid, determined by its infrastructure, such as
 line capacities and transformer limits, and independent of specific load profiles. By analyzing the power flow equations at
 each node, the boundary is derived as the point beyond which the grid can no longer sustain additional load.

(b) Theoretical basis highlighting how the SOB changes with demand and climate impacts.

$$\text{Risk}_h = \sum_{c=1}^t p(T_c) \cdot p(\Phi[\mathbf{P}|T_c, h] \leq 0)$$

Here, T_c represents climate change scenario c , $p(T_c)$ denotes the probability of climate change scenario c , and $\Phi[\mathbf{P}|T_c, h]$
 determines the margin of the demand profile \mathbf{P} under temperature T_c at hour h to the SOB. If $\Phi \leq 0$, the grid remains within
 safe operating conditions. Otherwise, a blackout will occur. We refer to this method as the climate-blackout response model,

(c) Introducing the dose-response relationship between climate change and grid vulnerability.

Figure R18: Conceptual design revisions in Theoretical Foundation.

24 The global climate change (GCC) polarizes the electricity supply^{1,2} and demand, which challenges the reliable electricity

1. Bartos, M. D. and Chester, M. V. (2015). Impacts of climate change on electric power supply in the Western United States. *Nature Climate Change* 5, 748–752. <https://doi.org/10.1038/nclimate2648>.
2. Cox, S. L., Hotchkiss, E. L., Bilello, D. E., Watson, A. C., and Holm, A. (2017). *Bridging Climate Change Resilience and Mitigation in the Electricity Sector Through Renewable Energy and Energy Efficiency: Emerging Climate Change and Development Topics for Energy Sector Transformation*. Tech. rep. National Renewable Energy Lab. (NREL), Golden, CO (United States). <https://doi.org/10.2172/1411521>.

(a) Bartos and Chester (2015) study the impacts of climate change on electric power supply, particularly how climate change can stress water resources for power generation. This supports the statement that GCC polarizes the electricity supply.

is one of the most critical infrastructures supporting modern civilization, e.g., for residential customers. The reliability and
 cost of the electricity distribution service significantly influence regional economic advantage and life quality. And, every
 blackout in DG causes a severe economic loss³. When climate change exacerbates extreme cooling and heating weather, it

4. Jaglom, W. S., McFarland, J. R., Colley, M. F., et al. (2014). Assessment of projected temperature impacts from climate change on the U.S. electric power sector using the Integrated Planning Model. *Energy Policy* 73, 524–539. <https://doi.org/10.1016/j.enpol.2014.04.032>.

(b) Cox et al. (2017) explore how renewable energy and energy efficiency play a role in mitigating the effects of climate change on energy systems, supporting the notion that GCC influences electricity supply.

However, DG does not have the redundancy and protection that the transmission grid has^{7,8}. Such a fact makes DG

8. Ward, D. M. (2013). The effect of weather on grid systems and the reliability of electricity supply. *Climatic Change* 121, 103–113. <https://doi.org/10.1007/s10584-013-0916-z>.

(c) Jaglom et al. (2014) predict that electricity production costs could increase significantly by 2050 due to temperature changes caused by GCC, supporting the claim that electricity distribution service impacts regional economic advantage.

Figure R19: Reference images supporting the key claims in the text. Each subfigure presents evidence from the cited literature, clarifying how DG faces challenges in GCC (Part1).

during the peak-demand hours in summers^{10–12}. One simple solution is to upgrade the distribution grids. However, it is
costly to upgrade all the distribution systems worldwide, giving the scale of the distribution systems for business, industrial,
and residential users¹³. Moreover, the higher demand in peak hours and the increasing cooling-degree days will elevate the
13. Stephens, J. C., Wilson, E. J., Peterson, T. R., and Meadowcroft, J. (2013). Getting Smart? Climate Change and the
Electric Grid. *Challenges* 4, Article 2. <https://doi.org/10.3390/challe4020201>.

(a) Ward (2013) reviews weather events’ impacts on grid systems in Europe and North America, emphasizing the vulnerabilities of electricity grids, particularly transmission systems, which support the statement about DG lacking redundancy and protection.

Other forms of physical damage have also been considered in this context^{16–19}. The second approach addresses the issue of
19. Dumas, M., Kc, B., and Cunliff, C. I. (2019). *Extreme Weather and Climate Vulnerabilities of the Electric Grid: A Summary of Environmental Sensitivity Quantification Methods*. Tech. rep. Oak Ridge National Lab. (ORNL), Oak Ridge, TN (United States). <https://doi.org/10.2172/1558514>.

(b) Stephens et al. (2013) discuss the challenges and costs of upgrading global distribution systems to accommodate the impacts of GCC, supporting the argument that upgrading DG is costly due to its scale.

placing renewable energy sources can postpone the need for grid upgrades in response to climate change. This aligns
with the argument by [28], who demonstrated the feasibility of a 100% renewable energy system while maintaining grid
stability. However, our analysis goes further by considering how regional grid vulnerabilities evolve with varying levels of
renewable energy penetration, such as PV and EV technologies. Moreover, if a grid upgrade is required, adding a single
28. Jacobson, M. Z., Delucchi, M. A., Cameron, M. A., and Frew, B. A. (2017). The United States can keep the grid stable
at low cost with 100% clean, renewable energy in all sectors despite inaccurate claims. *Proceedings of the National
Academy of Sciences* 114, E5021–E5023. <https://doi.org/10.1073/pnas.1708069114>.

(c) Jacobson et al. (2017) argue for the viability of a 100% renewable energy system, and while their focus is on stability, this research supports the discussion of regional DG vulnerabilities evolving under different renewable energy penetration levels.

Figure R20: Reference images supporting the key claims in the text. Each subfigure presents evidence from the cited literature, clarifying how DG faces challenges in GCC (Part2).

- Second, we add new figures and updated old figures to support our conclusion and evidences, as shown in Figure R21 and R22.

(c) Temperature-blackout Risk Response Relation (TB3R) Relationship

(a) The added theorem illustration and TB3R.

(d) The Systematic Framework

(b) The revised framework.

Figure R21: We add new figures and updated old figures to support our conclusion and evidences(part1).

(e) The log-log plot for hourly risk in different scenarios and years

(a) Log-Log plot analysis of hourly risk in 2000, 2050 and 2100 under RCP4.5 and RCP8.5. From the figure, we can find the distribution of hourly blackout risk to approach the longer tail distribution

(a) The average risk in summer under SSP1

(c) The average risk in SSP2 with separate growth of PV and EV penetration rates

(e) The average risk of 2050 with different PV penetration rates

(b) The average risk in summer under SSP2

(d) The average risk in SSP2 with simultaneous growth of PV and EV penetration rates

(f) The average risk of 2050 with different EV penetration rates

(b) This is the simulation result of GCC alongside GDP growth, energy transition, and grid infrastructure, demonstrating that our framework can incorporate additional influencing factors and explore the interactions between GCC and these factors.

Figure R22: We add new figures and updated old figures to support our conclusion and evidences(part2).

- Finally, we enhance the analysis to more effectively link the conclusions with the evidence, aligning them with the observations in Figure R23. Specifically, we include detailed explanations on the general applicability of our findings from the California case, the nonlinear relationship between risk and temperature, the rationale behind placing PV at vulnerable locations, and the complex mechanism through which GDP growth influences risk under GCC, and so on.

We argue that our findings on peak-load hours are applicable beyond California. In California Primary 1 system, the
 GCC has a stronger impact in the peak-load hours rather than off-peak load hours because that the demand is more sensitive
 to the temperature in hotter scenarios. Literature has demonstrated that the relationship between DG demand and temper-
 ature in most areas in the world is represented by a U-shaped curve, where demand spikes during both hotter summer and
 colder winter hours, which are also peak-load periods^{18,32}. Therefore, we conclude that GCC will similarly increase DG
 blackout risks during peak-load hours in most other regions in the world.

(a) The added paragraphs that explain why our conclusion on the California case has applicability in general.

GCC induces local temperature change for Figure 1(a) on the blackout risk. One value of our analysis is the analyti-
 101 cal solution for the complex climate-economic-engineering process. For example, the oval region Figure 1(a) with green
 boundary will move to the blue oval in Figure 1(b). For example, the maximal power-flow capacity of a line is also in-
 fluenced by temperature. According to the Temperature Coefficient Formula²⁹, the conductivity of a metal decreases with
 temperature, which implies that the power flow in a transmission line decreases as the temperature increases. Consequently,
 temperature changes the blue boundary in Figure 1(a). In Figure 1(b), we can see a shrunk version in red towards the origin
 when compared to the blue SOB region in Figure 1(a). Therefore, GCC-induced temperature change alters both the size and

(b) The added paragraphs that explain why the relationship between risk and temperature is nonlinear.

We assess how different technologies affect the impact of climate change on the blackout risk of California’s primary
 distribution Feeder 1. As shown in Figure 3(e), a 20% penetration of photovoltaic (PV) systems can reduce the blackout risk
 of distributed generation by 0.72% in 2050. This reduction is sufficient to fully offset the climate change impact under the
 RCP 4.5 scenario. Even in the more severe RCP 8.5 scenario, a 20% PV penetration still significantly mitigates the increased
 blackout risk caused by climate change. In contrast, Figure 3(f) indicates that a 20% penetration of electric vehicles (EVs)

(c) The added paragraphs that explain why our solution to put PV at the vulnerable location is correct.

We discovered a complex mechanism by which GDP growth influences the impact of global climate change (GCC)
 by analyzing blackout risk differences between the SSP 1 and SSP 2 scenarios. Due to differing GDP growth trajectories,
 the growth rates of DG nodal demands diverge in these scenarios. As GCC causes local temperatures to rise, the joint
 distribution of DG nodal demands shifts in different directions under the two SSP scenarios. Furthermore, the relationship
 between demand and temperature is dependent on GDP levels, meaning that the same rate of temperature increase results
 in varied demand growth across SSP 1 and SSP 2. Additionally, GDP-driven changes in demand affect the extent of DG
 grid expansion. The interplay of these factors leads to different impacts of GCC under varying GDP scenarios.

(d) The added paragraphs that explain why the impact of GDP growth on the risk with GCC is a complex mechanism.

Figure R23: We add analysis to link the conclusion and evidence better with the observations.

3. Improving Clarity and Readability:

To improve the clarity and readability of the article, we did the following.

- *Reorganized the paper structure and title:* We not only reorganize the paper structure, but also make the titles logically connected and consistent in the design in Figure

R24. In this revision, we introduced several key improvements to enhance the overall clarity and depth of the paper. First, we added a “theoretical foundation”, providing a stronger basis for understanding the framework and the impacts of GCC on distribution grids. Second, in addition to discussing the “pure effect” of GCC, we expanded the analysis to include “other influencing factors”, offering a more comprehensive view of the complex interactions at play. Third, we strengthened the “conclusion”, improving the readability and coherence of the paper by clearly summarizing the key findings and their implications. These modifications collectively make the paper more robust and accessible to readers.

INTRODUCTION	3
RESULTS	4
Framework for assessing the GCC’s impacts on ED service	4
Macro impact: energy advantage reshuffle in the U.S. and Europe	5
Climate change’s impact on the size of reliable DG system	5
Climate change redefines the economic competitive advantages from the electricity perspective	7
Micro impact: notably threat on ED’s reliability	7
DISCUSSION	9

Introduction	3
A Framework for Accessing GCC Impact onto Distribution Grids	4
Theoretical Foundation	4
A Framework	6
Micro perspective: Magnificent Threat from GCC to Distribution Grids	7
GCC’s Pure Effect	7
GCC’s Combined Effect Intervened by Changes of Other Factors	9
Macro perspective: Geographically-Heterogeneous Stresses from GCC to Regional Distribution Services	11
Distinct GCC Outcomes Across Diversified Regions	11
U.S. Results	11
EU Results	12
Why is There a Divergence?	13
Discussion: Efficiently Adapt the GCC	13
Conclusion	15

(a) The revised titles of section and subsection.

(b) The revised title.

Figure R24: We make the titles logically connected and consistent in the design.

- *Removed or Simplified Complex Terms:* In the revised manuscript, we have removed certain technical terms that were not essential for conveying our main arguments. For example, terms like “Experiment-based critical control point analysis,” “economic competitive advantages,” and “dose function” have been eliminated. These terms were not

central to our analysis, and their removal has helped make the text more streamlined and accessible.

- Original Term: Electricity distribution (ED)

Revision: We decided not to use the abbreviation "ED" for electricity distribution to avoid overloading the manuscript with too many acronyms, which could complicate readability. Instead, we consistently use the full term "electricity distribution" to maintain clarity throughout the text.

- Original Term: Size assessment

Revision: This term was introduced in the text without a clear explanation, causing confusion. To simplify the narrative, we have removed the use of this term and instead described the concept in more straightforward terms, such as "determining the scale of reliable DG systems."

- Original Term: Experiment-based critical control point analysis

Revision: Initially, this term was used to describe Step 5 of our evaluation framework. However, it led to misunderstandings regarding the experimental setup. To reduce complexity, we have removed all references to "critical control points" and simplified the explanation of the assessment process.

- Original Term: Economic competitive advantages

Revision: We initially introduced this term to discuss economic implications of climate change for energy systems, but the supporting analysis was insufficient. We decided to remove this discussion entirely to reduce the complexity of the manuscript and focus more on our primary findings related to grid vulnerability and climate risk.

- Original Term: Difference-in-difference (DID)

Revision: This experimental method was briefly mentioned in the Discussion Section but was not fully utilized in the analysis. Therefore, we replaced the technical term with a more straightforward explanation of the experiment. The revised text now states: "For instance, we tested two types of network structures: a tree structure (Figure 5(d)) and a mesh structure (Figure 5(h)). For each structure, we analyzed vulnerable nodes under three scenarios: a benchmark scenario, a climate change scenario, and a climate change scenario with consumer reallocation."

- Original Term: Climate change-robust matching planning

Revision: This planning strategy was introduced as an engineering solution based

on our experimental results. To simplify the terminology and avoid unnecessary jargon, we rephrased it as follows: "We also suggest a vital strategy for DG adaptation, which involves strategically deploying distributed energy technologies at vulnerable nodes with sensitive users to enhance overall system resilience." This revision eliminates the term "climate change-robust matching planning" while retaining the essence of the proposed solution.

- *Addressed Normative Statements:* We carefully reviewed the manuscript for any normative language that might suggest prescriptive or subjective statements about climate change and grid infrastructure. In places where such statements existed, we have either reworded them for neutrality or removed them altogether. Our goal was to ensure that the manuscript reflects a balanced view, presenting conclusions supported by evidence, uncertainty ranges, and well-grounded reasoning. Where applicable, we revised statements to reflect potential variability in outcomes and the influence of multiple factors rather than implying inevitable or overly deterministic results.

- Original Statement: "Climate change redefines the economic competitive advantages from the electricity perspective."

Revision: We removed the entire discussion on economic competitive advantages, as the analysis did not provide sufficient data or evidence to support such broad conclusions. This deletion helps to streamline the manuscript and eliminate unsupported assertions.

- Original Statement: "the DG must be heavily invested in capacity expansion if the DG planner seeks a reliable electricity distribution service."

Revision: We reworded this statement to reflect a more evidence-based approach. The revised text now states: "If DG planners are unwilling to accept the high rate of asset redundancy associated with expanded grid capacity, consumers must face a heightened risk of blackouts during peak-load periods, such as extreme heat days requiring increased cooling." This revision includes specific references to observed results in the study, grounding the conclusion in data rather than making a prescriptive claim.

load hours, the DG must be heavily invested in capacity expansion if the DG planner seeks a reliable electricity distribution
 service. However, the expanded grid capacity is only valuable during peak-load hours. During off-peak hours, climate

311 **If DG planners are unwilling to accept the high rate of asset redundancy associated with expanded grid capacity, consumers**
 312 **must face a heightened risk of blackouts during peak-load periods, such as extreme heat days requiring increased cooling.**

Figure R25: The revised statement1.

- Original Statement: “climate change’ s impacts on ED services fundamentally change our society.”

Revision: We refined this statement to provide more specificity and context. The revised version reads: “Beyond GDP growth, we also examined the GCC impact considering the energy transition, which involves the adoption of various distribution-side technologies that fundamentally reshape DG.” This revision focuses on the tangible factors influencing DG, such as energy transition and technological changes, providing a more precise and evidence-based statement.

In conclusion, climate change’s impacts on ED services fundamentally change our society.
 Beyond GDP growth, we also examined the GCC impact considering the energy transition, which involves the adoption
 of various distribution-side technologies that fundamentally reshape DG. These technologies have varying effects on DG

Figure R26: The revised statement2.

- *Improved Introduction and Definition of Key Terms:* To enhance readability and ensure clarity, we have revised the manuscript to introduce key terms such as “climate change,” “macro and micro impact,” “risk,” “vulnerability,” “hazard,” and “size assessment” earlier and in a more structured way. These terms are crucial for the integrity of the manuscript and are standard within the field of climate risk assessment, derived from widely recognized literature (e.g., [R17]). We now provide clear explanations and definitions when first introducing these terms, ensuring that readers unfamiliar with the terminology can understand their meaning and relevance in context.

- Global climate change

Definition: Global Climate Change refers to the long-term, significant changes in the Earth’s climate systems, particularly changes in temperature, precipitation patterns, and extreme weather events.

^aGlobal Climate Change is the long-term, significant changes in the Earth’s climate systems, particularly changes in temperature, precipitation patterns, and extreme weather events.

Figure R27: Added a footnote to the first occurrence of Global climate change in the paper.

- Risk

Definition: Risk is the probability of an adverse outcome, such as a power blackout, occurring due to the interaction between climate change and the distribution grid system.

^bRisk is the probability of an adverse outcome, such as a power blackout, occurring due to the interaction between climate change and the distribution grid system.

Figure R28: Added a footnote to the first occurrence of Risk in the paper.

– Hazard

Definition: Hazard refers to the potential physical events or phenomena that could cause harm or disrupt the power distribution system.

is visualized using a centered flower-style plot that summarizes the hazard-vulnerability-response model¹². Hazard refers to the potential physical events or phenomena that could cause harm or disrupt the power distribution system. Vulnerability is the degree to which a

Figure R29: Added a footnote to the first occurrence of Hazard in the paper.

– Vulnerability

Definition: Vulnerability is the degree to which a power distribution system and its consumers are susceptible to the adverse impacts of climate change.

physical events or phenomena that could cause harm or disrupt the power distribution system. Vulnerability is the degree to which a power distribution system and its consumers are susceptible to the adverse impacts of climate change. Response refers to how the power

Figure R30: Added a footnote to the first occurrence of Vulnerability in the paper.

– Response

Definition: Response refers to how the power distribution grid reacts to climate-induced hazards.

power distribution system and its consumers are susceptible to the adverse impacts of climate change. Response refers to how the power distribution grid reacts to climate-induced hazards. Step 1: Modeling the hazard of GCC-driven local temperature changes. Step 2: Scope

Figure R31: Added a footnote to the first occurrence of Response in the paper.

– Macro and Micro perspectives

Definition: Micro-perspective analyzes the localized impact of climate change on specific DG, focusing on individual moments, nodes, and influencing factors. Macro-perspective analyzes the large-scale geographic impact of climate change across entire regions or countries.

^cMicro-perspective analyzes the localized impact of climate change on specific DG, focusing on individual moments, nodes, and influencing factors.

^dMacro-perspective analyzes the large-scale geographic impact of climate change across entire regions or countries.

Figure R32: Added a footnote to the first occurrence of Macro and Micro perspectives in the paper.

4. Relaxing Strong Assumptions:

Thank you for your insightful feedback on the strong assumptions in the previous version of the manuscript. We agree that relaxing these assumptions is crucial for improving the generality and applicability of our analysis. In this revision, we have made several changes to address these concerns. Below, we outline the key assumptions and how we have relaxed them:

- Assumption: Constant User Temperature Sensitivity
 - Previous Version: We originally assumed a constant user temperature sensitivity in the pure effect analysis.
 - Revision: In response to the feedback, we have removed the assumption of constant temperature sensitivity. Instead, we now consider the spatial and temporal variability in user temperature sensitivity across regions and temperature levels. We added scenario analyses that reflect changes in temperature sensitivity due to economic growth, using widely accepted projections. These revisions are discussed in the **GCC’s Combined Effect Intervened by Changes of Other Factors** section, and the expanded analysis is illustrated in Figure R33.

580 **Modeling the Impact of Economic Growth and Climate on Energy Demand and DG System Ex-** 581 **pansion**

In this model, the impact of economic growth on electricity demand is incorporated through the per-capita GDP adjusted
growth factor. As the economy grows, it typically drives higher energy demand due to increased industrial activity, com-
mercial growth, and residential usage. To capture this effect, we introduce GDP growth as a key variable in the calculation
of the annual demand mean. By combining GDP with heating degree days (HDD, $T < 12.5^{\circ}\text{C}$) and cooling degree days
(CDD, $T > 27.5^{\circ}\text{C}$), the model comprehensively accounts for both economic and climate influences on the energy system.
The coefficients used for each factor are based on the values provided by Ref. [32].

The annual demand mean for each year is calculated as follows:

$$\text{dmean}(y) = \text{gdp}_b \cdot \text{GDP}(y) + \text{hdd}_b \cdot \text{HDD}(y) + \text{cdd}_b \cdot \text{CDD}(y), \quad (8)$$

where the coefficients are $\text{gdp}_b = 0.3763$, reflecting the impact of economic growth (GDP) on energy demand. $\text{cdd}_b =$
0.0103 , representing the influence of HDD on energy consumption. $\text{hdd}_b = 0$, indicating no significant effect from CDD in
the reference model.

Next, based on the historical data (2015-2019), we derive the U-curve, which captures how consumers’ electricity
demand responds to temperature variations. The U-curve illustrates how demand changes with temperature, with higher
demand during extreme heat or cold, and lower demand at moderate temperatures.

The hourly demand for each future year y is calculated by adjusting the 2019 baseline demand with a scaling factor
based on the U-curve and the annual demand gap, as follows:

$$\text{demand}(y, i) = \text{multiple} \times (\text{dmean}(y) - \text{dmean}(2019)) + \text{U_curve}(\text{temp}), \quad (9)$$

where *multiple* is calculated as the ratio of the demand mean derived from the U-curve in 2019, $\text{dmean_U}(2019)$, to the
baseline demand mean for 2019, $\text{dmean}(2019)$. The DG system expansion for year y is proportional to the growth in annual
demand relative to 1950, ensuring the system scales with increasing electricity demand.

Figure R33: The modeling of how economic growth affects temperature sensitivity.

- Assumption: Constant Distribution Grid Infrastructure
 - Previous Version: We initially used the assumption of constant DG infrastructure in the pure effect analysis, which may have caused confusion about the generality of our framework.
 - Revision: We have removed the assumption of constant infrastructure in the revised manuscript and clarified that this assumption was only applied to isolate the impact of climate change in the pure effect analysis. We now provide a more detailed discussion of DG infrastructure changes in the context of population and economic growth, as well as energy transitions involving PV and EV penetration. These revisions are discussed in the grid expansion scenarios and energy transition section, with figures illustrating the changes (see Figure R34 and Figure R35).

grid capacity, and SOB. For instance, [36] shows that GDP growth can affect electricity demand. As demand rises, a growing
 GDP may encourage grid upgrades. To model GDP growth scenarios, we use the Shared Socioeconomic Pathways (SSP)
 framework³⁷. For each GDP growth scenario, we consider three scenarios involving varying demand, SOB, and GCC:
 (1) a benchmark scenario without GCC, (2) the RCP 4.5 scenario with moderate GCC, and (3) the RCP 8.5 scenario with
 significant climate change. Using these scenarios, we apply the U-shaped curve to model load changes and infrastructure
 upgrades, resulting in specific SOB outcomes in the TB3R framework.

We can now benchmark our results against the pure effect presented in the previous subsection for deeper insights.
 Specifically, Figure 3(a) and (b) indicate that the pure effect of temperature increase leads to a rise in risk before 2030.
 However, in the same figures from this section, the risk increases occur after 2030 when multiple factors are considered.
 This suggests that the grid upgrades driven by GDP growth effectively mitigate the risk increase caused by temperature
 changes before 2030. However, these GDP-driven grid upgrades are insufficient to counteract the risk increases caused
 by temperature changes after 2030, as the GCC's influence on DG blackout risk remains relatively moderate before this
 point. This observation is made possible by our two-step approach: analyzing the pure effect in the previous subsection and
 incorporating additional factors in this section.

Figure R34: The discussion about grid expansion in the revised manuscript.

In summary, we have taken significant steps to relax the assumptions mentioned and improve the generality of our analysis. So we change the limitation, as shown in Figure ???. We hope that these revisions address your concerns and enhance the robustness of the conclusions drawn from our study.

Beyond GDP growth, we also examined the GCC impact considering the energy transition, which involves the adoption
 of various distribution-side technologies that fundamentally reshape DG. These technologies have varying effects on DG
 reliability. For example, the Paris Agreement has catalyzed the development of Photovoltaics (PV) and Electric Vehicles
 (EVs)³⁸. Increased PV penetration enhances DG reliability by mitigating the power distracting from the nodes, which helps
 mitigate the risk of blackouts. However, the growth of EVs introduces higher loads and thus can amplify the GCC's impact.
 Therefore, the pace and mix of different technologies will critically determine how energy transitions affect DG blackout
 risk under GCC.

We assess how different technologies affect the impact of climate change on the blackout risk of California's primary
 distribution Feeder 1. As shown in Figure 3(e), a 20% penetration of photovoltaic (PV) systems can reduce the blackout risk
 of distributed generation by 0.72% in 2050. This reduction is sufficient to fully offset the climate change impact under the
 RCP 4.5 scenario. Even in the more severe RCP 8.5 scenario, a 20% PV penetration still significantly mitigates the increased
 blackout risk caused by climate change. In contrast, Figure 3(f) indicates that a 20% penetration of electric vehicles (EVs)
 in the same area will increase the DG blackout risk by approximately 1.19% in 2050. This increase corresponds to a higher
 load on the distribution network nodes as EV penetration rises. Moreover, this effect is compounded by typical EV charging
 behaviors, as shown by a statistical analysis of EV usage patterns³⁹.

In the real world, the GCC, GDP growth, and energy transition will simultaneously impact the DG's blackout risk. We
 examined the impact of the energy transition on GCC's effects in the SSP 1 scenario. We generated three energy transition
 scenarios for the next two decades: the PV-only scenario, the EV-only scenario, and the scenario where the penetrations
 of PV and EV simultaneously grow. The penetration growth rates are set according to the Los Angeles 100% Renewable
 Energy(LA100) study conducted by the National Renewable Energy Laboratory (NREL)⁴⁰. Our results in Figure 3(c)
 suggest that the PV penetration rate, as projected in the LA100 study, is roughly sufficient to offset GCC's impact if EV
 penetration is absent. However, if EV penetration grows according to the LA100 projections, Figure 3(d) shows that the
 mitigating effect of PV will be offset. Consequently, the GCC's effect will still drive the DG blackout risk to substantially
 grow.

Figure R35: The discussion about energy transition with growing PV/EV in the revised manuscript.

Figure R36: The revised limitation section, which addresses assumptions.

Comment R2.4: *There are significant issues with the conceptual design:*

- The manuscript claims to innovate a “comprehensive and systematic framework to assess and analyze the profound effects of climate change on electricity distribution services.” However, both the conceptual design and related analysis the manuscript offers insufficiently incorporates climate change impacts i.e., beyond temperature extremes, changes along the human dimensions (e.g., norms, behavior, and attitudes toward energy consumption), energy infrastructure (e.g., broad shifts in energy supply, smart grids, decentralized energy, etc.) and the inter dependencies.

Response to R2.4: Thanks for the comment and the reviewer has a good suggestion. The comprehensiveness was referring to the three elements of risk and five steps. For example, three elements of risk are from reference [R17] saying “an interaction or aggregation of the determinants of risk—hazard, exposure, and vulnerability”. And, five steps are from [R1] saying “NAS describes the risk assessment paradigm as a process consisting of four major components: hazard identification, dose-response assessment, exposure assessment, and risk characterization”. These framework are the foundation for risk assessment [R15], while providing the flexibility and systematic logics. It is due to the flexibility, we add other factors to the pure design in the last version. For example, we add more factors like economic factors and energy transition to the framework. We also analyze the inter-dependency among various factors. At the same time, we also changed our title, abstract, introduction to reflect that our focus in this paper is temperature induced risk evaluation. The goal is to make the paper clearly tied to the temperature-induced factors. This is similar to other works focusing on temperature impact to GCC [R3]. Due to such a focus, our core innovation on conceptual design is on TB3R, where we integrate the SOB and U with RCP 4.5 and 8.5. Such a focus is why we do not want to consider all possible factors exclusively, which is hard and lose the focus. To clarify our innovation on the conceptual design and to explain what comprehensiveness and systematic means, we did the following to the revision.

- **Comprehensiveness:** As shown in Figure R37, we expanded the framework by providing more detailed explanations. This includes a deeper exploration of the interactions between the key risk determinants—hazard, exposure, and vulnerability, as referenced in [R17], as well as the incorporation of economic factors and energy transition. These additions enhance the comprehensiveness of the framework, allowing it to account for a broader range of influences and interactions, thereby improving its applicability in

assessing climate-induced risks.

risk in distribution systems. The framework, inspired by [32] and illustrated in Figure 1(d), involves five steps and is built
 on three foundational pillars: hazard, vulnerability, and response. The hazard pillar focuses on understanding the type and
 magnitude of the threat induced by the GCC. By incorporating the scenario of GCC-induced hazard, we apply the TB3R
 to examine clarify the climate-economic-engineering process and assess the degree of the DG's vulnerability to the GCC's
 threat. The vulnerability analysis further provides valuable insights into potential engineering solutions for adaptation. Our
 framework is capable of addressing various impacts of GCC on DG reliability. Below, we outline the detailed steps for
 assessing the impact caused by GCC-driven temperature changes:

- • **Modeling the Hazard:** Model the magnitude of GCC-driven local temperature changes.
- • **Scoping:** Identify whether and what other progress is included when assessing the GCC's impact. When the inter-
 vention of all other progresses such as GDP growth is excluded, the assessment calibrates the size of the GCC's pure
 effect. Otherwise, the assessment verifies the GCC's effect intervened by the change of other factors.
- • **Calculating the climate-blackout response:** Apply the TB3R to various GCC scenarios to generate the climate-
 blackout response, which manifests the DG's vulnerability to the GCC.
- • **Assessing and comparing the GCC's impact:** Calibrate the increase of the blackout risk driven by the GCC, and
 compare the GCC-induced stress on reliable-electricity distribution over regions.
- • **Designing the adaptation strategy:** Analyze how GCC-driven temperature change raises the DG's blackout risk and
 design adaptation strategies

Figure R37: Our updated explanation of framework

- **Systematic:** For systematic, it means “acting according to a fixed plan methodically.” The fixed plan refers to the five steps in Figure 1(d). And, such a method is based on our theoretical work on using power flow equation and system topology to identify operational margin and risks. Therefore, we call our method to be systematic. Figure R38 shows the revised the framework.

(d) The Systematic Framework

Figure R38: The figure for framework

- For making the key conceptual design clear to the readers, we did the following:
 1. We define our key conceptual design as TB3R, so that readers can easily follow our critical innovation. At the same time, we explain what comprehensive is, what systematic is, and what the key conceptual design is in the revision.

Figure R39: The illustration for TB3R

2. We add explanations that our focus is on modeling the relationship between factors that mainly contribute to the temperature changes. This is why our analysis includes the pure effect and combined effects, where combined effects also consider demographic and energy transition. The inter dependencies among these factors are also analyzed. Therefore, we did the following to improve our scope and claims, as shown in Figure R40.

of GCC and project the DG’s future blackout risk under various climate scenarios. To fully understand GCC’s impact, we
 employ a two-step analysis that examines both the “pure effect” and the “combined effect” of GCC-driven local temperature
 changes. This approach is critical for the following reasons: When assessing the pure effect, we exclude the moderating
 influences of other natural and human activities on DG blackout risk. This allows us to determine whether and when GCC-
 driven temperature changes have begun to manifest in the studied area, as well as to understand the characteristics of the
 GCC’s impact. The pure effect, as detailed in the subsection “GCC’s Pure Effect,” provides a benchmark for investigating
 how GCC interacts with other factors, such as GDP growth, to influence DG reliability. This is explored further in the
 subsection “GCC’s Combined Effect Intervened by Changes of Other Factors.” By contrast, examining the combined effect
 of GCC reveals whether and when GCC begins to pose a threat to DG reliability across various future scenarios. Within
 each scope, the average effect of GCC, its effect during vulnerable hours, and the degree of exposure are assessed using
 indices such as annual and seasonal average risks, risks during peak-load hours, and the number of vulnerable hours. The

Figure R40: The illustration for scope

- We carefully analyzed the factors mentioned in the reviewer’s comments, including considerations beyond temperature extremes, such as changes along the human dimensions (e.g., norms, behavior, and attitudes toward energy consumption), energy infrastructure (e.g., shifts in energy supply, smart grids, decentralized energy, etc.), and the interdependencies between these factors. In response, we have made several significant additions to our framework:
 1. Human Dimensions in Figure R41: We incorporated economic growth and energy consumption behavior into our model. At the micro level, we modeled demand-temperature relationships based on historical data, capturing individual behaviors

In the second path, the demographic, socio-cultural, and economic effects are used to estimate the demand-temperature
 relation. Such a calculation can use data-driven methods^{21,22}. Although our result proposed in this paper focuses more
 on the temperature-related factors as a general trend for GCC, the demand-temperature relation can still capture demo-
 graphic characteristics. For instance, although the local temperatures in Colorado and Kansas are similar in the US, the
 demand-temperature relations are different due to different demographic, socio-cultural, and economic characteristics.

Figure R41: Discussion about Human Dimensions

and consumption patterns. At the macro level, we applied functions from existing literature to account for broader cultural, and economic factors that influence regional energy usage. This ensures our model comprehensively addresses how human behavior impacts grid vulnerability under varying temperature conditions.

Beyond GDP growth, we also examined the GCC impact considering the energy transition, which involves the adoption
 of various distribution-side technologies that fundamentally reshape DG. These technologies have varying effects on DG
 reliability. For example, the Paris Agreement has catalyzed the development of Photovoltaics (PV) and Electric Vehicles
 (EVs)³⁸. Increased PV penetration enhances DG reliability by mitigating the power distracting from the nodes, which helps
 mitigate the risk of blackouts. However, the growth of EVs introduces higher loads and thus can amplify the GCC's impact.
 Therefore, the pace and mix of different technologies will critically determine how energy transitions affect DG blackout
 risk under GCC.

We assess how different technologies affect the impact of climate change on the blackout risk of California's primary
 distribution Feeder 1. As shown in Figure 3(e), a 20% penetration of photovoltaic (PV) systems can reduce the blackout risk
 of distributed generation by 0.72% in 2050. This reduction is sufficient to fully offset the climate change impact under the
 RCP 4.5 scenario. Even in the more severe RCP 8.5 scenario, a 20% PV penetration still significantly mitigates the increased
 blackout risk caused by climate change. In contrast, Figure 3(f) indicates that a 20% penetration of electric vehicles (EVs)
 in the same area will increase the DG blackout risk by approximately 1.19% in 2050. This increase corresponds to a higher
 load on the distribution network nodes as EV penetration rises. Moreover, this effect is compounded by typical EV charging
 behaviors, as shown by a statistical analysis of EV usage patterns³⁹.

In the real world, the GCC, GDP growth, and energy transition will simultaneously impact the DG's blackout risk. We
 examined the impact of the energy transition on GCC's effects in the SSP 1 scenario. We generated three energy transition
 scenarios for the next two decades: the PV-only scenario, the EV-only scenario, and the scenario where the penetrations
 of PV and EV simultaneously grow. The penetration growth rates are set according to the Los Angeles 100% Renewable
 Energy(LA100) study conducted by the National Renewable Energy Laboratory (NREL)⁴⁰. Our results in Figure 3(c)
 suggest that the PV penetration rate, as projected in the LA100 study, is roughly sufficient to offset GCC's impact if EV
 penetration is absent. However, if EV penetration grows according to the LA100 projections, Figure 3(d) shows that the
 mitigating effect of PV will be offset. Consequently, the GCC's effect will still drive the DG blackout risk to substantially
 grow.

Figure R42: Discussion about Energy Infrastructure

2. Energy Infrastructure in Figure R42: We incorporated Energy Infrastructure into

our analysis by considering the impact of key distribution-side technologies, such as photovoltaics (PV) and electric vehicles (EV), on the distribution grid (DG). Specifically, we examined how increased penetration of these technologies, driven by the energy transition, affects DG reliability under different climate change scenarios. Our model evaluates the effects of varying PV and EV penetration rates on blackout risk, showing that while PV can mitigate blackout risks by reducing load on distribution nodes, EVs can increase risks due to higher loads. By integrating these elements, our framework captures how shifts in energy infrastructure shape the DG’s resilience to climate-induced stresses.

3. Interdependencies in Figure R43: In our revised framework, we explored the combined effects of GCC, GDP growth, and energy transitions on DG blackout risk. Under the SSP1 scenario, we analyzed PV-only, EV-only, and combined PV+EV transitions using projections from the LA100 study. Our results show that while PV growth can offset GCC’s impact, the addition of EV growth negates this effect, increasing blackout risk. GCC’s impact is also most pronounced during peak-load hours, with non-linear growth varying across GDP scenarios, presenting challenges for future grid adaptation.

After analyzing both the direct and combined effects of GCC, we observed that its impact on electricity distribution
is most pronounced during peak-load hours, with non-linear growth trajectories across various GDP scenarios. These two
characteristics present significant challenges for future adaptation strategies. The first challenge is the trade-off between
expanding Distributed Generation (DG) capacity and tolerating a higher blackout risk. Since climate change substantially
increases blackout risk during peak-load hours, ensuring a reliable electricity distribution service would require substantial
investment in expanding DG capacity. However, the additional capacity is primarily useful during these peak periods.
During off-peak hours, climate change does not significantly affect power flow, leaving the expanded capacity underutilized.
If DG planners are unwilling to accept the high rate of asset redundancy associated with expanded grid capacity, consumers
must face a heightened risk of blackouts during peak-load periods, such as extreme heat days requiring increased cooling.
The second challenge lies in determining the optimal timing for investing in DG adaptation to climate change. As climate
impacts intensify non-linearly, the reliability of a DG system can rapidly decline after reaching a critical tipping point. An
economically efficient strategy would involve substantial investment in climate adaptation measures around this tipping
point. This approach necessitates accurately assessing consumer thresholds to predict when each DG system is likely to
reach its tipping point, beyond which system reliability sharply declines. Understanding these thresholds is crucial for
planning timely and cost-effective interventions to enhance system resilience.

Figure R43: Discussion about Interdependencies

Comment R2.5: *The study takes an energy-focused approach but does little to conceptually and analytically incorporate the complex dynamics involving demographic, socio-cultural, economic, and political dimensions, all of which can significantly shape energy demands and thus will be of critical interest to scientists, planners, and users (the manuscript corroborates this assertion in L. 92).*

Response to R2.5: Thanks for your specific suggestions on incorporating various factors conceptually and analytically. In the last version, we focused more on pure effect. In this version, we show that one feature of our framework is its flexibility to include other factors. For your four suggestions on “demographic, socio-cultural, economic, and political dimensions”, we add the first three in our revision and discuss the last one for future work. Please see the detailed discussion below.

- **Demographic and Socio-Cultural Dimensions:** These are important factors. In the revision, both of them are embedded in the U-shaped temperature-demand curve used in our framework. This curve reflects regional differences in energy consumption patterns as influenced by population characteristics (e.g., age distribution, urbanization) and socio-cultural norms (e.g., attitudes toward energy usage and efficiency). By utilizing regional-specific U-shaped curves, we capture how these dimensions affect electricity demand. For example, regions with higher urbanization or differing cultural attitudes toward cooling and heating have distinct demand behaviors, which are represented within the U-curve model. We have the statement in the revised manuscript, as shown in Figure R44.

In the second path, the demographic, socio-cultural, and economic effects are used to estimate the demand-temperature
relation. Such a calculation can use data-driven methods^{21,22}. Although our result proposed in this paper focuses more
on the temperature-related factors as a general trend for GCC, the demand-temperature relation can still capture demo-
graphic characteristics. For instance, although the local temperatures in Colorado and Kansas are similar in the US, the
demand-temperature relations are different due to different demographic, socio-cultural, and economic characteristics.

Figure R44: Discussion about Human Dimensions

- **Economic Dimensions:** In the revision, our framework models the impact of economic growth on energy demand by incorporating per-capita GDP as a key factor in the annual demand mean calculation. The model combines GDP growth with heating degree days (HDD) and cooling degree days (CDD) to capture both economic and climate influences on energy systems. The framework also examines how economic growth interacts with climate factors to drive electricity demand over time. The discussion about the impact of GDP growth on the risk is in Figure R45(a), and the modeling in Figure R45(b).

**GCC's Combined Effect Intervened by Changes of Other Factors**

We further investigated how other factors, such as GDP growth and energy transitions, could amplify or mitigate the impact
of GCC on DG blackout risk. These factors can indirectly influence DG's TB3R by altering demand-temperature sensitivity,
grid capacity, and SOB. For instance, [36] shows that GDP growth can affect electricity demand. As demand rises, a growing
GDP may encourage grid upgrades. To model GDP growth scenarios, we use the Shared Socioeconomic Pathways (SSP)
framework³⁷. For each GDP growth scenario, we consider three scenarios involving varying demand, SOB, and GCC:
(1) a benchmark scenario without GCC, (2) the RCP 4.5 scenario with moderate GCC, and (3) the RCP 8.5 scenario with
significant climate change. Using these scenarios, we apply the U-shaped curve to model load changes and infrastructure
upgrades, resulting in specific SOB outcomes in the TB3R framework.

We can now benchmark our results against the pure effect presented in the previous subsection for deeper insights.
Specifically, Figure 3(a) and (b) indicate that the pure effect of temperature increase leads to a rise in risk before 2030.
However, in the same figures from this section, the risk increases occur after 2030 when multiple factors are considered.
This suggests that the grid upgrades driven by GDP growth effectively mitigate the risk increase caused by temperature
changes before 2030. However, these GDP-driven grid upgrades are insufficient to counteract the risk increases caused
by temperature changes after 2030, as the GCC's influence on DG blackout risk remains relatively moderate before this
point. This observation is made possible by our two-step approach: analyzing the pure effect in the previous subsection and
incorporating additional factors in this section.

While GDP is the dominant factor driving electricity demand growth by 2030, the DG expansion responding to the
GDP-driven demand change is sufficient to keep the DG blackout risk relatively stable, even though the underlying effects
of GCC have been present for decades. However, our analysis shows that the tipping point—where the GCC impact on
blackout risk starts to escalate—occurs around 2020–2030, as indicated by the pure effect shown in Figure 2(c). After 2030,
the rate of increase in blackout risk during peak-load hours and the number of vulnerable hours accelerates more rapidly in
GCC scenarios compared to the benchmark scenario. Consequently, DG expansion to accommodate GDP-driven demand
growth will no longer counterbalance the effects of GCC. This divergence results in distinctly different risk trajectories for
DG blackout across the three GCC scenarios for the remainder of the century.

We found that the GCC's impact on the DG's blackout risk varies significantly across different GDP scenarios. In
the SSP 1 scenario, the summer-average blackout risk of Figure 3(a) by 2100 is projected to more than double compared
to 2000, increasing from 3.36% to 7.16% in the RCP 8.5 scenario. In the SSP 2 scenario, which assumes slower GDP
growth, the same RCP 8.5 scenario results in a near doubling of blackout risk from 3.36% to 6.86% by 2100. The GCC's
impact caused the blackout risk in the SSP 1 scenario grows additional 0.3% than in the SSP 2 scenario, which is nearly 9%
higher. Therefore, the faster growth of GDP will amplify the GCC's impact larger. The additional 0.3% risk increase in the
SSP 1 scenario is attributed to both a higher hourly blackout risk during peak-load hours and an increase in the number of
vulnerable hours. The GDP growth will more largely amplify the GCC's impact in peak-load hours. Compared to SSP 2,
the GCC impact in SSP 1 will lead the blackout risk growth 13.6% higher by the end of this century.

(a) Discussion

**Modeling the Impact of Economic Growth and Climate on Energy Demand and DG System Ex-**
**ansion**

In this model, the impact of economic growth on electricity demand is incorporated through the per-capita GDP adjusted
growth factor. As the economy grows, it typically drives higher energy demand due to increased industrial activity, com-
mercial growth, and residential usage. To capture this effect, we introduce GDP growth as a key variable in the calculation
of the annual demand mean. By combining GDP with heating degree days (HDD, $T < 12.5^\circ\text{C}$) and cooling degree days
(CDD, $T > 27.5^\circ\text{C}$), the model comprehensively accounts for both economic and climate influences on the energy system.
The coefficients used for each factor are based on the values provided by Ref. [32].

The annual demand mean for each year is calculated as follows:

$$\text{dmean}(y) = \text{gdp}_0 \cdot \text{GDP}(y) + \text{hdd}_0 \cdot \text{HDD}(y) + \text{cdd}_0 \cdot \text{CDD}(y), \tag{8}$$

where the coefficients are $\text{gdp}_0 = 0.3763$, reflecting the impact of economic growth (GDP) on energy demand. $\text{cdd}_0 =$
0.0103 , representing the influence of HDD on energy consumption. $\text{hdd}_0 = 0$, indicating no significant effect from CDD in
the reference model.

Next, based on the historical data (2015-2019), we derive the U-curve, which captures how consumers' electricity
demand responds to temperature variations. The U-curve illustrates how demand changes with temperature, with higher
demand during extreme heat or cold, and lower demand at moderate temperatures.

The hourly demand for each future year y is calculated by adjusting the 2019 baseline demand with a scaling factor
based on the U-curve and the annual demand gap, as follows:

$$\text{demand}(y, i) = \text{multiple} \times (\text{dmean}(y) - \text{dmean}(2019)) + \text{U_curve}(\text{temp}_i), \tag{9}$$

where *multiple* is calculated as the ratio of the demand mean derived from the U-curve in 2019, $\text{dmean_U}(2019)$, to the
baseline demand mean for 2019, $\text{dmean}(2019)$. The DG system expansion for year y is proportional to the growth in annual
demand relative to 1950, ensuring the system scales with increasing electricity demand.

(b) Modeling

Figure R45: The revised manuscript about Economic Dimensions

- Political Dimensions: Thanks for mentioning this dimension. We did not explicitly incorporate political dimensions into our framework, primarily due to the high uncertainty and variability of political factors over the long time horizon considered in our analysis [R16]. Political influences, such as regulatory policies and international

agreements, can have significant short- and medium-term effects on energy systems. However, their impact can be highly contingent on regional governance, policy shifts, and geopolitical factors, which are difficult to forecast with the precision needed for our quantitative modeling framework [R11]. This is why similar work [R4, R8, R10] did not add such dimension. While we recognize the importance of political dimensions in shaping energy infrastructure and policies, incorporating them reliably would require an additional layer of complexity and speculative assumptions, which could detract from the robustness of our current analysis. Therefore, we add them to the limitation, as shown in Figure R46.

457 • The third limitation is the lack of political dimensions. Although political factors were not explicitly included in our
458 current framework, future work could explore scenario-based approaches or qualitative methods to account for the
459 impact of regulatory policies, geopolitical risks, and international agreements.

Figure R46: Limitation for political dimensions

For reflecting the changes according to the reviewer

- We have expanded the theoretical section to explain the complex dynamics, particularly emphasizing the core relationship between blackout risk and key influencing factors. Based on theoretical analysis, we introduced the concept of the dose-response relationship of climate change impacts on Distribution Grids (DG). Additionally, we strengthened the theoretical analysis to explain that the direct influencing factors are the demand-temperature sensitivity and the State of Balance (SOB) of the Grid. All other factors, such as demographics, social customs, and economic conditions, indirectly affect these two direct factors. Thus, based on the above theoretical analysis, we clarified that assessing the impact of climate change on DG requires considering the concept of “scope”. The revised Theoretical Foundation is shown in Figure R47.
- In micro analysis, we added three parts to address the issues raised by the reviewer regarding the needed relationships:
 1. We considered the impact of economic growth in our analysis, particularly how it affects energy demand and consumption patterns. By modeling the influence of economic growth on the reliability of distribution grids (DG), we assess how climate change exacerbates blackout risks under different economic scenarios. The revised manuscript is shown in Figure R48.

GCC induces local temperature change for Figure 1(a) on the blackout risk. One value of our analysis is the analytical solution for the complex climate-economic-engineering process. For example, the oval region Figure 1(a) with green
 boundary will move to the blue oval in Figure 1(b). For example, the maximal power-flow capacity of a line is also influenced by temperature. According to the Temperature Coefficient Formula²⁹, the conductivity of a metal decreases with
 temperature, which implies that the power flow in a transmission line decreases as the temperature increases. Consequently,
 temperature changes the blue boundary in Figure 1(a). In Figure 1(b), we can see a shrunk version in red towards the origin
 when compared to the blue SOB region in Figure 1(a). Therefore, GCC-induced temperature change alters both the size and
 shape of the SOB region and the oval region. Such two regions jointly decide the risk probability of a blackout. We call such
 a relationship as the temperature-blackout risk response relation (TB3R), shown in Figure 1(c). In such a flow chart, the
 energy transition based on GCC impacts the generation, loads, and grids. Such information is used to calculate the SOB.
 In the second path, the demographic, socio-cultural, and economic effects are used to estimate the demand-temperature
 relation. Such a calculation can use data-driven methods^{18,19}. Although our result proposed in this paper focuses more
 on the temperature-related factors as a general trend for GCC, the demand-temperature relation can still capture demographic
 characteristics. For instance, although the local temperatures in Colorado and Kansas are similar in the US, the
 demand-temperature relations are different due to different demographic, socio-cultural, and economic characteristics.

(a) The analysis about the other factors in Theoretical Foundation.

We further investigated how other factors, such as GDP growth and energy transitions, could amplify or mitigate the impact
 of GCC on DG blackout risk. These factors can indirectly influence DG's TB3R by altering demand-temperature sensitivity,
 grid capacity, and SOB. For instance, [36] shows that GDP growth can affect electricity demand. As demand rises, a growing

(b) In micro analysis, the other factors is indirect.

Figure R47: The revised manuscript about Economic Dimensions

**GCC's Combined Effect Intervened by Changes of Other Factors**

We further investigated how other factors, such as GDP growth and energy transitions, could amplify or mitigate the impact
 of GCC on DG blackout risk. These factors can indirectly influence DG's TB3R by altering demand-temperature sensitivity,
 grid capacity, and SOB. For instance, [36] shows that GDP growth can affect electricity demand. As demand rises, a growing
 GDP may encourage grid upgrades. To model GDP growth scenarios, we use the Shared Socioeconomic Pathways (SSP)
 framework³⁷. For each GDP growth scenario, we consider three scenarios involving varying demand, SOB, and GCC:
 (1) a benchmark scenario without GCC, (2) the RCP 4.5 scenario with moderate GCC, and (3) the RCP 8.5 scenario with
 significant climate change. Using these scenarios, we apply the U-shaped curve to model load changes and infrastructure
 upgrades, resulting in specific SOB outcomes in the TB3R framework.

We can now benchmark our results against the pure effect presented in the previous subsection for deeper insights.
 Specifically, Figure 3(a) and (b) indicate that the pure effect of temperature increase leads to a rise in risk before 2030.
 However, in the same figures from this section, the risk increases occur after 2030 when multiple factors are considered.
 This suggests that the grid upgrades driven by GDP growth effectively mitigate the risk increase caused by temperature
 changes before 2030. However, these GDP-driven grid upgrades are insufficient to counteract the risk increases caused
 by temperature changes after 2030, as the GCC's influence on DG blackout risk remains relatively moderate before this
 point. This observation is made possible by our two-step approach: analyzing the pure effect in the previous subsection and
 incorporating additional factors in this section.

While GDP is the dominant factor driving electricity demand growth by 2030, the DG expansion responding to the
 GDP-driven demand change is sufficient to keep the DG blackout risk relatively stable, even though the underlying effects
 of GCC have been present for decades. However, our analysis shows that the tipping point—where the GCC impact on
 blackout risk starts to escalate—occurs around 2020–2030, as indicated by the pure effect shown in Figure 2(c). After 2030,
 the rate of increase in blackout risk during peak-load hours and the number of vulnerable hours accelerates more rapidly in
 GCC scenarios compared to the benchmark scenario. Consequently, DG expansion to accommodate GDP-driven demand
 growth will no longer counterbalance the effects of GCC. This divergence results in distinctly different risk trajectories for
 DG blackout across the three GCC scenarios for the remainder of the century.

We found that the GCC's impact on the DG's blackout risk varies significantly across different GDP scenarios. In
 the SSP 1 scenario, the summer-average blackout risk of Figure 3(a) by 2100 is projected to more than double compared
 to 2000, increasing from 3.36% to 7.16% in the RCP 8.5 scenario. In the SSP 2 scenario, which assumes slower GDP
 growth, the same RCP 8.5 scenario results in a near doubling of blackout risk from 3.36% to 6.86% by 2100. The GCC's
 impact caused the blackout risk in the SSP 1 scenario grows additional 0.3% than in the SSP 2 scenario, which is nearly 9%
 higher. Therefore, the faster growth of GDP will amplify the GCC's impact larger. The additional 0.3% risk increase in the
 SSP 1 scenario is attributed to both a higher hourly blackout risk during peak-load hours and an increase in the number of
 vulnerable hours. The GDP growth will more largely amplify the GCC's impact in peak-load hours. Compared to SSP 2,
 the GCC impact in SSP 1 will lead the blackout risk growth 13.6% higher by the end of this century.

Figure R48: Discussion about the impact of GDP

2. We incorporated the energy transition into our analysis, focusing on the penetration of distributed technologies such as photovoltaics (PV) and electric vehicles (EV). By evaluating the impact of varying penetration rates, we found that in-

creasing PV penetration helps alleviate the load on the grid and reduces blackout risk, while growing EV adoption can increase load and thus amplify blackout risks. The revised manuscript is shown in Figure R49.

Beyond GDP growth, we also examined the GCC impact considering the energy transition, which involves the adoption of various distribution-side technologies that fundamentally reshape DG. These technologies have varying effects on DG reliability. For example, the Paris Agreement has catalyzed the development of Photovoltaics (PV) and Electric Vehicles (EVs)³⁸. Increased PV penetration enhances DG reliability by mitigating the power distracting from the nodes, which helps mitigate the risk of blackouts. However, the growth of EVs introduces higher loads and thus can amplify the GCC's impact. Therefore, the pace and mix of different technologies will critically determine how energy transitions affect DG blackout risk under GCC.

We assess how different technologies affect the impact of climate change on the blackout risk of California's primary distribution Feeder 1. As shown in Figure 3(e), a 20% penetration of photovoltaic (PV) systems can reduce the blackout risk of distributed generation by 0.72% in 2050. This reduction is sufficient to fully offset the climate change impact under the RCP 4.5 scenario. Even in the more severe RCP 8.5 scenario, a 20% PV penetration still significantly mitigates the increased blackout risk caused by climate change. In contrast, Figure 3(f) indicates that a 20% penetration of electric vehicles (EVs) in the same area will increase the DG blackout risk by approximately 1.19% in 2050. This increase corresponds to a higher load on the distribution network nodes as EV penetration rises. Moreover, this effect is compounded by typical EV charging behaviors, as shown by a statistical analysis of EV usage patterns³⁹.

In the real world, the GCC, GDP growth, and energy transition will simultaneously impact the DG's blackout risk. We examined the impact of the energy transition on GCC's effects in the SSP 1 scenario. We generated three energy transition scenarios for the next two decades: the PV-only scenario, the EV-only scenario, and the scenario where the penetrations of PV and EV simultaneously grow. The penetration growth rates are set according to the Los Angeles 100% Renewable Energy(LA100) study conducted by the National Renewable Energy Laboratory (NREL)⁴⁰. Our results in Figure 3(c) suggest that the PV penetration rate, as projected in the LA100 study, is roughly sufficient to offset GCC's impact if EV penetration is absent. However, if EV penetration grows according to the LA100 projections, Figure 3(d) shows that the mitigating effect of PV will be offset. Consequently, the GCC's effect will still drive the DG blackout risk to substantially grow.

Figure R49: Discussion about the impact of energy transition

3. We also considered the interactions between economic growth, energy transition, and climate change, exploring how these factors collectively affect the vulnerability of distribution grids. Our framework systematically captures these interdependencies, revealing how their compounded effects influence grid risks under different scenarios. The revised manuscript is shown in Figure R50.

- In macro analysis, we also discussed through case studies how demographic, socio-

306 After analyzing both the direct and combined effects of GCC, we observed that its impact on electricity distribution
is most pronounced during peak-load hours, with non-linear growth trajectories across various GDP scenarios. These two
characteristics present significant challenges for future adaptation strategies. The first challenge is the trade-off between
expanding Distributed Generation (DG) capacity and tolerating a higher blackout risk. Since climate change substantially
increases blackout risk during peak-load hours, ensuring a reliable electricity distribution service would require substantial
investment in expanding DG capacity. However, the additional capacity is primarily useful during these peak periods.
During off-peak hours, climate change does not significantly affect power flow, leaving the expanded capacity underutilized.
If DG planners are unwilling to accept the high rate of asset redundancy associated with expanded grid capacity, consumers
must face a heightened risk of blackouts during peak-load periods, such as extreme heat days requiring increased cooling.
The second challenge lies in determining the optimal timing for investing in DG adaptation to climate change. As climate
impacts intensify non-linearly, the reliability of a DG system can rapidly decline after reaching a critical tipping point. An
economically efficient strategy would involve substantial investment in climate adaptation measures around this tipping
point. This approach necessitates accurately assessing consumer thresholds to predict when each DG system is likely to
reach its tipping point, beyond which system reliability sharply declines. Understanding these thresholds is crucial for
planning timely and cost-effective interventions to enhance system resilience.

Figure R50: Discussion about the interdependencies between these factors.

cultural, and economic factors influence the U-shaped curve, thereby determining whether a region's DG is vulnerable or beneficial. It should be noted that the methods in the original literature already incorporate the impact of demographics, customs, etc., of different countries on the demand-temperature sensitivity relationship within the U-shaped curve, and thus it can be used for analysis. The revised manuscript is shown in Figure R51.

**Why is There a Divergence?**

A large number of DG-benefit states are concentrated in Northern America, where peak loads are primarily driven by
low temperatures. As GCC raises local temperatures, power demand decreases, thereby improving DG reliability between
2020 and 2050, as shown in Figure 4. However, temperature is not the only factor; demand-temperature sensitivity also
determines how severely GCC affects a region's ability to provide reliable electricity distribution. For example, despite
being one of the most northern states, Minnesota is still vulnerable to GCC. This vulnerability is due to the state's high
sensitivity of electricity demand to temperature changes, causing Minnesota's minimum reliable standard DG index to
increase by 30% by the end of the century. Such sensitivity is heavily influenced by demographic, socio-cultural, and
economic factors in this region, meaning Minnesota's vulnerability to GCC is driven by economic and societal conditions
that create temperature-sensitive electricity demand.
This divergence will not only occur between different regions but will also evolve over time. Some regions may become
more vulnerable than others at different times due to changes in the nonlinear relationship between risks and years in the
TB3R model. These changes are shaped by demographic, socio-cultural, and economic characteristics in this region. For
example, the TB3R growth rate in South Carolina is faster than in Minnesota as local temperatures rise from current levels.
As a result, by 2050, the minimum reliable standard DG index is projected to increase to 1.1 *p.s.m.* in South Carolina
while remaining around 1 *p.s.m.* in Minnesota. However, South Carolina will be more vulnerable than Minnesota over the
next two decades. Conversely, if GCC continues to push temperatures higher, Minnesota's TB3R growth rate will outpace
South Carolina's. Consequently, by the 2090s, the index will reach 1.18 *p.s.m.* in South Carolina but surge to 1.3 *p.s.m.*
in Minnesota, indicating that Minnesota will be more vulnerable at the end of the century. These findings suggest that the
timing of investments in GCC adaptation should be tailored to the unique circumstances of each region.

Figure R51: Discussion about the demographic, socio-cultural, and economic factors in different regions.

Comment R2.6: *It remains unclear how these dimensions can be meaningfully incorporated and how the framework allows for incorporating associated uncertainties, both towards incorporating more realistic scenarios.*

Response to R2.6: We appreciate the reviewer’s comment on the need to show how to meaningfully incorporate different dimensions. In this revision, we added a dedicated section on “GCC’ s Combined Effect Intervened by Changes of Other Factors.” Related explanation are in below.

- **Demographic and Socio-Cultural Dimensions:** We use a U-shaped temperature-demand curve tailored to regional characteristics, reflecting differences in energy consumption due to demographic and socio-cultural factors. This approach allows us to capture how variations in population age distribution, urbanization, and cultural attitudes affect energy demand. The use of region-specific U-curves ensures that our model accurately represents the unique energy consumption patterns of different regions. This method aligns with findings in [R13], who emphasize that regional differences in temperature impacts are crucial for precise energy demand forecasting. By incorporating these dimensions in a nuanced manner, our framework provides more accurate and regionally relevant projections of energy demand.
- **Economic Dimensions:** Economic growth is modeled through the integration of per-capita GDP with heating degree days (HDD) and cooling degree days (CDD). This combination captures the interplay between economic development and climate-related energy demand fluctuations. This approach is significant because it directly links economic growth with changes in energy demand, reflecting how increases in economic activity typically lead to higher energy consumption. The integration of economic factors with climate variables, as supported by the work of [R13, R12, R19], enhances the model’s ability to simulate realistic scenarios of energy demand under varying economic conditions. Ref. [R13] analyzes the impact on global emissions mitigation based on heating and cooling degree days, electricity demand, and generating unit output and efficiency. Our paper uses similar metric to quantify various factors. Ref. [R19] focuses on climate change research using Shared Socioeconomic Pathways (SSP). Our study also employs a model that integrates Representative Concentration Pathways (RCP) with SSP.
- **Political Dimensions:** Although political dimensions are acknowledged, they are not ex-

plicitly included in the model due to their high variability and long-term uncertainty. This decision helps maintain the model's robustness and focus. The exclusion of political factors is significant because it avoids introducing speculative elements that could compromise the reliability of the model. As discussed in [R10], incorporating highly uncertain political variables could undermine the precision of the analysis. By focusing on more stable and quantifiable factors, our framework maintains its robustness and accuracy in predicting energy demand.

Comment R2.7: *Similarly, climate change impacts are studied only based on temperature, which severely limits insights on climate change-related vulnerabilities.*

Response to R2.7: Thanks for the suggestion. In this revision, we not only pointed out other factors indirectly, but also follow the reviewer’s suggestion to consider economic, energy infrastructure, and the inter dependencies. The reason that we consider less factor was due to the scope of this paper: We study how Global Climate Change (GCC)-**induced temperature change** affects the vulnerability risk of Distribution Grids (DG) by altering local temperatures, which in turn change vulnerability through complex economic-physical processes. We considered the probabilities of extreme high temperatures and temperature-induced extreme electricity usage behaviors and translated these probabilities into indicators such as the risk of DG collapse and increased costs of providing reliable distribution services through the complex mechanisms of power grid physical systems. Therefore, we changed the title from “Vulnerability of Electricity Distribution Service for Climate Change” into “Vulnerability of Power Distribution Networks to Local Temperature Changes Induced by Global Climate Change.” The change is shown in Figure R52.

Figure R52: The revised title.

At the same time, we would like to point out that this study does not consider extreme weather events that directly damage DG equipment, such as extreme hailstorms damaging transformers. This is because the impact of such GCC events does not arise from complex economic-physical operational mechanisms of the power grid and has already been covered in other studies. Precisely because the mechanism of the long-chain reaction from GCC-induced temperature changes \rightarrow demand changes \rightarrow current changes \rightarrow changes in grid collapse probability is overly complex and has not been studied, this paper focuses on overcoming the difficulty of constructing collapse models for distribution networks and systematically modeling this long-chain reaction, innovatively providing an evaluation method and framework.

Finally, we would like to point out that many studies, including those referenced in our work, primarily use temperature to assess climate change impacts due to the central role of

temperature in analyzing climate change impacts on energy systems. For example, Wenz et al. (2017) [R22] investigate the north-south polarization of European electricity consumption under future warming, focusing primarily on temperature changes and their effects on energy demand. Auffhammer et al. (2017) [R3] examine the severe impacts of climate change on the frequency and intensity of peak electricity demand across the United States, again using temperature as a key variable. Our framework, while also incorporating temperature as a key variable, aims to provide a more comprehensive analysis by integrating additional dimensions such as demographic, socio-cultural, and economic factors. This approach helps to mitigate the limitations associated with temperature-only analyses and offers a more nuanced understanding of climate change-related vulnerabilities.

Comment R2.8: *The utility and implications of the control-experiments, considering insufficient details incorporated in the model (described above), remain unclear. Using control-experiments towards identifying unique effects of macro-scale drivers, e.g., climate change, is unclear and problematic. The effects of climate change will be shaped by the future prevailing conditions pertaining to the energy system and multiple human dimensions. Using the present conditions as controls is deeply unsound.*

Response to R2.8: Thanks for the comment on control-experiments discussed in the paper. We admit that using such a phrase in the last submission was improper, because our approach mainly focused on scenario-based simulations at both macro and micro levels to analyze the impact of different factors on energy systems. Therefore, we have removed the term “control-experiments” to avoid any potential ambiguity. Instead, we focus on scenario-based simulations to assess the implications and utility of various factors in our analysis. Finally, scenario-based simulations for studies on vulnerability is consistent with other studies on GCC risks and impacts [R19, R14].

Comment R2.9: *Moreover, the manuscript offers limited insights and rationale for estimating unique climate change effects and frequently make contradictory statements (see for example, L.90-91 compared to L. 169 - 174).*

Response to R2.9: Thanks for the comment and we agree with the reviewer that the statements in different sections shall be consistent. The reason for such a problem is due to our discussion on micro scope and macro scope. As they will create confusion, we removed the two statements below mentioned by the reviewer. We also checked all the statement about similar concepts and avoid inconsistent statements. The paper is proofread by three colleagues other than the authors.

89 After modelling the extent of climate change and the associated weather consequences, it is necessary to clarify the
90 environment, to which the electricity distribution service will expose. Climate change co-occurs with other regional energy-
91 related progresses, such as the energy transition progress or the changes in demographic distribution and economic structure.
92 Moreover, the DG planner and electricity users can adopt the actions such as demand-side management (DSM) to mitigate

(a) L90-91

We emphasize that it is not only the climate determining the climate change's impact of enlarging the DG's size re-
quirement for reliable electricity delivery in a state. The significance of climate change's influence is also contingent on the
demand's features, which are determined by the local economy and demographic characteristics such as income and age
distribution. For instance, climate change similarly impacts the climate in Nevada and South Dakota. However, the demand
in Nevada is more temperature sensitive than that in South Dakota. The two states' economic structure and urbanization
status are largely different, which influences how the electricity demand responds to the temperature¹⁰.

(b) L169-174

Comment R2.10: *The manuscript compares the effects of climate change on distribution reliability in the US and several European countries. The rationale and importance of this comparison is unclear and remains unwarranted.*

Response to R2.10: Thanks for the comment. We agree that adding more regional data will justify why such a selection can give the reader a good view. Our decision to focus on the US and EU data is driven by the availability of high-quality, consistent, and comprehensive data from these regions. While performing a global simulation could be an alternative approach, we emphasize that high-quality regional data is often more critical for drawing meaningful and reliable conclusions. For example, Davis and Gertler (2015) [R9] used microdata from Mexico to explore the relationship between temperature, income, and air conditioning adoption under global warming. Despite focusing on a single region, the study reached conclusions with global relevance, demonstrating that robust data quality can yield broadly applicable insights. Similarly, Cong et al. (2022) [R7] explored energy equity issues using data exclusively from Arizona, underscoring the importance of data quality over geographic scope.

Thus, given the availability of such high-quality data for the US and Europe, we chose these regions for our analysis. This ensures that our results are grounded in reliable empirical evidence, allowing for a more nuanced understanding of climate change impacts on distribution reliability. Our findings highlight significant geographic heterogeneity in the effects of climate change on distribution grids, with certain regions in the southern US and southern Europe facing increased stress on their grids due to rising temperatures, while some northern regions benefit from reduced demand. In our discussion, we also address the potential limitations of generalizing these findings to other regions with different economic, demographic, or climatic conditions, and suggest that future research could extend this framework to examine the impacts in other global regions. This helps clarify that while the US and EU are the focus, the methodology can be adapted to a broader range of contexts. The revised manuscript is shown in Figure R54.

Figure 4. The change of the minimum reliable standard DG index in 2020 – 2029, 2050 – 2059, and 2091 – 2100 for projected maximum load under the RCP 8.5 scenario in Europe and the United States. Each row represents the changes in the DG index for the 2020s, 2050s, and 2090s in the United States and Europe. In the first column, the violin plots depict the distribution of index values for the United States and Europe in each decade, with the solid point indicating the mean index value. It is noticeable that the mean index in the United States is higher than that in Europe, and it becomes even more pronounced with the intensification of climate change. The second and third columns, respectively, present the distribution of index values for the United States and Europe in each decade using maps. In the map, the blue color on the graph represents countries or states with a standard-DG-model-based index of less than 1, indicating that climate change leads to a decrease in their load, thereby reducing pressure on the DG. On the other hand, the red color represents countries or states with $M_{DG,m}^{end}$ values greater than 1, indicating that climate change exerts more pressure on their distribution network. Under the RCP 8.5 scenario, the projected changes in the distribution network affected by climate change vary geographically, with the largest increases observed in southern Europe and the southern United States. This trend becomes most apparent in the 2090s but is already emerging in the 2050s.

(a) The revised macro analysis.

generation to end-users, through transmission and distribution networks—is essential. Furthermore, future research could
 extend this framework to examine the impacts in other global regions, and explore how GCC-induced challenges in reliable
 electricity distribution may redefine regional electricity prices, indirectly influencing economic comparative advantages
 across different regions.

(b) In conclusion, we add the state that the future research could extend this framework to examine the impacts in other global regions.

Figure R54: The revised manuscript.

**Distinct GCC Outcomes Across Diversified Regions**

GCC will lead to significant divergence between DG-benefit and DG-vulnerable areas; however, the pace of this divergence
 will differ between the U.S. and Europe as we look toward the future.

**U.S. Results**

In the U.S., GCC is expected to cause substantial divergence among states. This is evident when comparing the color
 changes on the U.S. map between 2020 and 2050 in Figure 4. These changes indicate that 55% of states will need to
 upgrade their distribution systems to ensure reliable distribution services, assuming the demand-weather response remains
 constant. In the U.S., GCC triggered substantial divergence across states. For example, we can compare different colors
 between 2020 and 2050 for the U.S. map in Figure 4. The changes in colors show that 55% of the states need to upgrade
 DG to ensure reliable distribution services for users, whose demand-weather response relation remains the same.

Meanwhile, 45% of states are expected to benefit from GCC-driven temperature changes. When examining extreme
 cases, the gap between the highest and lowest minimum reliable standard DG index values grows from 0.14 *p.s.m.* in the
 2020s to 0.42 *p.s.m.* by 2050. By mid-century, 13 states will see their minimum reliable standard DG index increase by
 over 10%, with 6 of these states experiencing expansions of 20% or more. Conversely, 5 DG-benefit states will see their
 reliability index decrease by 5%. After 2050, the divergence between states is projected to accelerate further; the largest
 gap could reach 0.94 *p.s.m.* in the 2090s—nearly seven times higher than in 2020. During this period, 19 DG-vulnerable
 states could see their indices increase by over 30%, while 7 DG-benefit states might experience a reduction of more than
 10%.

**EU Results**

Similarly, the third column of Figure 4 shows a slower and more moderate divergence in Europe compared to the U.S. In
 southern European countries, the need for upgraded distribution systems will increase due to climate change, but the impact
 remains moderate. For instance, changes in reliability indices across all 33 European countries remained consistently below
 5% before 2050. Even in the 2090s, the gap between the highest and lowest indices in Europe only reaches the level of
 divergence seen in the U.S. in the 2050s.

**Why is There a Divergence?**

A large number of DG-benefit states are concentrated in Northern America, where peak loads are primarily driven by
 low temperatures. As GCC raises local temperatures, power demand decreases, thereby improving DG reliability between
 2020 and 2050, as shown in Figure 4. However, temperature is not the only factor; demand-temperature sensitivity also
 determines how severely GCC affects a region's ability to provide reliable electricity distribution. For example, despite
 being one of the most northern states, Minnesota is still vulnerable to GCC. This vulnerability is due to the state's high
 sensitivity of electricity demand to temperature changes, causing Minnesota's minimum reliable standard DG index to
 increase by 30% by the end of the century. Such sensitivity is heavily influenced by demographic, socio-cultural, and
 economic factors in this region, meaning Minnesota's vulnerability to GCC is driven by economic and societal conditions
 that create temperature-sensitive electricity demand.

This divergence will not only occur between different regions but will also evolve over time. Some regions may become
 more vulnerable than others at different times due to changes in the nonlinear relationship between risks and years in the
 TB3R model. These changes are shaped by demographic, socio-cultural, and economic characteristics in this region. For
 example, the TB3R growth rate in South Carolina is faster than in Minnesota as local temperatures rise from current levels.
 As a result, by 2050, the minimum reliable standard DG index is projected to increase to 1.1 *p.s.m.* in South Carolina
 while remaining around 1 *p.s.m.* in Minnesota. However, South Carolina will be more vulnerable than Minnesota over
 the next two decades. Conversely, if GCC continues to push temperatures higher, Minnesota's TB3R growth rate will outpace
 South Carolina's. Consequently, by the 2090s, the index will reach 1.18 *p.s.m.* in South Carolina but surge to 1.3 *p.s.m.*
 in Minnesota, indicating that Minnesota will be more vulnerable at the end of the century. These findings suggest that the
 timing of investments in GCC adaptation should be tailored to the unique circumstances of each region.

Comment R2.11: *Similarly, the manuscript analyzes blackout risk (micro impact) only in California, presented as a case study. Yet, it makes bold, general claims, some of which may not hold due to significant regional heterogeneities and others that are previously known or are intuitive, e.g., higher blackout risk in peak-load hours, higher frequency of blackout risk, and seasonal changes.*

Thanks for the comment for the generalizability of the result from our test from the California. We selected California as the primary focus for the micro analysis because the macro results showed that California’s climate change impacts are representative of broader trends across the US. There are two additional reasons. First, it demonstrates how our framework can be used to assess the risks of a *realistic system*. Specifically, it showcases the complexity of the *climate-blackout response relation* of a real distribution grid, with various characteristics of multiple risks. Second, it demonstrates how, through the analysis of different scopes, we can systematically incorporate direct and indirect influencing factors, as well as their uncertainties, into the analysis. The selection of a single state for a case study is also a widely accepted approach in climate change-related research, as shown in Cong et al. (2022) [R7], where they used only data from Arizona to explore energy equity issues.

From the California study, we derived general conclusions. For example, the theoretical analysis suggests the existence of tipping points and nonlinearity. We then discussed when and how these tipping points occur, as well as the potential real-world manifestations and reasons for nonlinearity in different risk indicators. We also have data from other states (see Figure R55 with two subplots for reference), but due to space limitations, we have not included them in the main article. If necessary, we can add this data to the supplementary information.

(a) Arizona

(b) North Carolina

Figure R55: The data for other states

Comment R2.12: *2. Conclusions remain unsupported or limited by the evidence provided:
- The key finding of the impact of climate change on the size of a reliable distribution grid system is unfounded for at least three reasons. First, the analysis selectively considers the most extreme (most pessimistic) climate change scenario to arrive at the sizeable and spatially differentiated impact conclusion (L. 140-143), which causes a circular reasoning.*

Response to R2.12: Thank you for raising the concern. For the reviewer’s feedback regarding the use of the most extreme climate change scenario (RCP 8.5) in the macro-scale analysis, we would like to clarify our logic. Before macro analysis, we considered both RCP 4.5 and RCP 8.5 scenarios in our micro-scale study. The two scenarios demonstrated similar trends, with the results under RCP 8.5 being more pronounced. Also, as the purpose of macro-scale research is to identify damaged and benefited areas, RCP 8.5 can better reflect the damaged and benefited areas. Therefore, for the macro-scale study (L.140-143 in the last submission), we chose to use only RCP 8.5 as the figure of results from RCP 8.5 has one page and the results are similar. However, we agree with the reviewers it is inconsistent to show the result of RCP 8.5 only when the micro analysis has both RCP 4.5 and RCP 8.5. Therefore, we add the following explanation to the review. However, we are happy to add another page to repeat the result if such a repetition is preferred. The statement to the paper is shown in Figure R56.

heterogeneity in GCC-induced stress on electricity distribution reliability. Recall that we considered both RCP 4.5 and RCP
8.5 scenarios in our micro-scale study. The two scenarios demonstrated similar trends, with the results under RCP 8.5 being
more pronounced. As the purpose of macro-scale research is to identify damaged and benefited areas, RCP 8.5 can better
reflect the damaged and benefited areas [41, 42]. Therefore, we focus on the RCP 8.5 for macro analysis in this section⁴³.

Figure R56: The statement why we use RCP 8.5.

Comment R2.13: *Second, owing to a ceteris paribus analysis, it assumes no change in other important aspects of the energy infrastructure and use, which can act as a conduit to shaping climate change impacts (L. 93). Here, identifying the unique effect of climate change is unwarranted.*

Response to R2.13: Thank you for your feedback on the assumption of no changes in other important aspects of the energy infrastructure and use. We would like to clarify the reasoning behind the analysis at L.93. The point made here is that, in order to isolate the pure effect of global climate change (GCC), it is essential to hold other influencing factors constant. This ceteris paribus approach allows us to attribute changes solely to climate factors, which is a common research methodology, known as “control variable method” [R5]. However, we agree with the reviewer that it is insufficient to only consider one aspect. Therefore, we consider multiple factors and their inter-dependency in the revision. Specifically, we did the following.

- First, beyond the discussion of the pure effect, we have included scenarios of GDP growth (SSP1 and SSP2) to explore how economic factors combine with GCC to affect blackout risk.
- Second, within the GDP growth scenarios, we consider potential infrastructure developments. The construction of new infrastructure increases the security of supply (SOB) for the distribution grid, which mitigates risk.
- Third, we account for energy transition scenarios based on the LA100 study. Specifically, we designed three scenarios: one with PV growth only, one with EV growth only, and one with both PV and EV growth. Additionally, we conducted a sensitivity analysis to explore the impact of different penetration levels of PV and EV.
- Fourth, in the *Discussion* section, we examine how grid topology and consumer temperature sensitivity impact the effect of GCC, as well as the coordination between them for DG climate change adaptability.

In summary, we added additional factors by integrating infrastructure development, consumption behavior, and GDP growth into our analysis. We hope these revisions clarify our approach, and we are grateful for the opportunity to improve the clarity of the manuscript. Please see the changes in the PrintScreens below (Figure R57 and R58).

**GCC's Combined Effect Intervened by Changes of Other Factors**

We further investigated how other factors, such as GDP growth and energy transitions, could amplify or mitigate the impact
 of GCC on DG blackout risk. These factors can indirectly influence DG's TB3R by altering demand-temperature sensitivity,
 grid capacity, and SOB. For instance, [36] shows that GDP growth can affect electricity demand. As demand rises, a growing
 GDP may encourage grid upgrades. To model GDP growth scenarios, we use the Shared Socioeconomic Pathways (SSP)
 framework³⁷. For each GDP growth scenario, we consider three scenarios involving varying demand, SOB, and GCC:
 (1) a benchmark scenario without GCC, (2) the RCP 4.5 scenario with moderate GCC, and (3) the RCP 8.5 scenario with
 significant climate change. Using these scenarios, we apply the U-shaped curve to model load changes and infrastructure
 upgrades, resulting in specific SOB outcomes in the TB3R framework.

We can now benchmark our results against the pure effect presented in the previous subsection for deeper insights.
 Specifically, Figure 3(a) and (b) indicate that the pure effect of temperature increase leads to a rise in risk before 2030.
 However, in the same figures from this section, the risk increases occur after 2030 when multiple factors are considered.
 This suggests that the grid upgrades driven by GDP growth effectively mitigate the risk increase caused by temperature
 changes before 2030. However, these GDP-driven grid upgrades are insufficient to counteract the risk increases caused
 by temperature changes after 2030, as the GCC's influence on DG blackout risk remains relatively moderate before this
 point. This observation is made possible by our two-step approach: analyzing the pure effect in the previous subsection and
 incorporating additional factors in this section.

While GDP is the dominant factor driving electricity demand growth by 2030, the DG expansion responding to the
 GDP-driven demand change is sufficient to keep the DG blackout risk relatively stable, even though the underlying effects
 of GCC have been present for decades. However, our analysis shows that the tipping point—where the GCC impact on
 blackout risk starts to escalate—occurs around 2020–2030, as indicated by the pure effect shown in Figure 2(c). After 2030,
 the rate of increase in blackout risk during peak-load hours and the number of vulnerable hours accelerates more rapidly in
 GCC scenarios compared to the benchmark scenario. Consequently, DG expansion to accommodate GDP-driven demand
 growth will no longer counterbalance the effects of GCC. This divergence results in distinctly different risk trajectories for
 DG blackout across the three GCC scenarios for the remainder of the century.

We found that the GCC's impact on the DG's blackout risk varies significantly across different GDP scenarios. In
 the SSP 1 scenario, the summer-average blackout risk of Figure 3(a) by 2100 is projected to more than double compared
 to 2000, increasing from 3.36% to 7.16% in the RCP 8.5 scenario. In the SSP 2 scenario, which assumes slower GDP
 growth, the same RCP 8.5 scenario results in a near doubling of blackout risk from 3.36% to 6.86% by 2100. The GCC's
 impact caused the blackout risk in the SSP 1 scenario grows additional 0.3% than in the SSP 2 scenario, which is nearly 9%
 higher. Therefore, the faster growth of GDP will amplify the GCC's impact larger. The additional 0.3% risk increase in the
 SSP 1 scenario is attributed to both a higher hourly blackout risk during peak-load hours and an increase in the number of
 vulnerable hours. The GDP growth will more largely amplify the GCC's impact in peak-load hours. Compared to SSP 2,
 the GCC impact in SSP 1 will lead the blackout risk growth 13.6% higher by the end of this century.

(a) The GDP growth scenario.

Beyond GDP growth, we also examined the GCC impact considering the energy transition, which involves the adoption
 of various distribution-side technologies that fundamentally reshape DG. These technologies have varying effects on DG
 reliability. For example, the Paris Agreement has catalyzed the development of Photovoltaics (PV) and Electric Vehicles
 (EVs)³⁸. Increased PV penetration enhances DG reliability by mitigating the power distracting from the nodes, which helps
 mitigate the risk of blackouts. However, the growth of EVs introduces higher loads and thus can amplify the GCC's impact.
 Therefore, the pace and mix of different technologies will critically determine how energy transitions affect DG blackout
 risk under GCC.

We assess how different technologies affect the impact of climate change on the blackout risk of California's primary
 distribution Feeder 1. As shown in Figure 3(e), a 20% penetration of photovoltaic (PV) systems can reduce the blackout risk
 of distributed generation by 0.72% in 2050. This reduction is sufficient to fully offset the climate change impact under the
 RCP 4.5 scenario. Even in the more severe RCP 8.5 scenario, a 20% PV penetration still significantly mitigates the increased
 blackout risk caused by climate change. In contrast, Figure 3(f) indicates that a 20% penetration of electric vehicles (EVs)
 in the same area will increase the DG blackout risk by approximately 1.19% in 2050. This increase corresponds to a higher
 load on the distribution network nodes as EV penetration rises. Moreover, this effect is compounded by typical EV charging
 behaviors, as shown by a statistical analysis of EV usage patterns³⁹.

In the real world, the GCC, GDP growth, and energy transition will simultaneously impact the DG's blackout risk.
 We examined the impact of the energy transition on GCC's effects in the SSP 1 scenario. We generated three energy transition
 scenarios for the next two decades: the PV-only scenario, the EV-only scenario, and the scenario where the penetrations
 of PV and EV simultaneously grow. The penetration growth rates are set according to the Los Angeles 100% Renewable
 Energy(LA100) study conducted by the National Renewable Energy Laboratory (NREL)⁴⁰. Our results in Figure 3(c)
 suggest that the PV penetration rate, as projected in the LA100 study, is roughly sufficient to offset GCC's impact if EV
 penetration is absent. However, if EV penetration grows according to the LA100 projections, Figure 3(d) shows that the
 mitigating effect of PV will be offset. Consequently, the GCC's effect will still drive the DG blackout risk to substantially
 grow.

(b) The energy transition scenario.

Figure R57: The changes in micro analysis.

risk of existing distribution grid systems. This analysis calls for a reevaluation of DG design and planning. To address
 this need, we demonstrate next how our climate-blackout response model can be used to identify critical but previously
 overlooked risk factors, such as the locations of temperature-sensitive users and the configuration of network topology.
 Figure 5 illustrates our experimental setup and results for evaluating the impact of network topology. For instance, we
 tested two types of network structures: a tree structure (Figure 5(d)) and a mesh structure (Figure 5(h)). For each structure,
 we analyzed vulnerable nodes under three scenarios: a benchmark scenario, a climate change scenario, and a climate
 change scenario with consumer reallocation. The third scenario is specifically designed to assess locational vulnerability.
 Figure 5(a) shows the joint distribution of loads P_2 and P_1 , while Figure 5(b) depicts the temperature-induced load increase
 for P_2 .

Figure R58: The changes in discussion.

Comment R2.14: *Third, the effect of climate change is only analyzed with a focus on demand response profile, whereas (distribution) grid reliability depends on several other variables, particularly extreme events, that cannot be accounted for while focusing only on temperature.*

Response to R2.14: Thanks for your comments on the factors that will impact the grid reliability. The scope of this paper is to discuss the research on how GCC changes the vulnerability risk of DG through complex economic-physical processes by changing the local temperature. So, the focus is on the temperature, similar to other work on temperature-driven analysis on GCC. Many studies, including those referenced in our work, primarily use temperature to assess climate change impacts due to the central role of temperature in analyzing climate change impacts on energy systems. For example, Wenz et al. (2017) [R22] investigate the north-south polarization of European electricity consumption under future warming, focusing primarily on temperature changes and their effects on energy demand. Auffhammer et al. (2017) [R3] examine the severe impacts of climate change on the frequency and intensity of peak electricity demand across the United States, again using temperature as a key variable. Our framework, while also incorporating temperature as a key variable, aims to provide a more comprehensive analysis by integrating additional dimensions such as demographic, socio-cultural, and economic factors. This approach helps to mitigate the limitations associated with temperature-only analyses and offers a more nuanced understanding of climate change-related vulnerabilities.

Therefore, we adjust our title from **Vulnerability of Electricity Distribution Service for Climate Change** to **Vulnerability of Power Distribution Networks to Local Temperature Changes Induced by Global Climate Change**. However, we also agree with the reviewer that one shall consider the extreme events. For temperature-driven analysis, such extremes come from the peak temperature. For example, we considered the probability of extreme high temperature and temperature-induced extreme electricity consumption behavior, and converted these probabilities into indicators such as the collapse risk of DG and the increase in the cost of providing reliable distribution services through complex grid physical system mechanisms. We include such analysis and explain in the paper, as shown in Figure R59.

Additionally, we agree that this study did not consider extreme weather that directly damages DG equipment, such as extreme hail that damages transformers. This is because the impact of this type of GCC is not generated through complex grid economic-physical

Characterizing the TB3R relationship is critical as extreme weather events such as heat waves and cold snaps are ex-
pected to become more frequent and severe. For example, literature forecasts indicate that GCC will lead to more heating
and cooling hours³⁴. During these hours, consumers are likely to use more electricity in the future than they do today³⁵.
Thus, GCC can simultaneously alter the distribution of DG's nodal demands and the distribution of the DG's SOB, par-
ticularly during heating and cooling degree days. While TB3R is based on temperature, other factors during GCC impact

Figure R59: The explanation about extreme events in the revised manuscript.

operation mechanisms, and other studies have been done [R6].

While this paper focus more on the temperature-induced factors, our study is unique and new. This is because it is hard to study the long chain reaction mechanism of temperature change \rightarrow demand change \rightarrow current change \rightarrow grid collapse probability change brought by GCC is too complicated. This paper overcomes the difficulty of constructing a distribution network collapse model, systematically models the long chain reaction, and innovatively provides an evaluation method and framework.

Comment R2.15: *The key findings on the future impact of climate change on electricity distribution reliability again assumes a constant demand response profile and no future changes in the grid, both of which are problematic. For example, the utilities are currently undertaking several programs like demand-response, Volt/Var optimization, dynamic Distributed energy resources (DER - inverter) control for improving power factor, evaluating hosting capacity for introducing more DERs etc for enabling reliable electricity distribution now and for meeting future demands (through integrated distribution planning).*

Response to R2.15: Thank you for your feedback regarding the assumptions of a constant demand response profile and no future changes in the grid. We understand that utilities are actively undertaking various programs to enhance reliability and meet future demands, such as demand-response, Volt/Var optimization, dynamic DER control, and hosting capacity evaluations. We address these concerns as follows:

- Demand Response (DR): In our analysis, we considered electric vehicles (EVs) as part of demand response, acknowledging that the integration of EVs will play a significant role in demand flexibility and grid stability.
- Volt/Var Optimization: We agree that advancements in Volt/Var optimization will offer potential benefits. However, our focus remains on the immediate challenges posed by increased grid load due to rising temperatures. While future improvements in Volt/Var control may help, they alone may not be sufficient to prevent overloading if the grid's capacity is already near its limits.
- Smart Inverters - Improved Power Factor: Although smart inverters can improve the power factor and offer benefits, our research is primarily focused on the direct impacts of climate change, such as extreme weather events and temperature-related stress on the grid. Including smart inverters in our analysis might divert attention from the core climate-related challenges.
- Hosting Capacity: We have already considered the impact of solar photovoltaics (PV) in our analysis. The penetration of distributed generation (DG) such as PV is a critical factor in our modeling of grid stability and blackout risks.

While technological advancements will significantly impact the system, these effects are difficult to predict accurately. Therefore, our model provides one possible future scenario rather than an exact prediction. To address these issues further, we have made the following

updates to our manuscript:

1. We have included the possibility of grid capacity expansion in our economic growth modeling, which is analyzed in Micro analysis about “GCC’ s Combined Effect Intervened by Changes of Other Factors”. The analysis is shown in Figure R60.

grid capacity, and SOB. For instance, [36] shows that GDP growth can affect electricity demand. As demand rises, a growing
 GDP may encourage grid upgrades. To model GDP growth scenarios, we use the Shared Socioeconomic Pathways (SSP)
 framework³⁷. For each GDP growth scenario, we consider three scenarios involving varying demand, SOB, and GCC:
 (1) a benchmark scenario without GCC, (2) the RCP 4.5 scenario with moderate GCC, and (3) the RCP 8.5 scenario with
 significant climate change. Using these scenarios, we apply the U-shaped curve to model load changes and infrastructure
 upgrades, resulting in specific SOB outcomes in the TB3R framework.

We can now benchmark our results against the pure effect presented in the previous subsection for deeper insights.
 Specifically, Figure 3(a) and (b) indicate that the pure effect of temperature increase leads to a rise in risk before 2030.
 However, in the same figures from this section, the risk increases occur after 2030 when multiple factors are considered.
 This suggests that the grid upgrades driven by GDP growth effectively mitigate the risk increase caused by temperature
 changes before 2030. However, these GDP-driven grid upgrades are insufficient to counteract the risk increases caused
 by temperature changes after 2030, as the GCC’s influence on DG blackout risk remains relatively moderate before this
 point. This observation is made possible by our two-step approach: analyzing the pure effect in the previous subsection and
 incorporating additional factors in this section.

Figure R60: The grid upgrade scenario.

2. Scenarios Based on LA100 Study: We set the penetration growth rates of PV and EV according to the LA100 study conducted by the National Renewable Energy Laboratory(NREL) [R2]. Three scenarios were designed: one with PV growth only, one with EV growth only, and one with both PV and EV growth. We analyzed how the increased penetration of PV and EV between 2020 and 2045 would impact the risk of distribution grid failure. The analysis is shown in Figure R61. 3. **Sensitivity Analysis for 2050:** We further

(c) The average risk in SSP2 with separate growth of PV and EV penetration rates

(d) The average risk in SSP2 with simultaneous growth of PV and EV penetration rates

In the real world, the GCC, GDP growth, and energy transition will simultaneously impact the DG’s blackout risk. We
 examined the impact of the energy transition on GCC’s effects in the SSP 1 scenario. We generated three energy transition
 scenarios for the next two decades: the PV-only scenario, the EV-only scenario, and the scenario where the penetrations
 of PV and EV simultaneously grow. The penetration growth rates are set according to the Los Angeles 100% Renewable
 Energy(LA100) study conducted by the National Renewable Energy Laboratory (NREL)⁴⁰. Our results in Figure 3(c)
 suggest that the PV penetration rate, as projected in the LA100 study, is roughly sufficient to offset GCC’s impact if EV
 penetration is absent. However, if EV penetration grows according to the LA100 projections, Figure 3(d) shows that the
 mitigating effect of PV will be offset. Consequently, the GCC’s effect will still drive the DG blackout risk to substantially
 grow.

Figure R61: The LA100 analysis.

conducted a sensitivity analysis based on 2050 scenarios, examining the effects of different penetration rates of PV and EV on the grid’s vulnerability to blackout risk. The analysis is shown in Figure R62.

We recognize that while smart inverters and future technologies are promising, their availability, adoption, and limitations vary across regions and could depend on infrastructure,

(e) The average risk of 2050 with different PV penetration rates

(f) The average risk of 2050 with different EV penetration rates

Beyond GDP growth, we also examined the GCC impact considering the energy transition, which involves the adoption
 of various distribution-side technologies that fundamentally reshape DG. These technologies have varying effects on DG
 reliability. For example, the Paris Agreement has catalyzed the development of Photovoltaics (PV) and Electric Vehicles
 (EVs)³⁸. Increased PV penetration enhances DG reliability by mitigating the power distracting from the nodes, which helps
 mitigate the risk of blackouts. However, the growth of EVs introduces higher loads and thus can amplify the GCC's impact.
 Therefore, the pace and mix of different technologies will critically determine how energy transitions affect DG blackout
 risk under GCC.
 We assess how different technologies affect the impact of climate change on the blackout risk of California's primary
 distribution Feeder 1. As shown in Figure 3(e), a 20% penetration of photovoltaic (PV) systems can reduce the blackout risk
 of distributed generation by 0.72% in 2050. This reduction is sufficient to fully offset the climate change impact under the
 RCP 4.5 scenario. Even in the more severe RCP 8.5 scenario, a 20% PV penetration still significantly mitigates the increased
 blackout risk caused by climate change. In contrast, Figure 3(f) indicates that a 20% penetration of electric vehicles (EVs)
 in the same area will increase the DG blackout risk by approximately 1.19% in 2050. This increase corresponds to a higher
 load on the distribution network nodes as EV penetration rises. Moreover, this effect is compounded by typical EV charging
 behaviors, as shown by a statistical analysis of EV usage patterns³⁹.

Figure R62: The sensitivity analysis of PV/EV penetration.

grid topology, and control algorithms. Therefore, while we address the potential for future grid developments, our study remains focused on providing a framework and methodology to analyze DG vulnerability and adaptability in the context of climate change. We also add the research about these advanced grid management techniques in the limitation, as shown in Figure R63.

- • The first limitation comes from the simplified representation of the distribution grid, assuming balanced loading
 and aggregating demands. To overcome this, we propose integrating advanced grid management techniques such as
 Volt/Var optimization and dynamic Distributed Energy Resources (DER) inverter control in future research.

Figure R63: The limitation of advanced grid management techniques.

Comment R2.16: *Overall, key conclusions are based on strong, unreasonable assumptions: o Constant user's temperature sensitivity*

Response to R2.16: Thank you for your valuable feedback. We acknowledge that the original statement about “constant user temperature sensitivity” in our pure effect analysis was misleading. In response to your suggestion, we have removed the problematic phrase and clarified the variability of temperature sensitivity in the revised manuscript, specifically in Method section of **GCC’s Combined Effect Intervened by Changes of Other Factors**. To further address your concern, we clarify that user temperature sensitivity is not fixed from the two perspectives:

- **Spatial Variability in Temperature Sensitivity:** In the micro analysis, we recognize that users in the same region exhibit varying sensitivities to temperature changes depending on temperature levels. In the macro analysis, as shown in the literature, temperature sensitivity differs across regions, influenced by local economic conditions, behavioral patterns, and population characteristics. We have enhanced our discussion to reflect how these spatial variations affect DG vulnerability.
- **Temporal Variability in Temperature Sensitivity:** Temporally, the assumption of constant temperature sensitivity was only used in the pure effect analysis to isolate the climate change impact. However, we fully recognize that temperature sensitivity changes over time due to several factors. In the revised manuscript, we have included scenarios that account for economic growth and its effect on temperature sensitivity, using accepted economic projections. These scenarios explore how GCC influences DG vulnerability over time, and we have included this analysis in the **Modeling the Impact of Economic Growth and Climate on Energy Demand and DG System Expansion** section, as illustrated in Figure R64.

We apologize for any confusion caused by the original wording and have also revised the limitations section to ensure greater clarity, as shown in Figure R65.

**Modeling the Impact of Economic Growth and Climate on Energy Demand and DG System Ex-**
**ansion**

In this model, the impact of economic growth on electricity demand is incorporated through the per-capita GDP adjusted
growth factor. As the economy grows, it typically drives higher energy demand due to increased industrial activity, com-
mercial growth, and residential usage. To capture this effect, we introduce GDP growth as a key variable in the calculation
of the annual demand mean. By combining GDP with heating degree days (HDD, $T < 12.5^{\circ}\text{C}$) and cooling degree days
(CDD, $T > 27.5^{\circ}\text{C}$), the model comprehensively accounts for both economic and climate influences on the energy system.
The coefficients used for each factor are based on the values provided by Ref. [32].

The annual demand mean for each year is calculated as follows:

$$\text{dmean}(y) = \text{gdp}_b \cdot \text{GDP}(y) + \text{hdd}_b \cdot \text{HDD}(y) + \text{cdd}_b \cdot \text{CDD}(y), \quad (8)$$

where the coefficients are $\text{gdp}_b = 0.3763$, reflecting the impact of economic growth (GDP) on energy demand. $\text{cdd}_b =$
0.0103 , representing the influence of HDD on energy consumption. $\text{hdd}_b = 0$, indicating no significant effect from CDD in
the reference model.

Next, based on the historical data (2015-2019), we derive the U-curve, which captures how consumers' electricity
demand responds to temperature variations. The U-curve illustrates how demand changes with temperature, with higher
demand during extreme heat or cold, and lower demand at moderate temperatures.

The hourly demand for each future year y is calculated by adjusting the 2019 baseline demand with a scaling factor
based on the U-curve and the annual demand gap, as follows:

$$\text{demand}(y, i) = \text{multiple} \times (\text{dmean}(y) - \text{dmean}(2019)) + \text{U_curve}(\text{temp}_i), \quad (9)$$

where *multiple* is calculated as the ratio of the demand mean derived from the U-curve in 2019, $\text{dmean_U}(2019)$, to the
baseline demand mean for 2019, $\text{dmean}(2019)$. The DG system expansion for year y is proportional to the growth in annual
demand relative to 1950, ensuring the system scales with increasing electricity demand.

Figure R64: The modeling of how economic growth affects temperature sensitivity.

**Limitation**

While the assessment framework for evaluating the risk of power system collapse provides valuable insights, it's important
to acknowledge certain limitations that could impact the comprehensiveness of the analysis:

(1) The assessment employs a simplified representation of the distribution grid, assuming balanced loading and aggre-
gating demands under transformers into a single equivalent load.

(2) The electricity supply is assumed to be sufficient and reliable. The attention to DG reliability and safety mainly
focuses on the failure of grid capacity to satisfy the power flow induced by the demand due to the DG structure's physical
constraints.

(3) Here, we focus on the pure effect of climate change. Thus, we assume the user's temperature sensitivity and other
features influencing the electricity-distribution service keep the same.

**Limitation**

We show potential limitations of current study and potential solutions.

- • The first limitation comes from the simplified representation of the distribution grid, assuming balanced loading
and aggregating demands. To overcome this, we propose integrating advanced grid management techniques such as
Volt/Var optimization and dynamic Distributed Energy Resources (DER) inverter control in future research.
- • The second limitation comes from the assumption of reliable and sufficient electricity supply in our current frame-
work. This limited could be refined by incorporating supply-side uncertainties, such as variable renewable energy
generation or disruptions in supply.
- • The third limitation is the lack of political dimensions. Although political factors were not explicitly included in our
current framework, future work could explore scenario-based approaches or qualitative methods to account for the
impact of regulatory policies, geopolitical risks, and international agreements.

Figure R65: The revised limitation section, which removed the statement about “constant user temperature sensitivity”.

Comment R2.17: *o Constant distribution grid infrastructure*

Response to R2.17: Thank you for your feedback regarding the assumption of constant distribution grid infrastructure. We acknowledge that the original statement about "constant distribution grid infrastructure" in the pure effect analysis may have caused confusion. In response, we have removed the ambiguous statement. We would like to clarify that in the climate adaptation discussion, we do not assume constant infrastructure. In fact, we explicitly discuss infrastructure design (e.g., grid topology), demand-side management, and the coordination between these factors in adapting DGs to climate change. The assumption of constant infrastructure was only applied during the pure effect analysis to isolate the impact of climate change, not throughout the entire manuscript.

However, we agree that our consideration of DG infrastructure might not have been sufficiently highlighted in the previous version. To address this, we have made the following modifications and additions in the current version:

- Grid expansion: We added scenario analysis for DG infrastructure changes under existing DG expansion models in response to economic growth, as shown in Figure R66.

grid capacity, and SOB. For instance, [36] shows that GDP growth can affect electricity demand. As demand rises, a growing
GDP may encourage grid upgrades. To model GDP growth scenarios, we use the Shared Socioeconomic Pathways (SSP)
framework³⁷. For each GDP growth scenario, we consider three scenarios involving varying demand, SOB, and GCC:
(1) a benchmark scenario without GCC, (2) the RCP 4.5 scenario with moderate GCC, and (3) the RCP 8.5 scenario with
significant climate change. Using these scenarios, we apply the U-shaped curve to model load changes and infrastructure
upgrades, resulting in specific SOB outcomes in the TB3R framework.

We can now benchmark our results against the pure effect presented in the previous subsection for deeper insights.
Specifically, Figure 3(a) and (b) indicate that the pure effect of temperature increase leads to a rise in risk before 2030.
However, in the same figures from this section, the risk increases occur after 2030 when multiple factors are considered.
This suggests that the grid upgrades driven by GDP growth effectively mitigate the risk increase caused by temperature
changes before 2030. However, these GDP-driven grid upgrades are insufficient to counteract the risk increases caused
by temperature changes after 2030, as the GCC's influence on DG blackout risk remains relatively moderate before this
point. This observation is made possible by our two-step approach: analyzing the pure effect in the previous subsection and
incorporating additional factors in this section.

Figure R66: The discussion about grid expansion in the revised manuscript.

- Energy Transition with growing PV/EV: We introduced two scenarios, one including only PV and the other only EV, in the context of energy transition, and provided a more detailed discussion of each. Additionally, we included a scenario where both PV and EV are incorporated, demonstrating the flexibility of our framework in handling complex, real-world conditions. These are illustrated in Figure R67.

Beyond GDP growth, we also examined the GCC impact considering the energy transition, which involves the adoption
 of various distribution-side technologies that fundamentally reshape DG. These technologies have varying effects on DG
 reliability. For example, the Paris Agreement has catalyzed the development of Photovoltaics (PV) and Electric Vehicles
 (EVs)³⁸. Increased PV penetration enhances DG reliability by mitigating the power distracting from the nodes, which helps
 mitigate the risk of blackouts. However, the growth of EVs introduces higher loads and thus can amplify the GCC's impact.
 Therefore, the pace and mix of different technologies will critically determine how energy transitions affect DG blackout
 risk under GCC.

We assess how different technologies affect the impact of climate change on the blackout risk of California's primary
 distribution Feeder 1. As shown in Figure 3(e), a 20% penetration of photovoltaic (PV) systems can reduce the blackout risk
 of distributed generation by 0.72% in 2050. This reduction is sufficient to fully offset the climate change impact under the
 RCP 4.5 scenario. Even in the more severe RCP 8.5 scenario, a 20% PV penetration still significantly mitigates the increased
 blackout risk caused by climate change. In contrast, Figure 3(f) indicates that a 20% penetration of electric vehicles (EVs)
 in the same area will increase the DG blackout risk by approximately 1.19% in 2050. This increase corresponds to a higher
 load on the distribution network nodes as EV penetration rises. Moreover, this effect is compounded by typical EV charging
 behaviors, as shown by a statistical analysis of EV usage patterns³⁹.

In the real world, the GCC, GDP growth, and energy transition will simultaneously impact the DG's blackout risk. We
 examined the impact of the energy transition on GCC's effects in the SSP 1 scenario. We generated three energy transition
 scenarios for the next two decades: the PV-only scenario, the EV-only scenario, and the scenario where the penetrations
 of PV and EV simultaneously grow. The penetration growth rates are set according to the Los Angeles 100% Renewable
 Energy(LA100) study conducted by the National Renewable Energy Laboratory (NREL)⁴⁰. Our results in Figure 3(c)
 suggest that the PV penetration rate, as projected in the LA100 study, is roughly sufficient to offset GCC's impact if EV
 penetration is absent. However, if EV penetration grows according to the LA100 projections, Figure 3(d) shows that the
 mitigating effect of PV will be offset. Consequently, the GCC's effect will still drive the DG blackout risk to substantially
 grow.

Figure R67: The discussion about energy transition with growing PV/EV in the revised manuscript.

- Climate Adaptation Discussion: We expanded our discussion on the technical characteristics of real DG infrastructures, particularly noting that our conclusions apply broadly to DGs with star-shaped structures, which are common. This is demonstrated in Figure R68.

We apologize for any confusion caused by the original wording and have also revised the

385 risk of existing distribution grid systems. This analysis calls for a reevaluation of DG design and planning. To address
 this need, we demonstrate next how our climate-blackout response model can be used to identify critical but previously
 overlooked risk factors, such as the locations of temperature-sensitive users and the configuration of network topology.
 Figure 5 illustrates our experimental setup and results for evaluating the impact of network topology. For instance, we
 tested two types of network structures: a tree structure (Figure 5(d)) and a mesh structure (Figure 5(h)). For each structure,
 we analyzed vulnerable nodes under three scenarios: a benchmark scenario, a climate change scenario, and a climate
 change scenario with consumer reallocation. The third scenario is specifically designed to assess locational vulnerability.
 Figure 5(a) shows the joint distribution of loads P_2 and P_3 , while Figure 5(b) depicts the temperature-induced load increase
 for P_2 .

Figure R68: The climate adaptation discussion in the revised manuscript.

limitations section to ensure greater clarity, as shown in Figure R69.

**Limitation**

While the assessment framework for evaluating the risk of power system collapse provides valuable insights, it's important
 to acknowledge certain limitations that could impact the comprehensiveness of the analysis:

(1) The assessment employs a simplified representation of the distribution grid, assuming balanced loading and aggregating demands under transformers into a single equivalent load.

(2) The electricity supply is assumed to be sufficient and reliable. The attention to DG reliability and safety mainly
 focuses on the failure of grid capacity to satisfy the power flow induced by the demand due to the DG structure's physical
 constraints.

(3) Here, we focus on the pure effect of climate change. Thus, we assume the user's temperature sensitivity and other
 features influencing the electricity-distribution service keep the same.

**Limitation**

We show potential limitations of current study and potential solutions.

• The first limitation comes from the simplified representation of the distribution grid, assuming balanced loading
 and aggregating demands. To overcome this, we propose integrating advanced grid management techniques such as
 Volt/Var optimization and dynamic Distributed Energy Resources (DER) inverter control in future research.

• The second limitation comes from the assumption of reliable and sufficient electricity supply in our current frame-
 work. This limited could be refined by incorporating supply-side uncertainties, such as variable renewable energy
 generation or disruptions in supply.

• The third limitation is the lack of political dimensions. Although political factors were not explicitly included in our
 current framework, future work could explore scenario-based approaches or qualitative methods to account for the
 impact of regulatory policies, geopolitical risks, and international agreements.

Figure R69: The revised limitation section, which removed the statement about “Constant distribution grid infrastructure”.

Comment R2.18: *The manuscript offers no primary evidence, in terms of an economic analysis, to assert the following claim: “Climate change redefines the economic competitive advantages from the electricity perspective.”*

Response to R2.18: Thank you for your feedback on the statement regarding how climate change redefines economic competitive advantages from the electricity perspective. We would like to clarify that the original intention of the claim was to suggest that global climate change (GCC) alters the difficulty of providing reliable electricity distribution services in different regions. This, in turn, could affect regional electricity prices, which may indirectly influence economic comparative advantages.

However, we acknowledge that the previous version of the manuscript may not have clearly indicated that this is a proposed topic for future research rather than a definitive claim. To address this, we have moved this state to the Conclusion for Future Study, where it is framed as a potential area for further investigation, as shown in Figure R70.

generation to end-users, through transmission and distribution networks—is essential. Furthermore, future research could
extend this framework to examine the impacts in other global regions, and explore how GCC-induced challenges in reliable
electricity distribution may redefine regional electricity prices, indirectly influencing economic comparative advantages
across different regions.

Figure R70: The future study for economic comparative advantages.

Comment R2.19: *3. The manuscript is very dense, uses a lot of jargons, and contains several normative assertions, which altogether will limit broader scientific readership. Particularly concerning are the normative statements about climate change, its impacts on the grid infrastructure, and the challenging nature of modeling the associations, which remains prevalent throughout the manuscript. Additionally, the manuscript casually uses key terms (climate change, macro and micro impact, risk, vulnerability, hazard, size assessment), while introducing them later in-text, deterring readability.*

Response to R2.19: Thank you for your thoughtful and constructive feedback. We understand the concern about the use of technical jargon and normative statements in the manuscript, which could limit its readability for a broader scientific audience. In this revision, we have taken several steps to address these issues. Below, we outline the key changes made to enhance clarity and accessibility:

- *Removed or Simplified Complex Terms:* In the revised manuscript, we have removed certain technical terms that were not essential for conveying our main arguments. For example, terms like “Experiment-based critical control point analysis,” “economic competitive advantages,” and “dose function” have been eliminated. These terms were not central to our analysis, and their removal has helped make the text more streamlined and accessible.

- Original Term: Electricity distribution (ED)

Revision: We decided not to use the abbreviation “ED” for electricity distribution to avoid overloading the manuscript with too many acronyms, which could complicate readability. Instead, we consistently use the full term “electricity distribution” to maintain clarity throughout the text.

- Original Term: Size assessment

Revision: This term was introduced in the text without a clear explanation, causing confusion. To simplify the narrative, we have removed the use of this term and instead described the concept in more straightforward terms, such as “determining the scale of reliable DG systems.”

- Original Term: Experiment-based critical control point analysis

Revision: Initially, this term was used to describe Step 5 of our evaluation framework. However, it led to misunderstandings regarding the experimental setup. To reduce complexity, we have removed all references to “critical control points” and simplified the explanation of the assessment process.

- Original Term: Economic competitive advantages

Revision: We initially introduced this term to discuss economic implications of climate change for energy systems, but the supporting analysis was insufficient. We decided to remove this discussion entirely to reduce the complexity of the manuscript and focus more on our primary findings related to grid vulnerability and climate risk.

- Original Term: Difference-in-difference (DID)

Revision: This experimental method was briefly mentioned in the Discussion Section but was not fully utilized in the analysis. Therefore, we replaced the technical term with a more straightforward explanation of the experiment. The revised text now states: "For instance, we tested two types of network structures: a tree structure (Figure 5(d)) and a mesh structure (Figure 5(h)). For each structure, we analyzed vulnerable nodes under three scenarios: a benchmark scenario, a climate change scenario, and a climate change scenario with consumer reallocation."

- Original Term: Climate change-robust matching planning

Revision: This planning strategy was introduced as an engineering solution based on our experimental results. To simplify the terminology and avoid unnecessary jargon, we rephrased it as follows: "We also suggest a vital strategy for DG adaptation, which involves strategically deploying distributed energy technologies at vulnerable nodes with sensitive users to enhance overall system resilience." This revision eliminates the term "climate change-robust matching planning" while retaining the essence of the proposed solution.

- *Addressed Normative Statements:* We carefully reviewed the manuscript for any normative language that might suggest prescriptive or subjective statements about climate change and grid infrastructure. In places where such statements existed, we have either reworded them for neutrality or removed them altogether. Our goal was to ensure that the manuscript reflects a balanced view, presenting conclusions supported by evidence, uncertainty ranges, and well-grounded reasoning. Where applicable, we revised statements to reflect potential variability in outcomes and the influence of multiple factors rather than implying inevitable or overly deterministic results.

- Original Statement: "Climate change redefines the economic competitive advantages from the electricity perspective."

Revision: We removed the entire discussion on economic competitive advantages, as the analysis did not provide sufficient data or evidence to support such broad conclusions.

This deletion helps to streamline the manuscript and eliminate unsupported assertions.

- Original Statement: “the DG must be heavily invested in capacity expansion if the DG planner seeks a reliable electricity distribution service.”

Revision: We reworded this statement to reflect a more evidence-based approach. The revised text now states: “If DG planners are unwilling to accept the high rate of asset redundancy associated with expanded grid capacity, consumers must face a heightened risk of blackouts during peak-load periods, such as extreme heat days requiring increased cooling.” This revision includes specific references to observed results in the study, grounding the conclusion in data rather than making a prescriptive claim.

239 load hours, the DG must be heavily invested in capacity expansion if the DG planner seeks a reliable electricity distribution
240 service. However, the expanded grid capacity is only valuable during peak-load hours. During off-peak hours, climate
311 If DG planners are unwilling to accept the high rate of asset redundancy associated with expanded grid capacity, consumers
312 must face a heightened risk of blackouts during peak-load periods, such as extreme heat days requiring increased cooling.

Figure R71: The revised statement1.

- Original Statement: “climate change’s impacts on ED services fundamentally change our society.”

Revision: We refined this statement to provide more specificity and context. The revised version reads: “Beyond GDP growth, we also examined the GCC impact considering the energy transition, which involves the adoption of various distribution-side technologies that fundamentally reshape DG.” This revision focuses on the tangible factors influencing DG, such as energy transition and technological changes, providing a more precise and evidence-based statement.

277 In conclusion, climate change’s impacts on ED services fundamentally change our society.
280 Beyond GDP growth, we also examined the GCC impact considering the energy transition, which involves the adoption
281 of various distribution-side technologies that fundamentally reshape DG. These technologies have varying effects on DG

Figure R72: The revised statement2.

- *Improved Introduction and Definition of Key Terms:* To enhance readability and ensure clarity, we have revised the manuscript to introduce key terms such as “climate change,” “macro and micro impact,” “risk,” “vulnerability,” “hazard,” and “size assessment” earlier and in a more structured way. These terms are crucial for the integrity of the manuscript and are standard within the field of climate risk assessment, derived from widely recognized literature (e.g., [R17]). We now provide clear explanations and

definitions when first introducing these terms, ensuring that readers unfamiliar with the terminology can understand their meaning and relevance in context.

- Global climate change

Definition: Global Climate Change refers to the long-term, significant changes in the Earth's climate systems, particularly changes in temperature, precipitation patterns, and extreme weather events.

^aGlobal Climate Change is the long-term, significant changes in the Earth's climate systems, particularly changes in temperature, precipitation patterns, and extreme weather events.

Figure R73: Added a footnote to the first occurrence of Global climate change in the paper.

- Risk

Definition: Risk is the probability of an adverse outcome, such as a power blackout, occurring due to the interaction between climate change and the distribution grid system.

^bRisk is the probability of an adverse outcome, such as a power blackout, occurring due to the interaction between climate change and the distribution grid system.

Figure R74: Added a footnote to the first occurrence of Risk in the paper.

- Hazard

Definition: Hazard refers to the potential physical events or phenomena that could cause harm or disrupt the power distribution system.

is visualized using a centered flower-style plot that summarizes the hazard-vulnerability-response model¹². Hazard refers to the potential physical events or phenomena that could cause harm or disrupt the power distribution system. Vulnerability is the degree to which a

Figure R75: Added a footnote to the first occurrence of Hazard in the paper.

- Vulnerability

Definition: Vulnerability is the degree to which a power distribution system and its consumers are susceptible to the adverse impacts of climate change.

physical events or phenomena that could cause harm or disrupt the power distribution system. Vulnerability is the degree to which a power distribution system and its consumers are susceptible to the adverse impacts of climate change. Response refers to how the power

Figure R76: Added a footnote to the first occurrence of Vulnerability in the paper.

- Response

Definition: Response refers to how the power distribution grid reacts to climate-induced hazards.

power distribution system and its consumers are susceptible to the adverse impacts of climate change. Response refers to how the power distribution grid reacts to climate-induced hazards. Step 1: Modeling the hazard of GCC-driven local temperature changes. Step 2: Scope

Figure R77: Added a footnote to the first occurrence of Response in the paper.

- Macro and Micro perspectives

Definition: Micro-perspective analyzes the localized impact of climate change on specific DG, focusing on individual moments, nodes, and influencing factors. Macro-perspective analyzes the large-scale geographic impact of climate change across entire regions or countries.

^cMicro-perspective analyzes the localized impact of climate change on specific DG, focusing on individual moments, nodes, and influencing factors.
^dMacro-perspective analyzes the large-scale geographic impact of climate change across entire regions or countries.

Figure R78: Added a footnote to the first occurrence of Macro and Micro perspectives in the paper.

We hope that these changes, including the removal of unnecessary jargon, revision of normative statements, and clarification of essential terms, have made the manuscript more accessible while maintaining its academic rigor. We greatly appreciate your feedback, as it has been instrumental in helping us improve the clarity and reach of our work for a broader scientific audience.

Comment R2.20: *The manuscript distances itself with the rich existing literature on the topic. Incorporating the existing literature, e.g., [R4, R8, R10, R12, R13, R18, R21], can help clarify the critical knowledge gap(s), minimize normative assertions, and better inform the conceptual design.*

Response to R2.20: Thank you for providing these relevant references. We appreciate the suggestions and have carefully reviewed each of them. Below, we outline the commonalities and distinctions between our work and the cited studies.

- [R4] Bartos and Chester (2015) explored how climate change impacts electricity supply in the Western United States, specifically addressing risks related to water resources for power generation. While our study focuses on DG reliability, Bartos and Chester’s findings highlight the broader context of power system vulnerability, particularly the interdependencies between climate factors and power generation resources. Our work complements theirs by extending the analysis to how climate change affects DG vulnerability and blackout risks.
- [R8] Cox et al. (2017) addressed the transformative role of renewable energy and energy efficiency in bridging climate resilience and mitigation. Our study builds on this by analyzing how climate change impacts the reliability of DGs, adding a focus on climate-induced vulnerability at the local level, complementing their broader view of renewable mitigation strategies.
- [R10] Dumas et al. (2019) assessed grid vulnerabilities to extreme weather events and the gaps in quantitative methods for evaluating climate impacts on grid components. Our study similarly identifies vulnerabilities in DG systems but emphasizes the need for a systematic framework that incorporates both direct and indirect factors (e.g., economic and behavioral changes) and their uncertainties.
- [R12] Jacobson et al. (2017) argued for the viability of a 100% renewable energy system, countering claims that such a transition would destabilize the grid. While Jacobson et al. focused on the stability of the grid under a fully renewable energy system, our work adds a layer of analysis by considering how regional grid vulnerabilities evolve under different penetration levels of PV and EV technologies. We examine the implications of varying levels of renewable energy penetration, much like Jacobson et al., but in the specific context of DG resilience and regional impacts.

- [R13] Jaglom et al. (2014) explored how temperature changes due to climate change affect electricity demand and grid costs. Their findings, particularly regarding the projected 14% increase in electricity production costs by 2050 without mitigation efforts, provide a basis for our work on how temperature sensitivity and demand fluctuations impact DG reliability.
- [R18] Stephens et al. (2013) examined the role of smart grid (SG) technologies in mitigating and adapting to climate change. Our study shares common ground with theirs, particularly in terms of how SG measures can enhance grid flexibility and resilience.
- [R21] Ward (2013) provided a comprehensive review of weather events and their impacts on grid systems in Europe and North America. While Ward’ s study focused more on transmission grid systems, our work fills an important gap by concentrating on distribution grids, particularly those that serve residential areas. This complements Ward’ s analysis by offering a more localized perspective on grid vulnerability to extreme weather, an issue that is increasingly important as climate change intensifies.

In summary, our study builds upon the foundational work in the existing literature but adds new insights by focusing on the regional vulnerabilities of distribution grids, particularly in the context of climate-induced risks. Additionally, we have incorporated all these references into the manuscript, as illustrated in Figure R79.

24 **The global climate change (GCC) polarizes the electricity supply^{1,2} and demand, which challenges the reliable electricity**

1. Bartos, M. D. and Chester, M. V. (2015). Impacts of climate change on electric power supply in the Western United States. *Nature Climate Change* 5, 748–752. <https://doi.org/10.1038/nclimate2648>.
2. Cox, S. L., Hotchkiss, E. L., Bilello, D. E., Watson, A. C., and Holm, A. (2017). *Bridging Climate Change Resilience and Mitigation in the Electricity Sector Through Renewable Energy and Energy Efficiency: Emerging Climate Change and Development Topics for Energy Sector Transformation*. Tech. rep. National Renewable Energy Lab. (NREL), Golden, CO (United States). <https://doi.org/10.2172/1411521>.

(a) Bartos and Chester (2015) study the impacts of climate change on electric power supply, particularly how climate change can stress water resources for power generation. This supports the statement that GCC polarizes the electricity supply.

**is one of the most critical infrastructures supporting modern civilization, e.g., for residential customers. The reliability and**
**cost of the electricity distribution service significantly influence regional economic advantage and life quality. And, every**
**blackout in DG causes a severe economic loss⁴. When climate change exacerbates extreme cooling and heating weather, it**

4. Jaglom, W. S., McFarland, J. R., Colley, M. F., et al. (2014). Assessment of projected temperature impacts from climate change on the U.S. electric power sector using the Integrated Planning Model. *Energy Policy* 73, 524–539. <https://doi.org/10.1016/j.enpol.2014.04.032>.

(b) Cox et al. (2017) explore how renewable energy and energy efficiency play a role in mitigating the effects of climate change on energy systems, supporting the notion that GCC influences electricity supply.

**However, DG does not have the redundancy and protection that the transmission grid has^{7,8}. Such a fact makes DG**

8. Ward, D. M. (2013). The effect of weather on grid systems and the reliability of electricity supply. *Climatic Change* 121, 103–113. <https://doi.org/10.1007/s10584-013-0916-z>.

(c) Jaglom et al. (2014) predict that electricity production costs could increase significantly by 2050 due to temperature changes caused by GCC, supporting the claim that electricity distribution service impacts regional economic advantage.

**during the peak-demand hours in summers^{10–12}. One simple solution is to upgrade the distribution grids. However, it is**
**costly to upgrade all the distribution systems worldwide, giving the scale of the distribution systems for business, industrial,**
**and residential users¹³. Moreover, the higher demand in peak hours and the increasing cooling-degree days will elevate the**

13. Stephens, J. C., Wilson, E. J., Peterson, T. R., and Meadowcroft, J. (2013). Getting Smart? Climate Change and the Electric Grid. *Challenges* 4, Article 2. <https://doi.org/10.3390/challe4020201>.

(d) Ward (2013) reviews weather events' impacts on grid systems in Europe and North America, emphasizing the vulnerabilities of electricity grids, particularly transmission systems, which support the statement about DG lacking redundancy and protection.

**Other forms of physical damage have also been considered in this context^{16–19}. The second approach addresses the issue of**

19. Dumas, M., Kc, B., and Cunliff, C. I. (2019). *Extreme Weather and Climate Vulnerabilities of the Electric Grid: A Summary of Environmental Sensitivity Quantification Methods*. Tech. rep. Oak Ridge National Lab. (ORNL), Oak Ridge, TN (United States). <https://doi.org/10.2172/1558514>.

(e) Stephens et al. (2013) discuss the challenges and costs of upgrading global distribution systems to accommodate the impacts of GCC, supporting the argument that upgrading DG is costly due to its scale.

**placing renewable energy sources can postpone the need for grid upgrades in response to climate change. This aligns**
**with the argument by [28], who demonstrated the feasibility of a 100% renewable energy system while maintaining grid**
**stability. However, our analysis goes further by considering how regional grid vulnerabilities evolve with varying levels of**
**renewable energy penetration, such as PV and EV technologies. Moreover, if a grid upgrade is required, adding a single**

28. Jacobson, M. Z., Delucchi, M. A., Cameron, M. A., and Frew, B. A. (2017). The United States can keep the grid stable at low cost with 100% clean, renewable energy in all sectors despite inaccurate claims. *Proceedings of the National Academy of Sciences* 114, E5021–E5023. <https://doi.org/10.1073/pnas.1708069114>.

(f) Jacobson et al. (2017) argue for the viability of a 100% renewable energy system, and while their focus is on stability, this research supports the discussion of regional DG vulnerabilities evolving under different renewable energy penetration levels.

Figure R79: Reference images supporting the key claims in the text. Each subfigure presents evidence from the cited literature, clarifying how climate change impacts electricity supply and demand, and how DG faces challenges in a changing climate.

Comment R2.21: *Overall, the manuscript will benefit from improving the conceptual design, strengthening the evidence base for, and more accurately capturing, the conclusions, explaining the concepts upfront, relaxing the stringent assumptions regarding constant demand profiles and grid infrastructure, and incorporating relevant existing literature.*

Response to R2.21: Thank you for your constructive comments. In this revision, we have taken a thorough approach to address all the aspects mentioned, making significant improvements to the manuscript. Below, we provide details on the steps we took to enhance the conceptual design, support conclusions with evidence, clarify concepts, relax strong assumptions, and incorporate relevant literature.

1. Conceptual Design:

We agree that the conceptual design in the original manuscript could benefit from more explicit clarification of our framework’s novelty. In response to this, we have added a detailed theoretical section explaining the complex dynamics of the climate change-blackout risk relationship. Specifically, we emphasized the core concept of the blackout risk, framed within the hazard-vulnerability-response structure. This provides a clear depiction of how climate change impacts the demand and security operating boundaries (SOB) of distribution grids (DGs), leading to changes in blackout risks. We also introduced the concept of a dose-response relationship, which illustrates how climate change impacts grid vulnerability in proportion to shifts in demand and grid stability. It is shown in Figure R80.

A distribution grid has a maximum capacity for meeting nodal demands, constrained by the power flow in grid lines as
dictated by Kirchhoff’s laws. An outage occurs when the power flow through any line exceeds its capacity. Consequently,
we define the safe-operation region as the set of all nodal load profiles that prevent such overflows. The boundary of this
region is known as the SOB. The SOB is determined by the physical characteristic of the DG, including the capacities of grid

(a) Core conceptual design focusing on grid capacity under normal conditions and its relationship to climate change.

The theorem formalizes the process of determining the SOB, which represents the maximum load the distribution grid
can handle without collapsing. The SOB is a physical characteristic of the grid, determined by its infrastructure, such as
line capacities and transformer limits, and independent of specific load profiles. By analyzing the power flow equations at
each node, the boundary is derived as the point beyond which the grid can no longer sustain additional load.

(b) Theoretical basis highlighting how the SOB changes with demand and climate impacts.

$$\text{Risk}_h = \sum_{c \in \mathcal{C}} p(T_c) \cdot p(\Phi[\mathbf{P}|T_c, h] \leq 0)$$

Here, T_c represents climate change scenario c , $p(T_c)$ denotes the probability of climate change scenario c , and $\Phi[\mathbf{P}|T_c, h]$
determines the margin of the demand profile \mathbf{P} under temperature T_c at hour h to the SOB. If $\Phi \leq 0$, the grid remains within
safe operating conditions. Otherwise, a blackout will occur. We refer to this method as the climate-blackout response model,

(c) Introducing the dose-response relationship between climate change and grid vulnerability.

Figure R80: Revised conceptual design framework (Theoretical Foundation section).

2. Supporting Key Conclusions with Evidence:

We strengthened the evidence base for key conclusions by:

Citing Relevant Literature: We added multiple references to support our conclusions. These citations were carefully selected to strengthen the theoretical foundations and empirical observations. Below are some examples of supporting references added to the manuscript, as shown in Figure R81.

24 **The global climate change (GCC) polarizes the electricity supply^{1,2} and demand, which challenges the reliable electricity**

1. Bartos, M. D. and Chester, M. V. (2015). Impacts of climate change on electric power supply in the Western United States. *Nature Climate Change* 5, 748–752. <https://doi.org/10.1038/nclimate2648>.
2. Cox, S. L., Hotchkiss, E. L., Bilello, D. E., Watson, A. C., and Holm, A. (2017). *Bridging Climate Change Resilience and Mitigation in the Electricity Sector Through Renewable Energy and Energy Efficiency: Emerging Climate Change and Development Topics for Energy Sector Transformation*. Tech. rep. National Renewable Energy Lab. (NREL), Golden, CO (United States). <https://doi.org/10.2172/1411521>.

(a) Bartos and Chester (2015) studying climate change impacts on electricity supply, providing evidence of the polarization of electricity supply due to GCC.

26 **is one of the most critical infrastructures supporting modern civilization, e.g., for residential customers. The reliability and**
27 **cost of the electricity distribution service significantly influence regional economic advantage and life quality. And, every**
28 **blackout in DG causes a severe economic loss⁴. When climate change exacerbates extreme cooling and heating weather, it**

4. Jaglom, W. S., McFarland, J. R., Colley, M. F., et al. (2014). Assessment of projected temperature impacts from climate change on the U.S. electric power sector using the Integrated Planning Model. *Energy Policy* 73, 524–539. <https://doi.org/10.1016/j.enpol.2014.04.032>.

(b) Cox et al. (2017) discussing renewable energy’s role in mitigating climate impacts, supporting our framework’ s climate-energy interaction analysis.

Figure R81: Reference images supporting the manuscript’s key claims.

Enhanced Figures: We revised and added figures to visually support our conclusions. These figures further elucidate the key points, ensuring that conclusions are not only theoretically but also visually supported by the analysis. For example, it is shown in Figure R82.

3. Improving Clarity and Readability:

We made several modifications to improve the manuscript’s clarity and structure:

Reorganized Paper Structure: We revised the section titles and overall structure for logical flow, ensuring that the paper is easier to follow. It is shown in Figure R83.

Removed or Simplified Complex Terms: We simplified complex jargon to make the manuscript more accessible to a broader audience. For instance:

- *Electricity distribution (ED)*: We removed the abbreviation ”ED” to avoid overloading the manuscript with too many acronyms.
- *Economic competitive advantages*: We removed this term as it was not fully substantiated by the analysis.

Figure R82: Revised framework showing the interaction between GDP growth, energy transition, and grid infrastructure.

INTRODUCTION	3
RESULTS	4
Framework for assessing the GCC's impacts on ED service	4
Macro impact: energy advantage reshuffle in the U.S. and Europe	5
Climate change's impact on the size of reliable DG system	5
Climate change redefines the economic competitive advantages from the electricity perspective	7
Micro impact: notably threat on ED's reliability	7
DISCUSSION	9
	
Introduction	3
A Framework for Accessing GCC Impact onto Distribution Grids	4
Theoretical Foundation	4
A Framework	6
Micro perspective: Magnificent Threat from GCC to Distribution Grids	7
GCC's Pure Effect	7
GCC's Combined Effect Intervened by Changes of Other Factors	9
Macro perspective: Geographically-Heterogeneous Stresses from GCC to Regional Distribution Services	11
Distinct GCC Outcomes Across Diversified Regions	11
U.S. Results	11
EU Results	12
Why is There a Divergence?	13
Discussion: Efficiently Adapt the GCC	13
Conclusion	15

Figure R83: The revised titles of sections and subsections to improve clarity and logical flow.

- *Difference-in-difference (DID)*: Simplified as "analyzed vulnerable nodes under three scenarios" to ensure better readability.

4. Relaxing Strong Assumptions:

We carefully addressed the feedback on strong assumptions, particularly:

Constant User Temperature Sensitivity: In the previous version, this was assumed constant. In the revision, we considered the variability of temperature sensitivity across regions and over time, introducing scenario analyses that reflect changes due to economic growth. It is shown in Figure R84.

**Modeling the Impact of Economic Growth and Climate on Energy Demand and DG System Ex-**
**pansion**

In this model, the impact of economic growth on electricity demand is incorporated through the per-capita GDP adjusted
growth factor. As the economy grows, it typically drives higher energy demand due to increased industrial activity, com-
mercial growth, and residential usage. To capture this effect, we introduce GDP growth as a key variable in the calculation
of the annual demand mean. By combining GDP with heating degree days (HDD, $T < 12.5^\circ\text{C}$) and cooling degree days
(CDD, $T > 27.5^\circ\text{C}$), the model comprehensively accounts for both economic and climate influences on the energy system.
The coefficients used for each factor are based on the values provided by Ref. [32].

The annual demand mean for each year is calculated as follows:

$$\text{dmean}(y) = \text{gdp}_b \cdot \text{GDP}(y) + \text{hdd}_b \cdot \text{HDD}(y) + \text{cdd}_b \cdot \text{CDD}(y), \quad (8)$$

where the coefficients are $\text{gdp}_b = 0.3763$, reflecting the impact of economic growth (GDP) on energy demand. $\text{cdd}_b =$
0.0103 , representing the influence of HDD on energy consumption. $\text{hdd}_b = 0$, indicating no significant effect from CDD in
the reference model.

Next, based on the historical data (2015-2019), we derive the U-curve, which captures how consumers' electricity
demand responds to temperature variations. The U-curve illustrates how demand changes with temperature, with higher
demand during extreme heat or cold, and lower demand at moderate temperatures.

The hourly demand for each future year y is calculated by adjusting the 2019 baseline demand with a scaling factor
based on the U-curve and the annual demand gap, as follows:

$$\text{demand}(y, i) = \text{multiple} \times (\text{dmean}(y) - \text{dmean}(2019)) + \text{U_curve}(\text{temp}, i), \quad (9)$$

where *multiple* is calculated as the ratio of the demand mean derived from the U-curve in 2019, $\text{dmean_U}(2019)$, to the
baseline demand mean for 2019, $\text{dmean}(2019)$. The DG system expansion for year y is proportional to the growth in annual
demand relative to 1950, ensuring the system scales with increasing electricity demand.

Figure R84: Modeling the impact of economic growth on temperature sensitivity.

Constant Distribution Grid Infrastructure: We removed the assumption of constant infrastructure. The revised manuscript now discusses how infrastructure evolves in response to population and economic growth, as well as the integration of renewable energy technologies. It is shown in Figure R85.

5. Incorporating Relevant Existing Literature:

We incorporated all the suggested references, bridging the gaps with existing studies. For example:

- [R4] Bartos and Chester's (2015) analysis on water resources for power generation complements our study on how climate change impacts electricity distribution.
- [R12] Jacobson et al. (2017) discussed the viability of 100% renewable energy systems, which supports our analysis of grid resilience under varying renewable energy penetration levels.

Beyond GDP growth, we also examined the GCC impact considering the energy transition, which involves the adoption
 of various distribution-side technologies that fundamentally reshape DG. These technologies have varying effects on DG
 reliability. For example, the Paris Agreement has catalyzed the development of Photovoltaics (PV) and Electric Vehicles
 (EVs)³⁸. Increased PV penetration enhances DG reliability by mitigating the power distracting from the nodes, which helps
 mitigate the risk of blackouts. However, the growth of EVs introduces higher loads and thus can amplify the GCC's impact.
 Therefore, the pace and mix of different technologies will critically determine how energy transitions affect DG blackout
 risk under GCC.

We assess how different technologies affect the impact of climate change on the blackout risk of California's primary
 distribution Feeder 1. As shown in Figure 3(e), a 20% penetration of photovoltaic (PV) systems can reduce the blackout risk of
 distributed generation by 0.72% in 2050. This reduction is sufficient to fully offset the climate change impact under the
 RCP 4.5 scenario. Even in the more severe RCP 8.5 scenario, a 20% PV penetration still significantly mitigates the increased
 blackout risk caused by climate change. In contrast, Figure 3(f) indicates that a 20% penetration of electric vehicles (EVs)
 in the same area will increase the DG blackout risk by approximately 1.19% in 2050. This increase corresponds to a higher
 load on the distribution network nodes as EV penetration rises. Moreover, this effect is compounded by typical EV charging
 behaviors, as shown by a statistical analysis of EV usage patterns³⁹.

In the real world, the GCC, GDP growth, and energy transition will simultaneously impact the DG's blackout risk. We
 examined the impact of the energy transition on GCC's effects in the SSP 1 scenario. We generated three energy transition
 scenarios for the next two decades: the PV-only scenario, the EV-only scenario, and the scenario where the penetrations
 of PV and EV simultaneously grow. The penetration growth rates are set according to the Los Angeles 100% Renewable
 Energy(LA100) study conducted by the National Renewable Energy Laboratory (NREL)⁴⁰. Our results in Figure 3(c)
 suggest that the PV penetration rate, as projected in the LA100 study, is roughly sufficient to offset GCC's impact if EV
 penetration is absent. However, if EV penetration grows according to the LA100 projections, Figure 3(d) shows that the
 mitigating effect of PV will be offset. Consequently, the GCC's effect will still drive the DG blackout risk to substantially
 grow.

Figure R85: Discussion of energy transition with growing PV/EV integration.

We believe these revisions comprehensively address the reviewer's feedback and have significantly improved the manuscript. We look forward to your further comments.

Response to Reviewer 3

Comment R3.1: *I found the work to be extremely interesting and closely aligned with current research themes.*

Response to R3.1: Thanks for the feedback that our paper is extremely interesting. We also appreciate the reviewer's comment that our paper is closely aligned with current research themes.

The paper is based on several existing projects from Department of Energy, National Science Foundation, and diversified utility data. From the industrial feedback and sponsor's comments, we found great values for the topic. When writing the paper, we discovered more foundational ideas. We will use this revision to answer your comments.

Comment R3.2: *Regarding the abstract, I would suggest further emphasizing the real innovation of the presented work.*

Response to R3.2: Thanks for the suggestion. We agree with the reviewer that we shall highlight the real innovation of the presented work. Therefore, we revised the abstract for several times and asked experts in the climate change domain to read and give us feedback. In this revision, we use the following changes to emphasizing the real innovations.

- We specifically added the limitation from existing methods. For example, we added “Existing studies lack a quantitative and analytical framework to assess blackout risks and devise effective adaptation strategies to mitigate GCC’s adverse impacts.”
- We explain why the past work has such a limitation. We added “due to the absence of model converting the temperature change into variations in blackout risk.”
- We also clearly state how we address the gap. For example, we added ”To address this gap, we propose a theoretical framework for assessing the vulnerability of arbitrary distribution grids to temperature-induced changes from GCC and for comparing the GCC-driven stress on reliable electricity service.”
- We highlight the observations that only our nonlinear model based on theoretical result can show. Specifically, We rewrite the special observation: “We observed existing GCC threats and intensifying trend on the reliability of real-world distribution grids, potentially increasing blackout risks during peak-load hours by 4% to 6%, depending on GDP growth rates. Our findings reveal that over 20% of U.S. states will require a minimum increase of over 10% in distributed generation (DG) capacity before 2050, with six states needing more than a 20% increase. This suggests that DG capacity in vulnerable regions may need to double by the end of the century. ”
- We also show a contrast to show our special view between two continents. For example, we added “In contrast, the impact of GCC on power distribution in Europe is significantly moderate.
- We add our unique recommendation for policy makers. For example, we added “ Our results recommend that policymakers focus on managing peak loads and addressing nonlinear risk trends across different geographic areas.”

- We also show engineering solution from our work. For example, we added “Furthermore, our study provides new insights for cost-effective grid upgrades, strategic placement of distributed energy resources, and matching temperature-sensitive users with robust buses to enhance resilience.”

The original abstract and the revised abstract are shown in Figure R86 and R87, respectively.

8 **Abstract**

9 Climate change poses a dual challenge to electricity distribution (ED) services, with macro impacts reshaping the geographical distribution of service difficulty and micro impacts changing the blackout risk of current distribution grid (DG) systems. We propose a systematic framework for assessing global climate change(GCC)’s impact on ED service. We overcome the challenges of estimating DG’s safe operation boundary and its change associated with temperature, which enables us to comprehensively assess the GCC’s macro and micro impact. Our research indicates that by 2050, GCC will markedly reshape the U.S. energy landscape, leading to significant impacts on the competitiveness of businesses and industries. These effects are expected to be more immediate and severe compared to the situation in Europe. Over 20% of U.S. states require a minimal reliable DG increase of over 10% before 2050, and six states experience a growth of more than 20%, suggesting a potential need for double-sized DG systems in vulnerable U.S. areas by the century’s end. In contrast, GCC’s impact in Europe is much more moderate. Our empirical study also highlights the micro impact of GCC on ED services, notably increasing blackout risks in crucial cities of venerable American states. DG’s exposure hours to high-risk levels primarily grew at a rate of about 30% from the 2000s to the 2050s. Our study on GCC’s micro impact on ED further provides the new insight into identifying critical control points to mitigate risks in the DG system. It is more economically efficient for DG’s adaptation to climate change by including more mesh structures than expanding the grid capacity of the current tree network. Moreover, suggesting a strategic match between temperature-sensitive users and robust nodes emerges as a novel approach to enhance the climate robustness of DG systems.

Figure R86: The original abstract

9 **Abstract**

10 Global climate change (GCC) changes temperatures, reshaping blackout risks in power distribution grids by gradually polarizing electricity demand. Existing studies lack a quantitative and analytical framework to assess blackout risks and devise effective adaptation strategies to mitigate GCC’s adverse impacts, due to the absence of model converting the temperature change into variations in blackout risk. To address this gap, we propose a theoretical framework for assessing the vulnerability of arbitrary distribution grids to temperature-induced changes from GCC and for comparing the GCC-driven stress on reliable electricity service. We observed existing GCC threats and intensifying trend on the reliability of real-world distribution grids, potentially increasing blackout risks during peak-load hours by 4% to 6%, depending on GDP growth rates. Our findings reveal that over 20% of U.S. will require a minimum increase of over 10% in distributed generation (DG) capacity before 2050, with six states needing more than a 20% increase. This suggests that DG capacity in vulnerable regions may need to double by the end of the century. In contrast, the impact of GCC on power distribution in Europe is significantly moderate. Our results recommend that policymakers focus on managing peak loads and addressing nonlinear risk trends across different geographic areas. Furthermore, our study provides new insights for cost-effective grid upgrades, strategic placement of distributed energy resources, and matching temperature-sensitive users with robust buses to enhance resilience.

Figure R87: The revised abstract

Global climate change (GCC) changes temperatures, reshaping blackout risks in power distribution grids by gradually polarizing electricity demand. Existing studies lack a quantitative and analytical framework to assess blackout risks and devise effective adaptation strategies to mitigate GCC’s adverse impacts, due to the absence of model converting the temperature change into variations in blackout risk. To address this gap, we propose a theoretic-

cal framework for assessing the vulnerability of arbitrary distribution grids to temperature-induced changes from GCC and for comparing the GCC-driven stress on reliable electricity service. We observed existing GCC threats and intensifying trend on the reliability of real-world distribution grids, potentially increasing blackout risks during peak-load hours by 4% to 6%, depending on GDP growth rates. Our findings reveal that over 20% of U.S. states will require a minimum increase of over 10% in distributed generation (DG) capacity before 2050, with six states needing more than a 20% increase. This suggests that DG capacity in vulnerable regions may need to double by the end of the century. In contrast, the impact of GCC on power distribution in Europe is significantly moderate. Our results recommend that policymakers focus on managing peak loads and addressing nonlinear risk trends across different geographic areas. Furthermore, our study provides new insights for cost-effective grid upgrades, strategic placement of distributed energy resources, and matching temperature-sensitive users with robust buses to enhance resilience.

Comment R3.3: *In the introduction, it is essential to focus on the gap identified in the literature, explaining the need to address this issue and outlining how your work deviates from existing research.*

Response to R3.3: Thank you for your valuable feedback regarding the structure of the introduction. We appreciate your suggestion to focus on the literature gap before introducing how our work addresses it. Following your recommendation, we have rewritten the introduction to first outline the key gaps in the existing literature and then explain how our research addresses these gaps. This revised structure provides a clearer explanation of the motivation behind our work and more effectively highlights our contributions.

The existing literature has highlighted the impacts of global climate change (GCC) on power systems, particularly focusing on the vulnerabilities caused by extreme weather events and increasing demand for electricity. However, a critical gap remains in understanding the specific vulnerabilities of distribution grids (DGs) in response to GCC. While transmission grids have been thoroughly studied, DGs—which serve residential and commercial areas—lack redundancy and are more susceptible to failure during extreme weather conditions. The vulnerability of DGs is further exacerbated by increasing demand, particularly during peak hours in both hot and cold climates. This is an area that requires urgent attention, as the global transition to electric vehicles and distributed energy generation places additional strain on DG systems.

Existing research has primarily examined two aspects of GCC’s impact on DGs: infrastructure damage caused by extreme weather events and insufficient capacity to meet rising demand. However, there has been little focus on systematically analyzing how GCC alters the demand profiles across different regions and timeframes. Additionally, there is no rigorous method to model the complex relationship between GCC-induced temperature changes and the subsequent increase in blackout risk for DG systems.

To address this gap, our work introduces a novel framework that integrates climate, economic, and engineering perspectives to assess the impacts of GCC on DGs. Unlike previous studies that focus mainly on transmission grids or isolated DG components, our research provides a comprehensive analysis of how GCC affects the reliability of DGs at both macro (geographic) and micro (node-level) scales. Our framework allows for a detailed examination of how climate change-induced factors, such as increased temperature variability, affect the vulnerability of DGs, providing a systematic approach for future grid resilience planning. We also incorporate real-world data and consider varying levels of renewable energy penetration,

such as photovoltaic (PV) and electric vehicle (EV) technologies, to understand their effects on DG vulnerability. This deviation from existing research allows us to propose tailored solutions, such as strategic placement of renewable energy sources and targeted grid upgrades to enhance DG resilience.

By addressing the identified gap in the literature, our research contributes to a deeper understanding of the specific risks faced by DGs in the face of GCC, ultimately providing practical insights for policymakers and engineers tasked with ensuring reliable electricity distribution in a rapidly changing climate.

The updated introduction is shown in Figure R88.

However, DG does not have the redundancy and protection that the transmission grid has^{7,8}. Such a fact makes DG
more vulnerable to temperature fluctuations and changes in demand profiles. For example, [9] pointed out that 90% of the
blackouts in the U.S. are caused by the breakdown of the DGs. Therefore, GCC will fundamentally alter consumer demand
patterns, exacerbating the vulnerability of the DG⁶. For instance, climate change is projected to increase the frequency of
cooling-degree days, intensifying peak demand hours¹⁰. Several studies predict a more frequent surge of power demand
during the peak-demand hours in summers^{10–12}. One simple solution is to upgrade the distribution grids. However, it is
costly to upgrade all the distribution systems worldwide, giving the scale of the distribution systems for business, industrial,
and residential users¹³. Moreover, the higher demand in peak hours and the increasing cooling-degree days will elevate the
probability of overloading, which is the primary source of electricity blackout risk¹⁴. The anticipated rise in temperatures
due to climate change will reduce the capacity of distribution grids, further elevating the risk of overloading¹⁵. Consequently,
climate change can exacerbate the blackout risk of the existing DGs.

To address this, a rigorous approach is needed to analyze how GCC amplifies the vulnerability of distribution grids,
mainly through temperature changes. Existing studies have primarily examined GCC threats to DGs in two ways. The first
focuses on the damage to infrastructure caused by extreme weather events. For instance, [16, 17] have highlighted that
current DG facilities, such as grid lines and transformers, are highly vulnerable to damage from extreme weather events.
Other forms of physical damage have also been considered in this context^{16–19}. The second approach addresses the issue of
insufficient capacity, which occurs when the total electricity demand exceeds substation capacity^{6,12,20}. To analyze changes
in load, [20–22] have proposed data-driven methods to estimate how electricity demand responds to local temperature
changes in the transmission grid.

While increased load is an indicator of grid vulnerability, existing studies lack analytical tools to evaluate this vulner-
ability across different parts of a grid^{6,23}. For instance, previous research has struggled to determine the safe operational
boundaries of distribution grids due to the highly non-linear variable coupling and non-convex nature of the optimization
problem¹⁵. Additionally, the relationship between GCC and blackout risk in distributed generation systems involves a com-
plex, long-chain climate-economic-engineering process²⁴, making it challenging to estimate real-world DG vulnerability
based solely on temperature changes. Consequently, it is difficult to conduct scenario studies for GCC with DG at different
nodes within a feeder or across different distribution grids over a large geographic area. This limitation prevents the proper
modeling of a dose-response function that captures the relationship between climate change and DG blackout risk, which
is essential for effective risk assessment and management^{21,25}.

To address these challenges, we present a framework that examines hazards, vulnerabilities, and responses through a
systematic four-step process. This framework is built on our research, which analytically derives the relationship between
power allocations and associated risk regions^{15,26,27}. Consequently, our approach integrates power flow analysis into climate
change assessments, providing performance guarantees. When applied to realistic data, our framework can quantitatively
assess the impact of global climate change on various parts of a grid (micro-perspective) and across different geographic
areas (macro-perspective). Its scalable design ensures computational efficiency. For instance, in the U.S. and Europe, our
tool indicates the necessity of considering peak hours, temperature trends, and economic growth together when evaluating
grid performance. The impacts of climate change are both immediate and severe. Our findings suggest that strategically
placing renewable energy sources can postpone the need for grid upgrades in response to climate change. This aligns
with the argument by [28], who demonstrated the feasibility of a 100% renewable energy system while maintaining grid
stability. However, our analysis goes further by considering how regional grid vulnerabilities evolve with varying levels of
renewable energy penetration, such as PV and EV technologies. Moreover, if a grid upgrade is required, adding a single
line to a distributed generation network can reduce vulnerability by up to 40%. This solution is particularly effective due
to the relatively short distances between DG nodes compared to the broader transmission grid.

Figure R88: The rewritten Introduction

Comment R3.4: *I suggest implementing the literature review.*

Response to R3.4: Thank you for your suggestion regarding the implementation of a more thorough literature review. We appreciate your feedback and have revised the manuscript accordingly. In the updated version, we have expanded the literature review to better highlight existing research in the field and identify the critical gaps that our work aims to address.

The expanded literature review now covers a broader range of studies on the vulnerabilities of DGs to climate change, focusing on the impact of extreme weather events, infrastructure resilience, and energy demand shifts due to GCC. Additionally, we have included key studies that explore the integration of renewable energy sources, EV, and distributed energy resources (DER) in relation to grid reliability and the broader implications for DG resilience. By synthesizing these studies, we have framed our contribution more clearly within the context of existing research.

The expanded literature review is shown in Figure R89, and we believe this revision enhances the depth and comprehensiveness of our study.

However, DG does not have the redundancy and protection that the transmission grid has^{7,8}. Such a fact makes DG
more vulnerable to temperature fluctuations and changes in demand profiles. For example, [9] pointed out that 90% of the
blackouts in the U.S. are caused by the breakdown of the DGs. Therefore, GCC will fundamentally alter consumer demand
patterns, exacerbating the vulnerability of the DG⁶. For instance, climate change is projected to increase the frequency of
cooling-degree days, intensifying peak demand hours¹⁰. Several studies predict a more frequent surge of power demand
during the peak-demand hours in summers^{10–12}. One simple solution is to upgrade the distribution grids. However, it is
costly to upgrade all the distribution systems worldwide, giving the scale of the distribution systems for business, industrial,
and residential users¹³. Moreover, the higher demand in peak hours and the increasing cooling-degree days will elevate the
probability of overloading, which is the primary source of electricity blackout risk¹⁴. The anticipated rise in temperatures
due to climate change will reduce the capacity of distribution grids, further elevating the risk of overloading¹⁵. Consequently,
climate change can exacerbate the blackout risk of the existing DGs.

To address this, a rigorous approach is needed to analyze how GCC amplifies the vulnerability of distribution grids,
mainly through temperature changes. Existing studies have primarily examined GCC threats to DGs in two ways. The first
focuses on the damage to infrastructure caused by extreme weather events. For instance, [16, 17] have highlighted that
current DG facilities, such as grid lines and transformers, are highly vulnerable to damage from extreme weather events.
Other forms of physical damage have also been considered in this context^{16–19}. The second approach addresses the issue of
insufficient capacity, which occurs when the total electricity demand exceeds substation capacity^{6,12,20}. To analyze changes
in load, [20–22] have proposed data-driven methods to estimate how electricity demand responds to local temperature
changes in the transmission grid.

While increased load is an indicator of grid vulnerability, existing studies lack analytical tools to evaluate this vulner-
ability across different parts of a grid^{6,23}. For instance, previous research has struggled to determine the safe operational
boundaries of distribution grids due to the highly non-linear variable coupling and non-convex nature of the optimization
problem¹⁵. Additionally, the relationship between GCC and blackout risk in distributed generation systems involves a com-
plex, long-chain climate-economic-engineering process²⁴, making it challenging to estimate real-world DG vulnerability
based solely on temperature changes. Consequently, it is difficult to conduct scenario studies for GCC with DG at different
nodes within a feeder or across different distribution grids over a large geographic area. This limitation prevents the proper
modeling of a dose-response function that captures the relationship between climate change and DG blackout risk, which
is essential for effective risk assessment and management^{21,25}.

Figure R89: The expanded literature review

Comment R3.5: *I noticed that Figure 1, although well-executed, is content-dense and sometimes challenging to read. I believe it would be appropriate to provide a more detailed description of the steps presented in Figure 1 to enhance accessibility.*

Response to R3.5: Thank you for your valuable feedback on Figure 1. We acknowledge that the figure was content-dense and challenging to read. In response, we have redesigned Figure 1 to enhance its clarity and accessibility. Below are the specific improvements made:

- We removed the confusing blue lines to simplify the visual structure.
- The steps in the process are now strictly aligned from left to right and top to bottom to ensure a clear, logical flow.
- We improved the alignment of different components and ensured consistency in the shapes used across the diagram for better readability.
- We defined the TB3R (Temperature-Blackout Risk Response relation) to explain the relationship between temperature and blackout risk more clearly.
- We removed Step 4, "Characterize the climate change's impact and define the index," and Step 5, "Experiment-based critical control point analysis," as these steps were redundant with the content presented in the main text.
- We reduced the number of scopes, retaining only the "Pure Effect" and "Combined Effects with the Consideration of Other Factor Change" to align with the focus of the paper.
- We incorporated specific adaptation strategies in the response section, divided into two categories: engineering solutions and policy approaches, to provide practical insights into mitigating the impacts discussed in the paper.

We believe that these modifications have significantly improved the readability and usefulness of Figure 1. And we also used bullet points to explain each step in the updated manuscript, as shown in Figure R91. The comparison is shown in Figure R90.

(a) The original Figure 1.

(d) The Systematic Framework

(b) The revised Figure 1.

Figure R90: The Comparison of Figure 1 before and after modification.

risk in distribution systems. The framework, inspired by [32] and illustrated in Figure 1(d), involves five steps and is built
on three foundational pillars: hazard, vulnerability, and response. The hazard pillar focuses on understanding the type and
magnitude of the threat induced by the GCC. By incorporating the scenario of GCC-induced hazard, we apply the TB3R
to examine clarify the climate-economic-engineering process and assess the degree of the DG's vulnerability to the GCC's
threat. The vulnerability analysis further provides valuable insights into potential engineering solutions for adaptation. Our
framework is capable of addressing various impacts of GCC on DG reliability. Below, we outline the detailed steps for
assessing the impact caused by GCC-driven temperature changes:

- • Modeling the Hazard: Model the magnitude of GCC-driven local temperature changes.
- • Scoping: Identify whether and what other progress is included when assessing the GCC's impact. When the inter-
vention of all other progresses such as GDP growth is excluded, the assessment calibrates the size of the GCC's pure
effect. Otherwise, the assessment verifies the GCC's effect intervened by the change of other factors.
- • Calculating the climate-blackout response: Apply the TB3R to various GCC scenarios to generate the climate-
blackout response, which manifests the DG's vulnerability to the GCC.
- • Assessing and comparing the GCC's impact: Calibrate the increase of the blackout risk driven by the GCC, and
compare the GCC-induced stress on reliable-electricity distribution over regions.
- • Designing the adaptation strategy: Analyze how GCC-driven temperature change raises the DG's blackout risk and
design adaptation strategies

Figure R91: The explanation of each step in the updated manuscript.

Comment R3.6: *Figure 4 is not clearly visible, and the interpretation of the graphs is very difficult. In my opinion, reconsidering the distribution of the graphs may improve overall readability.*

Response to R3.6: Thank you for your feedback on Figure 4. We agree that the original version was difficult to interpret and lacked clarity. To address this issue, we have made several improvements to the figure, as outlined below:

- We reran the simulations for a more significantly impacted consumer across two 3-bus models to enhance the visibility and clarity of the demonstration.
- The plots f, g, j, and k were revised to specifically show the collapse risk curves for mesh and tree distribution grids under different scenarios at 30-35°C. These changes provide a clearer representation of the differences between the two grid types under climate stress.
- To improve distinction between the subplots, we introduced light gray dashed lines between each plot for better separation and readability.
- In line with the writing logic, we swapped the content in the second and third rows. Now, the second row shows the results for the 3-bus tree network, and the third row shows results for the 3-bus mesh network. Columns two, three, and four represent the Benchmark, Climate Change, and Climate Change with Consumer Reallocation scenarios, respectively.
- We enlarged the font in the plots to improve legibility and simplified the x-axis and y-axis labels for plots a, b, and c to make the figure more concise.
- Finally, we changed the ovals in subplots e and i from green to pink to make them easier to observe and highlight important regions in the graph.

The comparison is shown in Figure R92 and R93.

Figure R92: Original Figure 4

Figure R93: Revised Figure 4

Comment R3.7: *Furthermore, more explanations are needed regarding the model used.*

Response to R3.7: Thank you for your valuable comment regarding the need for more explanations about the model used in our study. We have addressed this by adding a new section, titled “Theoretical Foundation,” under the chapter “A Framework for Assessing GCC Impact on Distribution Grids.” This section is displayed in Figure R95 and provides a comprehensive explanation of the model and its underlying principles.

Specifically, we introduce the process of calculating the Safe Operation Boundary (SOB) in the form of a theorem. This includes an explanation of how power flow equations are used to determine the operating limits of the grid. The theorem describes how the intersection of geometric circles in the voltage plane defines the grid’s safe region, and how the optimization problem is formulated to calculate the margin to the SOB. Additionally, the newly added Figure 1 (a) and (b) visually demonstrates these calculation principles to further enhance the reader’s understanding.

Given the broad readership of **Nature Communications**, we have opted not to include the full mathematical derivation of the equations in the main text to maintain accessibility. Instead, the detailed explanation of the formulas and their derivation is provided in the subsection “Calculating the SOB of DG system” which is shown in Figure R96 in the Methods section.

We believe these additions, including the visual aids in Figure 1 (a) and (b), will help readers better understand the theoretical foundation behind the model while keeping the manuscript clear and concise, as show in Figure R94.

Figure R94: The visual aids in Figure 1 (a) and (b), will help readers better understand the theoretical foundation

**Theoretical Foundation**

A distribution grid has a maximum capacity for meeting nodal demands, constrained by the power flow in grid lines as
 dictated by Kirchhoff's laws. An outage occurs when the power flow through any line exceeds its capacity. Consequently,
 we define the safe-operation region as the set of all nodal load profiles that prevent such overflows. The boundary of this
 region is known as the safety operation boundary (SOB). The SOB is determined by the physical characteristic of the DG,
 including the capacities of grid lines and transformers²⁹⁻³¹. The SOB is mathematically derived by analyzing the power
 flow equations for each node. We formalize this derivation in the following theorem, with a formal derivation provided in
 the following theorem and detailed further in the Method section.

 **Theorem 1.** *We can compute the margin to the SOB by solving the optimization problem 5 in Method section. This problem*
 *involves finding a direction \mathbf{z} such that the power flow is maximized while satisfying the physical constraints of the system.*
 *The power gradient \mathbf{h}_i for the voltage helps to trace the SOB. The margin to the SOB, denoted by Φ , can then be calculated.*
 *When the solution to the optimization problem yields $\Phi[\mathbf{P}] = 0$, the system is at the SOB, indicating that the grid is operating*
 *at the maximum load boundary without collapsing.*

The theorem formalizes the process of determining the SOB, which represents the maximum load the distribution grid
 can handle without collapsing. The SOB is a physical characteristic of the grid, determined by its infrastructure, such as
 line capacities and transformer limits, and independent of specific load profiles. By analyzing the power flow equations at
 each node, the boundary is derived as the point beyond which the grid can no longer sustain additional load.

Figure 1(a) shows how the grid characteristics and demand induce the SOB on the right. The power pairs are the dots in
 blue and green. The SOB is the boundary that separates the safe points in blue and the risky points in green. The risk of a
 DG blackout in any given hour h is determined by the probability that the demand profile exceeds the SOB. This blackout
 risk can be quantified as follows:

$$\text{Risk}_h = \sum_{c=1}^t p(T_c) \cdot p(\Phi[\mathbf{P}|T_c, h] \leq 0)$$

Here, T_c represents climate change scenario c , $p(T_c)$ denotes the probability of climate change scenario c , and $\Phi[\mathbf{P}|T_c, h]$
 determines the margin of the demand profile \mathbf{P} under temperature T_c at hour h to the SOB. If $\Phi \leq 0$, the grid remains within
 safe operating conditions. Otherwise, a blackout will occur. We refer to this method as the climate-blackout response model,
 which integrates the physical characteristics of the distribution grid, power flow equations, and real-time demand to assess
 and mitigate blackout risk, ensuring reliable operation under various load and temperature conditions.

GCC induces local temperature change for Figure 1(a) on the blackout risk. One value of our analysis is the analyti-
 104 cal solution for the complex climate-economic-engineering process. For example, the oval region Figure 1(a) with green
 boundary will move to the blue oval in Figure 1(b). For example, the maximal power-flow capacity of a line is also in-
 fluenced by temperature. According to the Temperature Coefficient Formula³³, the conductivity of a metal decreases with
 temperature, which implies that the power flow in a transmission line decreases as the temperature increases. Consequently,
 temperature changes the blue boundary in Figure 1(a). In Figure 1(b), we can see a shrunk version in red towards the origin
 when compared to the blue SOB region in Figure 1(a). Therefore, GCC-induced temperature change alters both the size and

Figure R95: Theoretical Foundation

**Calculating the SOB of DG system**

The safe operation boundary (SOB) of a DG system is defined as the margin to blackout at which the grid can operate safely
 without collapsing. In our analysis, the geometric circles of the power flow have nice properties for deriving the loading
 capacity bounds¹⁵. We adopt the rectangular coordinates for complex voltages. Then, the power flow equations become

$$P_i = d_{i,1} \cdot v_{i,r}^2 + d_{i,2} \cdot v_{i,r} + d_{i,1} \cdot v_{i,m}^2 + d_{i,3} \cdot v_{i,m}, \quad (2a)$$

$$Q_i = d_{i,4} \cdot v_{i,r}^2 - d_{i,3} \cdot v_{i,r} + d_{i,4} \cdot v_{i,m}^2 + d_{i,2} \cdot v_{i,m}, \quad (2b)$$

where

$$d_{i,1} = - \sum_{k \in \mathcal{N}(i)} g_{ki}, \quad d_{i,2} = \sum_{k \in \mathcal{N}(i)} (v_{k,r} g_{ki} - v_{k,m} b_{ki}), \quad (3a)$$

$$d_{i,3} = \sum_{k \in \mathcal{N}(i)} (v_{k,r} b_{ki} + v_{k,m} g_{ki}), \quad d_{i,4} = \sum_{k \in \mathcal{N}(i)} b_{ki}, \quad (3b)$$

where $\mathcal{N}(i)$ represents the neighbors of bus i , and P_i and Q_i are the active and reactive power injections at bus i . The voltage
 at bus i is represented by the complex phasor v_i . In rectangular coordinates, $v_{i,r} = |v_i| \cos \theta_i$ and $v_{i,m} = |v_i| \sin \theta_i$ represent
 the real and imaginary parts of the complex voltage at bus i , respectively. $\theta_{ik} = \theta_k - \theta_i$ denotes the phase angle difference
 between buses k and i , and g_{ki} and b_{ki} are the electrical conductance and susceptance between bus i and bus k . Together,
 $y_{ki} = g_{ki} + j \cdot b_{ki}$ forms the admittance, where j is the imaginary unit.

For fixed constants $d_{i,1}$, $d_{i,2}$, $d_{i,3}$, $d_{i,4}$, equations (2a) and (2b) describe two circles in the $v_{i,r}$ and $v_{i,m}$ space. The coordi-
 nates of the circle center E for the active power flow are $\left(-\frac{d_{i,2}}{2d_{i,1}}, -\frac{d_{i,3}}{2d_{i,1}}\right)$ for bus i , and its radius decreases as $P_i(t)$ increases.

Similarly, the center D for the reactive power flow is located at $\left(\frac{d_{i,3}}{2d_{i,4}}, -\frac{d_{i,2}}{2d_{i,4}}\right)$, and its radius decreases as $Q_i(t)$ increases.

Thus, if the active power circle and the reactive power circle do not intersect, the DG system will experience a blackout.
 This condition implies that an operating point on the SOB is tangent to both circles at a single point. Geometrically, a point
 lies on the boundary if there is no other point that can consume more power.

**Theorem 2.** *Checking whether an operating point is on the boundary of the feasible power flow region is equivalent to*
 *solving a linear programming problem, e.g.,*

$$\min_{\mathbf{z}} 1 \quad (4a)$$

$$\text{s.t. } \mathbf{z}^T \mathbf{h}_i \geq 0, \text{ for all } i = 1, \dots, n, \quad (4b)$$

$$\sum_{i=1}^n \mathbf{z}^T \mathbf{h}_i = 1, \mathbf{h}_i = \left[\frac{\partial P_i}{\partial v_{i,r}}, \frac{\partial P_i}{\partial v_{i,m}}, \frac{\partial Q_i}{\partial v_{i,r}}, \dots, \frac{\partial Q_i}{\partial v_{i,m}} \right] \in \mathbf{R}^{2n}, \quad (4c)$$

where \mathbf{h}_i is the gradient of P_i with respect to all state variables. Therefore, \mathbf{h}_i is the transpose of the i th row of the Jacobian
 matrix. Let $\mathbf{z} \in \mathbf{R}^{2n}$ be a direction in which the real and imaginary parts of the voltages are moved.

A point is on the boundary if no direction exists in which the consumption at one bus can increase without decreasing
 consumption at others. Therefore, we can check if there is a direction \mathbf{z} that makes the optimization problem feasible. The
 objective function is irrelevant as we are only concerned with feasibility. Finally, an operating point is on the boundary if
 and only if the problem in (4) is infeasible. Similarly, if we find a unit vector \mathbf{z} such that the sum of the active power is
 maximized, the value of the optimization problem, denoted by Φ , represents the margin to the SOB. Therefore, solving the
 following optimization problem yields the margin to the SOB, with the optimal value of the objective function being the
 margin itself.

$$\Phi = \max_{\mathbf{z}} \sum_{i=1}^n \mathbf{z}^T \mathbf{h}_i \quad (5a)$$

$$\text{s.t. } \mathbf{z}^T \mathbf{h}_i \geq 0, \text{ for all } i = 1, \dots, n, \quad (5b)$$

$$\|\mathbf{z}\|_2 \leq 1. \quad (5c)$$

Figure R96: Calculating the SOB of DG system

Comment R3.8: *An exploration of how the limitations presented in your study could be overcome might add value to the overall research.*

Response to R3.8: Thank you for your valuable feedback. We appreciate your suggestion to explore how the limitations presented in the study could be addressed, as it would indeed add value to the overall research. In response, we have revised the limitations section of the manuscript to acknowledge potential ways to overcome these constraints and improve the comprehensiveness of our analysis. Below are the key aspects that we have incorporated:

- The first limitation comes from the simplified representation of the distribution grid, assuming balanced loading and aggregating demands. To overcome this, we propose integrating advanced grid management techniques such as Volt/Var optimization and dynamic Distributed Energy Resources (DER) inverter control in future research.
- The second limitation comes from the assumption of reliable and sufficient electricity supply in our current framework. This limited could be refined by incorporating supply-side uncertainties, such as variable renewable energy generation or disruptions in supply.
- The third limitation is the lack of political dimensions. Although political factors were not explicitly included in our current framework, future work could explore scenario-based approaches or qualitative methods to account for the impact of regulatory policies, geopolitical risks, and international agreements.

We have updated the limitations section in the manuscript accordingly, as shown in Figure R97.

**Limitation**

We show potential limitations of current study and potential solutions.

- • The first limitation comes from the simplified representation of the distribution grid, assuming balanced loading
and aggregating demands. To overcome this, we propose integrating advanced grid management techniques such as
Volt/Var optimization and dynamic Distributed Energy Resources (DER) inverter control in future research.
- • The second limitation comes from the assumption of reliable and sufficient electricity supply in our current frame-
work. This limited could be refined by incorporating supply-side uncertainties, such as variable renewable energy
generation or disruptions in supply.
- • The third limitation is the lack of political dimensions. Although political factors were not explicitly included in our
current framework, future work could explore scenario-based approaches or qualitative methods to account for the
impact of regulatory policies, geopolitical risks, and international agreements.

Figure R97: Limitation

References

- [R1] Federal Emergency Management Agency (FEMA). *Risk Assessment Overview*. 2024. URL: <https://www.fema.gov/oet-tools/chemical-incident-consequence-management/4/2>.
- [R2] National Renewable Energy Laboratory (NREL). *LA100: The Los Angeles 100% Renewable Energy Study and Equity Strategies*. 2021. URL: <https://maps.nrel.gov/la100/about#study-overview>.
- [R3] Maximilian Auffhammer, Patrick Baylis, and Catherine H Hausman. “Climate change is projected to have severe impacts on the frequency and intensity of peak electricity demand across the United States”. In: *Proceedings of the National Academy of Sciences* 114.8 (2017), pp. 1886–1891.
- [R4] Matthew D. Bartos and Mikhail V. Chester. “Impacts of climate change on electric power supply in the Western United States”. In: *Nature Climate Change* 5.8 (2015), pp. 748–752. DOI: 10.1038/nclimate2648.
- [R5] Jeremy B Bernerth and Herman Aguinis. “A critical review and best-practice recommendations for control variable usage”. In: *Personnel psychology* 69.1 (2016), pp. 229–283.
- [R6] Zhaohong Bie et al. “Battling the extreme: A study on the power system resilience”. In: *Proceedings of the IEEE* 105.7 (2017), pp. 1253–1266.
- [R7] Shuchen Cong et al. “Unveiling hidden energy poverty using the energy equity gap”. In: *Nature Communications* 13.1 (2022), p. 2456.
- [R8] Sarah L. Cox et al. *Bridging Climate Change Resilience and Mitigation in the Electricity Sector Through Renewable Energy and Energy Efficiency: Emerging Climate Change and Development Topics for Energy Sector Transformation*. Tech. rep. National Renewable Energy Lab. (NREL), Golden, CO (United States), 2017. DOI: 10.2172/1411521.
- [R9] Lucas W Davis and Paul J Gertler. “Contribution of air conditioning adoption to future energy use under global warming”. In: *Proceedings of the National Academy of Sciences* 112.19 (2015), pp. 5962–5967.

- [R10] Michael Dumas, Binit Kc, and Colin I. Cunliff. *Extreme Weather and Climate Vulnerabilities of the Electric Grid: A Summary of Environmental Sensitivity Quantification Methods*. Tech. rep. Oak Ridge National Lab. (ORNL), Oak Ridge, TN (United States), 2019. DOI: 10.2172/1558514.
- [R11] Ahmed Imran Hunjra et al. “Sustainable development: The impact of political risk, macroeconomic policy uncertainty and ethnic conflict”. In: *International Review of Financial Analysis* 84 (2022), p. 102370.
- [R12] Mark Z. Jacobson et al. “The United States can keep the grid stable at low cost with 100% clean, renewable energy in all sectors despite inaccurate claims”. In: *Proceedings of the National Academy of Sciences* 114.26 (2017), E5021–E5023. DOI: 10.1073/pnas.1708069114.
- [R13] Wendy S. Jaglom, James R. McFarland, Matthew F. Colley, et al. “Assessment of projected temperature impacts from climate change on the U.S. electric power sector using the Integrated Planning Model”. In: *Energy Policy* 73 (2014), pp. 524–539. DOI: 10.1016/j.enpol.2014.04.032.
- [R14] Laibao Liu et al. “Climate change impacts on planned supply–demand match in global wind and solar energy systems”. In: *Nature Energy* 8.8 (2023), pp. 870–880.
- [R15] National Research Council, Division on Earth, Life Studies, et al. “Applications of Toxicogenomic Technologies to Predictive Toxicology and Risk Assessment”. In: *National Academies Press* (2007). URL: <https://www.ncbi.nlm.nih.gov/books/NBK10201/>.
- [R16] Zhiyong Niu and Xiaoyan Zhou. “Political uncertainty and corporate sustainability: how does official turnover affect environmental investment”. In: *Asia-Pacific Journal of Accounting & Economics* 30.5 (2023), pp. 1184–1203.
- [R17] Nicholas P Simpson et al. “A framework for complex climate change risk assessment”. In: *One Earth* 4.4 (2021), pp. 489–501.
- [R18] Jennie C. Stephens et al. “Getting Smart? Climate Change and the Electric Grid”. In: *Challenges* 4.2 (2013), Article 2. DOI: 10.3390/challe4020201.
- [R19] Bas J Van Ruijven, Enrica De Cian, and Ian Sue Wing. “Amplification of future energy demand growth due to climate change”. In: *Nature communications* 10.1 (2019), p. 2762.

- [R20] Russell S Vose et al. “Temperature changes in the United States”. In: *Climate science special report: Fourth national climate assessment* 1.GSFC-E-DAA-TN49028 (2017).
- [R21] David M. Ward. “The effect of weather on grid systems and the reliability of electricity supply”. In: *Climatic Change* 121.1 (2013), pp. 103–113. DOI: 10.1007/s10584-013-0916-z.
- [R22] Leonie Wenz, Anders Levermann, and Maximilian Auffhammer. “North–south polarization of European electricity consumption under future warming”. In: *Proceedings of the National Academy of Sciences* 114.38 (2017), E7910–E7918.

Response to the Reviewers

NCOMMS-23-63838A

Vulnerability of Power Distribution Networks to Local Temperature Changes Induced by Global Climate Change

Kishan Prudhvi Guddanti¹, Lin Chen¹, Yang Weng*, and Yang Yu*

We thank the editor and reviewers for their thoughtful comments which helped us to improve the paper. Based on the comments, we have carefully revised the paper. For example, the reviewers' comments are in *italics teal* color text and our answers are in regular font text. Changes to the original paper are in regular *blue* color text. Please see our point-to-point response to each of the reviewers' comments below. The references are attached at the end of the response document.

Response to Reviewer 1

Comment R1.1: *Thank you for taking my advice seriously. The article is complete and clearly explained, contains a lot of detail and methodological notes, and I think the logic and ideas are very clear and intuitive throughout the article. However, there are still some minor issues with the article that can be revised.*

Response to R1.1:

Thank you for your positive feedback and for recognizing the clarity and comprehensiveness of our article. We appreciate your comments and have carefully reviewed and addressed the minor issues you raised. Specifically, we have made revisions to improve the formatting and expression throughout the manuscript, ensuring greater clarity and consistency.

Comment R1.2: *The formatting of the literature cited in the article is not consistent, with some citations using the [16] format and others being superscripted in the top right-hand corner of the text. Please confirm the formatting problem.*

Response to R1.2: Thank you for your valuable feedback. The inconsistency in citation formatting arose because references using the [16] format were placed as part of the sentence subject, while other references in superscript were used to indicate the source of specific content in the sentence. We agree that this inconsistency is not ideal.

To address this, we have revised the sentence structures to maintain the original meaning while converting all citations to the superscript format. As a result, the citation style has now been unified throughout the manuscript.

Attached below are screenshots showing a few examples of the revisions made.

the GCC on various components of DG, including shifts in demand profiles and changes to the grid infrastructure. Data-
driven methods have been developed to estimate how electricity demand responds to local temperature fluctuations in the
transmission grid, providing insights into load variations¹²⁻¹⁴. Discussions on the GCC's impact on DG infrastructure
can be categorized into two main approaches. The first focuses on physical damage to infrastructure caused by extreme
weather events. For instance, studies have shown that DG components such as the grid lines and transformers are highly
susceptible to damage from extreme weather conditions^{15,16}. Other forms of structural damage have also been explored in

Figure R1: The revised citation formatting

Comment R1.3: *On page 8, in the sentence “As demand rises, a growing GDP may encourage grid upgrades.”, “a growing GDP” can be changed to “GDP growth” for consistency.*

Response to R1.3: Thank you for your suggestion. We have made the recommended changes to improve the consistency of the manuscript. The phrase “a growing GDP” has been updated to “GDP growth” as suggested.

Please find the screenshot below, which shows the revised sentence.

altering demand-temperature sensitivity and grid capacity. For instance, GDP growth can affect electricity demand²⁶. As
demand rises, **GDP growth** may encourage grid upgrades. To model GDP growth scenarios, we use the Shared Socio-

Figure R2: The phrase “a growing GDP” has been updated to “GDP growth”

Comment R1.4: *Consistency in Tense: Ensure that the tense is consistent throughout the passage. For example, if you are discussing future projections, use future tense consistently.*

Response to R1.4:

Thank you for pointing out the issue with tense consistency in our manuscript. We have carefully reviewed and revised the text to ensure that the tense is consistent throughout. Below are some examples of the changes we made:

Page 8:

Original Sentence: *The risk between 13:00 and 22:00 increases from around 12% in 2000 to 18% in 2100*

Change: *The risk between 13:00 and 22:00 increased from around 12% in 2000 to 18% in 2100*

Page 11:

Original Sentence: *Under the RCP 8.5 scenario, the projected changes in the distribution network affected by climate change vary geographically.*

Change: *Under the RCP 8.5 scenario, the anticipated changes in the distribution network affected by climate change are expected to vary geographically.*

Page 12:

Original Sentence: *In the U.S., GCC triggered substantial divergence across states.*

Change: *In the U.S., GCC is expected to cause substantial divergence among states.*

Page 16:

Original Sentence: *All models and data will be described in detail later in this section.*

Change: *All models and data are described in detail later in this section.*

Comment R1.5: *On page 12, “The changes in colors show that 55% of the states need to upgrade DG to ensure reliable distribution services for users, whose demand-weather response relation remains the same.” “relation” can be changed to “relationship.”*

Response to R1.5: Thank you for your constructive feedback. We have made the suggested change, replacing “relation” with “relationship” to improve clarity and consistency.

The revised sentence now reads as follows:

434 states need to upgrade DG to ensure reliable distribution services for users, whose demand-weather response relationship
remains the same.

Figure R3: Replacing “relation” with “relationship”

Comment R1.6: *Add “the” for nouns present in some sentences.*

Response to R1.6: Thank you for pointing out the issue with tense consistency in our manuscript. We have carefully reviewed and revised the text to ensure that the tense is consistent throughout. Below are some examples of the changes we made:

Page 2:

Original Sentence: *Furthermore, the nonlinear response of power flow can amplify these cumulative effects throughout the cascading process,*

Change: *Furthermore, the nonlinear response of the power flow can amplify these cumulative effects throughout the cascading process,*

Original Sentence: *Understanding GCC’ s impact on distribution grid by disturbing power flow is critical.*

Change: *Understanding the GCC’ s impact on the distribution grid by disturbing the power flow is critical.*

Original Sentence: *Every blackout in DG causes a severe economic loss.*

Change: *Every blackout in the DG causes a severe economic loss.*

Page 3:

Original Sentence: *We first model the mediated process through which the GCC’s impact on grid line and transformer capacities alters DG’s safe operational boundary.*

Change: *We first model the mediated process through which the GCC’s impact on the grid line and transformer capacities alters DG’s safe operational boundary.*

Comment R1.7: *In conclusion, this paper has analyzed the impact of global climate change (GCC) on electricity distribution in the United States and Europe and the mitigating or exacerbating effects of various technologies, such as photovoltaic (PV) systems and electric vehicles (EVs), on the impact of GCC. The paper details the methodology used to assess system risk, including climate modeling, load consumption data, historical temperature data, and distribution network data.*

Response to R1.7:

Thank you for your summary of the paper's conclusions. We appreciate your taking the time to understand our work's key findings and methodology. We would be happy to make the necessary revisions if you have further suggestions for improving clarity or adding detail.

Response to Reviewer 2

Comment R2.1: *The revised manuscript addresses several of the previously raised concerns. In its current form, it proposes a framework to analyze the vulnerability of distribution grids to climate change (temperature impacts) and applies it to various regions to derive vulnerability insights. However, the revised manuscript still demonstrates three key areas of improvement.*

Response to R2.1: Thank you for recognizing our previous revisions. Based on the suggestions, we have made the following three key improvements in this round of revisions:

1. **Terminology Modification:** We have replaced the term “systematic framework” with “framework of model integration” to better reflect the modeling-oriented nature of our study. This modification emphasizes that our research is based on model integration for analysis rather than being limited to a conceptual framework.
2. **Reorganizing the Paper Based on New Keywords:** Following the updated terminology, we have restructured the paper to highlight the integration of the temperature-demand model, temperature-facility model, and distributed grid power flow model. This reorganization clarifies the core ideas of the framework and demonstrates its role in capturing the complex interactions among climate change, socioeconomic factors, and grid reliability.
 - **Redefining the Model Integration Framework:** The integration of temperature-demand, temperature-facility, and DG power flow models forms a framework that captures the complex interactions between climate change, load, infrastructure, and power flow changes. This framework enables the assessment of blackout risks under various scenarios and supports the development of adaptive strategies that account for both individual and interfered effects of these factors.
 - **Micro Perspective:** At the micro level, the framework allows for analyzing how GCC impacts demand, power flow, and blackout risks. Using a California distribution grid as a case study, we evaluate how climate change alters risks under different scenarios and explore targeted strategies for optimizing grid topology and load distribution to improve resilience.
 - **Macro Perspective:** At the macro level, the framework enables analysis of how climate change affects DG systems across regions. It facilitates comparisons

of regional pressures in maintaining reliable electricity supply under GCC impacts, helping identify areas requiring stronger adaptation strategies to ensure stability in distribution services.

3. **Rewriting and Optimizing the Discussion Section:** We have moved the analysis of adaptation options from the original discussion section to earlier sections, aligning it with the micro-level analysis for a more coherent logical flow. The revised discussion section focuses on comparing our key findings with existing literature, clarifying the study's contributions and innovations. Additionally, we have enhanced the limitations section to provide insights for future research, including:

- **Model Incompleteness:** While we did not assume a static demand-temperature relationship, we explored potential dynamic changes based on existing literature. However, future studies should conduct a more comprehensive analysis of how socioeconomic dynamics influence grid vulnerabilities.
- **Limited Socioeconomic Scenarios:** Our study adopted commonly used scenarios from the literature, but there may be other significant, unexplored scenarios that lie beyond the scope of this research.
- **Uncertainty in Climate Projections:** Given the inherent assumptions and data quality issues in climate projection models, our results should be interpreted cautiously, considering the range of possible future variations.

We hope these updates address your concerns and provide a clearer and more comprehensive presentation of the study.

Comment R2.2: *First, the framework introduced is more modeling-oriented than conceptually oriented, in that it accounts for limited drivers and interactions particularly considering the socioeconomic impacts on energy supply (including distribution network) and energy demand, as well as on various broad-based climate adaptation strategies with energy implications. I will strongly recommend that the manuscript clarifies the “systematic framework” as a “modeling and analysis” framework and reframes the study accordingly.*

Response to R2.2: We appreciate your constructive feedback. Your comments made us discover that our paper provides a framework for model integration. Compared to other papers, the unique contribution is combined impacts through the chain effect. Therefore, we have changed the term “systematic framework” to “TB3R framework of model integration” to reflect our study’s modeling-oriented nature better. This revision emphasizes that our approach is based on model integration for analysis rather than a conceptual framework.

Accordingly, we have revised the abstract to highlight the modeling-oriented nature of our framework and its role in assessing the GCC’s interfered effect on DG blackout risk. The revised abstract is shown in the following screenshot:

**Abstract**
Global climate change (GCC) changes temperature, inducing a complex chain effect that is reshaping blackout risks in
power distribution grids. The GCC’s impacts on electricity supply, demand, and facilities combine to vary distribution
grid (DG)’s safe-operation boundary as well as power flow, which causes a change of DG’s blackout risk. To assess the
impact of the GCC’s interfered effect on the DG’s blackout risk, this work developed a framework of mode integration
for assessing and analyzing the chain effect. The framework of mode integration enables us to observe that the GCC’s
interfered effect has been threatening the DG’s reliability, potentially increasing blackout risks during peak-load hours
by 4% to 6%, depending on GDP growth rates. The GCC’s interfered effect is amplified by the Kirchhoff-laws ruled
complex intermediate process of the chain effect, and thus pushes the DG’s blackout risk growth through a non-linear
trajectory. Furthermore, the power-flow analysis enabled by our framework provides new insights for DG’s adaptation
strategies to GCC, including cost-effective grid upgrades, strategic placement of distributed energy resources, and match-
ing temperature-sensitive users with robust buses to enhance resilience. We developed an index to compare the intensi-
ty of the GCC’s combined impacts through the chain effect. Our analysis based on the index reveals that over 20% of
U.S. are expected to require a minimum increase of over 10% in the standard DG capacity before 2050, with six states
needing more than a 20% increase. This suggests that DG capacity in vulnerable regions may need to double by the end
of the century. In contrast, the impact of GCC on power distribution in Europe is moderate. Our results recommend
that policymakers focus on managing peak loads and addressing nonlinear risk trends across different geographic areas.

Figure R4: Revised Abstract.

In response, we have revised the introduction to clearly outline how it introduces the complex climate-economic-engineering chain effect process through which the combination of GCC’s impacts influences DG blackout risk and why this necessitates a model integration framework. The specific modifications are shown in Figure R5. GCC triggers changes in multiple components of the energy system, including nodal electricity demand, grid infrastructure capacity, and extreme weather events, all of which influence DG reliability. These impacts do not act independently but combine to propagate through power flow dynamics

governed by Kirchhoff’s law, forming a chain effect. This process introduces nonlinear relationships between GCC-induced changes and DG blackout risks, amplifying the combined impacts of climate change on the grid. While existing studies address individual aspects—such as demand profile changes or infrastructure vulnerabilities—they fail to capture GCC’s integrated and mediated effects on DG reliability.

Introduction

Global climate change (GCC) has been deeply impacting every component of the electricity sector, such as resizing the
service capacity of the grid facilities and reshaping the profile of electricity demand^{1,2}. For instance, GCC is projected to
increase the number of cooling-degree days³ and exacerbate the peak electricity demand for adapting to the extremely-high
temperature in hot hours³⁻⁵. Meanwhile, the anticipated rise in local temperatures in cooling-degree days due to GCC will
reduce the capacity of distribution grid (DG) lines and increase the electricity demand.

The combination of the impacts of GCC on various components can influence the reliability of the power system,
especially the blackout risk of DG. DG is the low-voltage grid that draws power from the high-voltage transmission grid
and delivers electricity to industrial, business, and residential users. In contrast to the high-voltage transmission grid, DG
has less redundancy and protection^{6,7}. Such a fact makes DG the vulnerable part of the energy sector. It has been pointed
out that 90% of the blackouts in the U.S. are caused by the breakdown of DGs⁸. Therefore, the DG’s blackout risk can be
more sensitive to the changes in grid infrastructure and electricity demand and supply during the GCC process.

The combined impacts of GCC affect the DG reliability through a cascading effect. Initially, GCC influences the DG’s
demand, grid topology, operational capacities, and overall reliability. These factors collectively reshape the DG’s power
flow and its maximum service capacity. Ultimately, these changes alter the probability of blackouts. Additionally, the
dynamic of the intermediate process of the chain effect makes the combination of the GCC’s impacts not simple. In the
intermediate process, Kirchhoff’s law rules how the combination of the GCC’s impacts changes the DG’s power flow, which
induces a complex and non-linear relation between the degree and temperature change and the range of the DG’s blackout
risk variation. Therefore, assessing and analyzing the GCC’s impacts on the DG’s blackout risk must capture the chain
process and adequately model the intermediate process.

It is crucial to understand how the combined impacts of GCC influence the DG reliability through a chain effect. On one
hand, these interconnected impacts can make DG more vulnerable. Even if each individual GCC’s impact is insufficient to
cause a blackout event, their interfered effects can push DG to the brink of failure. Furthermore, the nonlinear response of
the power flow can amplify these cumulative effects throughout the cascading process, exacerbating the DG vulnerability
and increasing the risk of system failure. Additionally, the reliability of the electricity distribution service plays a crucial
role in shaping regional economic advantages and quality of life. For instance, the electricity reliability in extremely hot and
cold hours influences the regional public health. Every DG blackout causes a severe economic loss⁹. Finally, the progress
of the energy transition drives a growing penetration of electric vehicles, distributed energy generation technologies, and
energy storage facilities, all connected to DG¹⁰. Therefore, a reliable electricity distribution service is essential for the
stability and advancement of human society¹¹.

To understand the combined impacts of GCC through the chain effect, it is essential to integrate models that capture both
the GCC’s influence on various components of the electricity sector and the way these combined impacts translate into the
DG’s blackout risk through the complex DG power flow dynamics. Existing studies have primarily examined the impacts
of GCC on various components of DG, including shifts in demand profiles and changes to the grid infrastructure. Data-
driven methods have been developed to estimate how electricity demand responds to local temperature fluctuations in the
transmission grid, providing insights into load variations¹²⁻¹⁴. Discussions on the GCC’s impact on the DG’s infrastructure
can be categorized into two main approaches. The first focuses on physical damage to infrastructure caused by extreme
weather events. For instance, studies have shown that the DG’s components such as the grid lines and transformers are highly
susceptible to damage from extreme weather conditions^{15,16}. Other forms of structural damage have also been explored in
this context^{17,18}. The second approach examines capacity constraints, which arise when electricity demand surpasses the
substation capacity, leading to operational inefficiencies and reliability concerns^{5,11,12}.

However, there is a lack of a framework that integrates the GCC’s impact on nodal load and infrastructure facilities,
while modeling the chain effect through power flow dynamics to assess and analyze the DG’s blackout risk. This gap arises
from the inherent challenge of converting changes in nodal load and grid infrastructure into the complex dynamics of the
DG’s power flow. As a result, the extent to which the DG reliability is reshaped by the interfered effects of load fluctuations
and infrastructure reliability during the GCC process remains unclear. The relationship between GCC and the blackout
risk in distributed generation systems is governed by a complex, chain climate-economic-engineering process¹⁹, making it
difficult to accurately estimate real-world the DG vulnerability based solely on temperature changes.

Figure R5: Revised introduction outlining the climate-economic-engineering chain effect and the necessity of a model integration framework.

To address this gap, the introduction motivates the necessity of a model integration framework. Such a framework is essential to connect models capturing GCC’s impacts on various components (e.g., climate, demand, infrastructure) and to analyze how these combined impacts propagate through power flow dynamics to influence DG blackout risk. Figure R6 presents these refinements. Our proposed “TB3R framework of model integration” systematically integrates these models, enabling a comprehensive assessment of DG blackout risk under GCC. This framework allows us to understand the intricate interactions between climate, economic, and engineering factors and provides insights into DG vulnerability and adaptation strategies. These revisions reframe our study as a modeling and analysis framework, aligning with your suggestion to clarify its purpose and scope.

To capture the complex climate-economic-engineering chain effect process through which the combined impacts of
GCC influence the DG’s blackout risk, we develop a framework of model integration for assessing and analyzing the DG’s
blackout risk during the GCC process. We applied our DG’s power-flow analysis method²⁰ to model the intermediate
process that converts changes in nodal loads and grid infrastructure into the probability of the DG’s blackout risk. Our
framework integrates the GCC model, the temperature-demand model, and the temperature-facility change model, using
the intermediate-process model as the nexus that connects these components. This integration enables us to analyze the
GCC’s impact on real-world DG, considering the DG’s micro-topological structures and other technical characteristics.
Additionally, we demonstrate that our framework can be applied to compare the GCC’s impact across different geographic
regions, providing insights into regional variations in the DG vulnerability. When applied to realistic data, our framework
can quantitatively assess the impact of global climate change on various parts of a grid (micro-perspective^a) and across
different geographic areas (macro-perspective^b). Its scalable design ensures computational efficiency. By capturing the
relationship between GCC and the DG’s power flow, our model integration framework reveals the detailed micro-dynamics
of the chain effect, providing key insights into the DG vulnerability and adaptation to GCC. For instance, our analysis
uncovers how the DG’s topology and technical characteristics influence its climate vulnerability at a granular level. In
response, we identify novel adaptation strategies for DG, including topological redesign and the strategic placement of
distributed renewable energy sources across the DG’s nodes, which enhance resilience against the GCC impacts.
Our framework of model integration also reveals that the combination of the GCC’s impact amplified by the intermediate
power-flow dynamics has been influencing the DG reliability in the real world. The influence was offset by the efforts of the
DG investment for adapting to economic growth before 2020. However, the GCC’s effect will be sufficiently large and can
no longer be offset by the DG expansion driven by economic growth after 2030. To our best knowledge, those observations
have not been discovered in the existing literature.

Figure R6: Further refinements in the revised introduction emphasize the necessity of model integration.

Below, we outline how the paper reorganizes its structure around the model integration framework in three key aspects:

1. Redefining TB3R Framework of Model Integration

The revised section introduces the theoretical basis of TB3R, which centers on the safe-operation boundary (SOB) concept in distribution grids (DG). Specifically, we explain how the GCC-induced impacts on grid capacities and nodal loads propagate through the power flow equations to alter the SOB. Figure R7 provides the revised theoretical foundation. The SOB represents the set of nodal load profiles under which power flows remain within the physical limits of grid lines and transformers. Analyzing the SOB

allows us to define the blackout risk of the DG, which is the probability that load demands exceed the operational limits of the grid under various scenarios.

**Theoretical foundation**

The chain effect is a complex climate-economic-engineering process: the GCC-induced temperature change reshapes the
 service capacity of the DG facilities as well as the demands withdrawing electricity from the DG nodes; the changes of
 facilities' capacities resize the DG's maximum capacity for meeting nodal demands while the changes of facilities and
 demand together reshuffle the DG power flow; Ultimately, the variations of DG's maximum capacity and power flow
 reshape the occurrence probability of the DG's blackout. Therefore, it is necessary to combine the models for GCC's
 impacts on DG facilities and demands with the models capturing the intermediate process for assessing and analyzing the
 GCC's interfered effect.

We first explain the intermediate process through which the GCC's impact on the grid line and transformer capacities
 alters the DG's maximum capacity for meeting nodal demands, constrained by the physical features of the grid lines and
 facilities according to Kirchhoff's laws. An outage occurs when the power flow through any line exceeds its capacity. We
 define the safe-operation region as the set of all nodal load profiles that prevent such overflows. The boundary of this
 region is known as the safe-operation boundary (SOB). The SOB is determined by the physical characteristics of the DG,
 including the capacities of grid lines and transformers²¹⁻²³. The SOB is mathematically derived by analyzing the power
 flow equations for each node. We formalize this derivation in the following theorem, with a formal derivation provided in
 the following theorem and detailed further in the Method section.

**Theorem 1.** *We can compute the margin to the SOB by solving the optimization problem 5 in the Method section. This
 problem involves finding a direction \mathbf{z} such that the power flow is maximized while satisfying the physical constraints of
 the system. The power gradient \mathbf{h}_i for the voltage helps to trace the SOB. The margin to the SOB, denoted by Φ , can then
 be calculated. When the solution to the optimization problem yields $\Phi[\mathbf{P}] = 0$, the system is at the SOB, indicating that the
 grid is operating at the maximum load boundary without collapsing.*

The above theorem formalizes the intermediate process of determining the SOB, which represents the maximum load
 the distribution grid can handle without collapsing. The SOB is a physical characteristic of the grid, determined by its
 infrastructure, such as line capacities and transformer limits, and independent of specific load profiles. By analyzing the
 power flow equations at each node, the boundary is derived as the point beyond which the grid can no longer sustain
 additional load.

We then examine the intermediate process by which changes in DG facilities and nodal demands reshape power flow
 dynamics. At any given time, DG nodal demands and the physical characteristics of DG infrastructure jointly determine the
 magnitude of power flow across grid lines, subject to topological constraints imposed by Kirchhoff's laws. Since both nodal
 demand and facility capacities are temperature-sensitive, GCC alters the probability distributions of demand fluctuations
 and facility property at each hour. As a result, the distribution of the power-flow size in each line changes.

The above two intermediate processes ultimately cause the change of DG's blackout risk. The risk of a DG blackout
 in any given hour h is determined by the probability that the demand profile exceeds the SOB. This blackout risk can be
 quantified as follows:

$$\text{Risk}_h = \sum_{c=1}^t p(T_c) \cdot p(\Phi[\mathbf{P}|T_c, h] \leq 0)$$

Here, T_c represents climate change scenario c , $p(T_c)$ denotes the probability of climate change scenario c , and $\Phi[\mathbf{P}|T_c, h]$
 determines the margin of the demand profile \mathbf{P} under temperature T_c at hour h to the SOB. If $\Phi \leq 0$, the grid remains within
 safe operating conditions. Otherwise, a blackout will occur.

Figure 1(a) and (b) explain the complex climate-economic-engineering dynamics in the chain effects of a three-node
 example. The possible power pairs in an hour are the dots in blue and green. The SOB is the boundary that separates the safe
 points in blue and the risky points in green. When GCC causes a local temperature rise, nodal loads increase accordingly.
 Consequently, the oval region Figure 1(a) with a green boundary will move to the blue oval in Figure 1(b). Meanwhile, the
 maximal power-flow capacity of every line is also influenced by temperature. According to the Temperature Coefficient
 Formula²³, the conductivity of a metal decreases with temperature, which implies that the power flow in a transmission line
 decreases as the temperature increases. Consequently, temperature changes the blue boundary in Figure 1(a). In Figure 1(b),
 we can see a shrunk version in red towards the origin when compared to the blue SOB region in Figure 1(a). Therefore, the
 GCC-induced temperature change alters both the size and shape of the SOB region and the oval region. These two regions
 jointly decide the risk probability of a blackout.

We note that the chain effect through which the GCC's combined impacts influence DG blackout risk can be modu-
 lated by engineering advancements and socio-economic developments. As illustrated in Figure 1(c), the energy transition
 process, including the deployment of distributed PV and energy storage, can alter how grid line capacity changes reshape
 the SOB. Additionally, the temperature-demand relationship is influenced by demographic, socio-cultural, and economic
 factors, making it highly context-dependent. Consequently, a DG's TB3R can vary across different energy transition and
 socio-economic scenarios, highlighting the need for region-specific adaptation strategies. For instance, although the local
 temperatures in Colorado and Kansas are similar in the US, the demand-temperature relations are different due to different
 demographic, socio-cultural, and economic characteristics. Consequently, the TB3R in Colorado and Kansas must be dif-
 ferent even if they have similar climate change patterns. Therefore, analyzing the interactive effect of the GCC alongside
 other factors, such as energy transition and economic development, requires modeling the intermediate process of the chain
 effect to capture these complex interdependencies accurately.

Figure R7: Revised theoretical foundation explaining the propagation of GCC-induced impacts through the power flow model.

The chain effect of GCC' s combined impacts on DG blackout risk can be influenced by other engineering and socio-economic progress, such as energy transitions and demographic changes. Meanwhile, demographic, socio-cultural, and economic factors influence the relationship between temperature and demand, making the TB3R relationship highly scenario-dependent. Therefore, analyzing the interactive effects of GCC, energy transitions, and economic development requires modeling the TB3R process under diverse scenarios.

In the Analysis Diagram Based on Model Integration within the TB3R framework, we systematically address this complexity by integrating multiple models. Figure R8 illustrates the revised framework. Specifically, we:

- Model the Hazard: Model the magnitude of the GCC-driven local temperature changes according to the GCC model.
- Scope the Analysis: Determine whether to isolate the pure effect of GCC or include the interactive effects of other factors, such as GDP growth or energy transitions.
- Calculate the Climate-Blackout Response: Developing the DG's TB3R by integrating the temperature-demand model, temperature-SOB model, and power flow model. We then apply TB3R across various GCC scenarios to quantify the climate-blackout response, which reveals DG's vulnerability to GCC.
- Assess and Compare GCC' s Impact: Evaluate the increase in blackout risk driven by GCC and compare regional variations in stress on reliable electricity distribution.
- Design Adaptation Strategies: Propose engineering and policy solutions, such as grid upgrades or topology redesigns, to mitigate the GCC-induced vulnerabilities identified in the analysis.

The TB3R framework allows us to assess changes in blackout risk under various climate change scenarios, while incorporating socio-economic factors such as energy demand, economic growth, and policy changes. The modifications in Figure R9 clarify these assessments. For example, based on temperature changes induced by climate change, we can analyze how energy demand shifts in different regions, which then informs the changes in power flow and their impact on DG system reliability. This framework can be used to simulate different climate scenarios (e.g., higher summer temperatures or extreme weather events) combined with socio-economic changes (e.g., GDP growth or

156 **TB3R Framework of Model Integration**

The previous section explains the chain effect how the GCC's combined impacts influences the blackout risk. Therefore, assessing the GCC's impact on DG reliability requires an integrated modeling approach that encompasses the GCC-driven local meteorological changes, electricity demand response, grid infrastructure adaptation, variations in the SOB, and DG power flow dynamics. To address this, we outline the framework of model integration, which is referred as the Temperature-Blackout Risk Response (TB3R) framework and combines the GCC model, temperature-demand sensitivity model, temperature-grid SOB model, and power flow model. The TB3R framework, inspired by reference²⁴ and illustrated in Figure 1(d), involves five steps and is built on three elements of risk: hazard, vulnerability, and response²⁴. The hazard pillar focuses on understanding the type and magnitude of the threat induced by GCC reshaping DG's SOB and power flow. By incorporating the scenario of the GCC-induced hazard, we apply the TB3R framework to examine and clarify the climate-economic-engineering process and assess the degree of the DG's vulnerability to the GCC's threat. The vulnerability analysis further provides valuable insights into potential engineering solutions for adaptation. Our framework is capable of addressing various impacts of GCC on DG reliability. Below, we outline the detailed steps for assessing the impact caused by the GCC-driven temperature changes:

- • Modeling the Hazard: Model the magnitude of the GCC-driven local temperature changes according to the GCC model.
- • Scoping: Identify whether and what other processes are included when assessing the GCC's impact. When the intervention of all other processes, such as GDP growth is excluded, the assessment calibrates the size of the GCC's pure effect. Otherwise, the assessment verifies the GCC's effect intervened by the change of other factors.
- • Calculating the climate-blackout response: Developing the DG's TB3R by integrating the temperature-demand model, temperature-SOB model, and power flow model. We then apply TB3R across various GCC scenarios to quantify the climate-blackout response, which reveals DG's vulnerability to GCC.
- • Assessing and comparing the GCC's impact: Calibrate the increase of the blackout risk driven by GCC, and compare the GCC-induced stress on reliable-electricity distribution over regions.
- • Designing the adaptation strategy: Analyze how the GCC-driven temperature change raises the DG's blackout risk and design adaptation strategies.

from each other results in an increased risk of DG collapse. (c) This diagram illustrates the chain effect of GCC's combined impacts and associated processes. It depicts how GCC-induced temperature changes influence DG blackout risk, with intermediate processes shaped by energy transitions, as well as demographic, socio-cultural, and economic factors. (d) The analysis diagram based on model integration for assessing the impact of climate change on electricity distribution services involves polarizing the demand. This analysis diagram is visualized using a centered flower-style plot that summarizes the hazard-vulnerability-response model²⁴. Hazard refers to the

Figure R8: Revised framework outlining the step-by-step integration of models within TB3R.

energy policy transformation) to assess how these factors together influence blackout risks. In the following sections, using the TB3R framework of model integration, we systematically analyze GCC's impacts on DG reliability through both micro and macro perspectives, enabling a comprehensive understanding of the climate-economic-engineering chain effect.

To gain a deeper understanding of the chain effect and the influences of technology advancements and social-economic
development, we employ a two-step analysis. This approach allows us to systematically apply the TB3R framework to
examine both the “pure effect” and the “interfered effect” of the GCC’s combined impacts. This approach is critical for
the following reasons: When assessing the pure effect, we exclude the moderating influences of other natural and human
activities on DG blackout risk. This allows us to determine whether and when the GCC-driven temperature changes have
begun to manifest in the studied area, as well as to understand the characteristics of the GCC’s impact. The pure effect,
as detailed in the subsection “GCC’s Pure Effect”, provides a benchmark for investigating how GCC interacts with other
factors, such as GDP growth, to influence DG reliability. This is explored further in the subsection “Interfered Effect of
GCC’s combined impacts”. In contrast, examining the “interfered effect” reveals whether and when GCC begins to pose
a threat to the reliability of the DG in various future technological and social-economic scenarios. Within each scope,
the average effect of GCC, its effect during vulnerable hours, and the degree of exposure are assessed using indices such
as annual and seasonal average risks, risks during peak-load hours, and the number of vulnerable hours. The detailed
methodology for this analysis is explained in the Methods section.

Beyond assessing the vulnerability of a specific DG to GCC, the TB3R framework also reveals how the chain effect
can influence the geographically heterogeneity of the significance that GCC’s combined impacts worldwide. As the GCC-
induced temperature rises drive up peak electricity demand, maintaining reliable electricity distribution becomes both more
challenging and costly. This increased stress on the system results in higher costs for ensuring reliability. Consequently,
the minimal DG capacity required to guarantee reliable electricity distribution serves as an indicator of the stress caused
by GCC. The nonlinear dynamic of the chain effect determines how significant the GCC’s combined impacts challenge the
reliable electricity delivery service. To compare the geographic differences of the stress, we applied the TB3R framework
to develop an index examining the minimal size of a DG to reliably distribute electricity. Given that DG blackout risk
is sensitive to system topology, consistent use of the same DG model is essential for comparing the GCC-induced stress
across regions. To standardize this comparison, we introduce the *Minimum Reliable Standard DG*, measured in units of *per*
*standard model (p.s.m.)*. This index quantifies the minimum size of the standardized IEEE DG model needed to reliably
distribute electricity in various GCC climate scenarios. We emphasize that this index captures the nonlinear impact of GCC,
shaped by Kirchhoff’s laws, as it accounts for the process of converting the GCC-induced changes in load and infrastructure
into variations in DG’s SOB and power flow dynamics. Therefore, by analyzing the index over time and across regions, we
can gain insights into how fundamental power system physics govern the temporal evolution and geographic variability of
the GCC’s impact on distributed electricity services. Details on the methodology for this index are provided in the Methods
Section.

In the following sections, we will design experiments to explore how various characteristics of demand and DG affect
the impact of climate change. Our analysis will be conducted at both macro and micro scales:

- • Micro perspective: We will analyze the vulnerability of the real-world power infrastructure within specific regions,
and discuss how the intermediate processes of the chain effect influence the vulnerability and adaptation strategies.
- • Macro perspective: We will compare the geographic heterogeneity of the GCC’s effect in aggravating the stress of
delivering reliable electricity distribution service.

We will also explore the DG’s adaptation to the GCC-driven local temperature change.

Figure R9: Revised TB3R framework integrating climate, economic, and engineering perspectives.

2. Micro Perspective: The Significant Threat of Climate Change to Distribution Grids

In this section, we use a distribution grid in California as a case study to examine the specific impacts of climate change on distribution systems. At the micro level, the integrated framework enables a detailed analysis of how GCC influences demand changes, which in turn alter power flows, ultimately leading to changes in blackout risks. This includes evaluating changes in blackout risks under given climate and socioeconomic scenarios, analyzing the independent effects of climate change and its interactions with other factors, and exploring targeted strategies for planning distribution grid topology and load distribution to better adapt to climate change. The specific analysis includes the following aspects:

- Analyzing the Pure Effects of Climate Change:

The integrated framework allows us to assess changes in blackout risk under various climate change scenarios, while incorporating socio-economic factors such as energy demand, economic growth, and policy changes. In this section, to Assessing pure effect of GCC in Blackout Risk, we selecte specific climate and socio-economic scenarios. Based on temperature changes induced by climate change, we can analyze how energy demand shifts in different regions, which then informs the changes in power flow and their impact on DG system reliability. Our findings reveal that GCC' s pure effects on reshaping power flow have significantly increased the DG' s exposure to high blackout risk over the past decades and are projected to intensify throughout this century. Even if all other factors remain unchanged, the rising temperatures caused by GCC would substantially elevate the DG' s average blackout risk and worsen vulnerabilities during critical hours, highlighting the urgent need for adaptive planning and targeted mitigation strategies.

**Pure Effect of GCC's combined impacts**

**The TB3R framework of model integration enables us to examine the GCC's pure effect on blackout risk through the chain**
**effect.** We discovered that the pure effect of GCC's combined impacts have already existed over the past few decades and
is projected to intensify throughout the remainder of this century. If everything except the local temperature keeps the same
as the current situation, the pure effect have increased the DG's blackout risk during the last half century and will worsen
the DG's blackout risk. We summarize the characteristics of the GCC's pure effect below. The pure effect only would
largely increase the studied DG's exposure to high blackout risk and the risk in the vulnerable hours. Consequently, the
DG's average risk would substantially rise even if everything except the local temperature keeps the same.

Figure R10: Reorganized paper from the perspective of model integration: revision on the pure effect.

The power-flow analysis highlights the underlying reasons for the non-linear impact of GCC's pure effect on blackout risk. According to Kirchhoff's laws and the DG grid topology, the same increase in demand at different nodes results in heterogeneous changes in power flow and associated blackout risks across the network. For instance, an increase in electricity demand at nodes farther from the substation leads to a disproportionately larger rise in blackout risk compared to nodes closer to it. Therefore, even if nodal demands exhibit similar patterns of change due to temperature increases, the blackout risk will grow non-linearly, with multiple tipping points emerging. Additionally, consumers at different nodes often have varying threshold temperatures for energy usage, causing uneven increases in nodal loads under the same temperature change. This further amplifies the non-linear response of blackout risk to GCC-induced temperature changes. The revision is shown in Figure R11.

Our TB3R framework uncovers a series of critical characteristics of the GCC's combined impacts on DG's blackout risk,
 which is shaped by the non linearity of chain effect and can only be revealed through power-flow analysis. For example, our
 results highlight the nonlinear relationship between GCC's combined impacts and DG blackout risk, a key factor in guiding
 future DG investment and management strategies. By comparing the plots in Figure 2(a), (b), and (c), we observed the
 changes in slope of the blackout risk's growth trajectory in the rest of this century. For example, in Figure 2(c), under the
 RCP 8.5 scenario, the impact of GCC on summer-average blackout risk reaches a tipping point around 2050. After this
 point, the average blackout risk grows at a faster rate compared to earlier years. This non-linearity is evident in different
 aspects of DG blackout risk, as seen in the varying curves for different scenarios.

In contrast, the risk trajectory for peak-load hours shown in Figure 2(b) reveals multiple tipping points. In the RCP 8.5
 scenario, the GCC-driven risk increase accelerates in the 2030s, flattens in the 2040s, and then accelerates again in the 2050s.
 This non-linear behavior arises from complex interactions between demand and DG engineering factors. On the demand
 side, electricity consumption responds non-linearly to temperature changes. As local temperatures exceed certain thresholds
 due to GCC, electricity demand surges sharply because of increased use of air conditioning and other weather-responsive
 equipment, creating tipping points in risk growth.

Because of the chain effect governed by Kirchhoff's laws and the DG topology, identical demand increases at different
 nodes lead to heterogeneous changes in power flow and, consequently, different degrees of blackout risk changes. For
 example, an increase in electricity demand at nodes farther from the substation results in a greater rise in DG blackout
 risk compared to nodes closer to it. Thus, even if nodal demand exhibits a uniform response to temperature increases,
 the blackout risk will grow nonlinearly, with multiple tipping points emerging along the way. Additionally, consumers at
 different nodes often have varying threshold temperatures that trigger changes in energy consumption. As a result, the same
 temperature increase leads an uneven rise in nodal loads, introducing a secondary effect that further amplifies the nonlinear
 relationship between the temperature changes and the blackout risk.

This non-linear pattern suggests that policymakers can use our TB3R framework to plan DG expansions more effectively,
 as the tool not only characterizes the non-linearity of risk growth but also provides reliable guarantees. The non-linear effect
 induced by power-flow dynamics further amplifies the polarization of the GCC's effect. Our TB3R framework manifested
 that the GCC's threat on DG's blackout risk is highly concentrated in vulnerable hours and simultaneously expands the
 number of vulnerable hours. For example, Figure 2(d) compares the average hourly blackout risk in each month for the
 279 years 2000, 2050, and 2100 under the RCP 8.5 scenario. The risk between 13:00 and 22:00 increased from around 12% in
 2000 to 18% in 2100, while risks at other hours remain relatively constant, even in the RCP 8.5 scenario. Lastly, our TB3R
 framework also enables the analysis of outliers in risk distributions. As shown in Figure 2(e), the capability for alignment
 with the dashed line indicates that risk distributions are skewing towards longer tails over time. The RCP 8.5 scenario, in
 particular, demonstrates a more pronounced long-tail distribution compared to RCP 4.5 by 2100. This suggests that extreme
 weather will likely result in increasingly extreme DG blackout risks in the future.

We argue that our findings on peak-load hours are applicable beyond California. In California DG, GCC has a stronger
 impact in the peak-load hours rather than off-peak load hours because that the demand is more sensitive to the temperature in
 hotter scenarios. Literature has demonstrated that the relationship between DG demand and temperature in most areas in the
 world is represented by a U-shaped curve, where demand spikes during both hotter summer and colder winter hours, which
 are also peak-load periods^{13,26}. Therefore, we conclude that GCC will similarly increase DG blackout risks during peak-
 load hours in most other regions in the world. Additionally, the power flow's effect causing the highly non-linear TB3R
 also world-wide exists. Therefore, the GCC's polarization effect on demand will be further amplified when converting to
 the impact on the DG's blackout risk.

Figure R11: Reorganized paper from the perspective of model integration: revision on the non-linear impact.

- Analyzing the Interfered Effect Intervened by Changes of Other Factors:
By reorganizing the paper through the lens of model integration, we systematically analyze the complex interplay between GCC and GDP growth on DG blackout risk. Using the T3RB framework, the paper highlights how GCC’s pure effects on power flow, such as temperature-driven load changes, interact with GDP-driven grid upgrades. While GDP dominates risk mitigation before 2030, the non-linear escalation of GCC’s impact becomes evident beyond this point as temperatures surpass threshold levels, triggering accelerated blackout risks under RCP 8.5 scenarios. The model integration approach also reveals how faster GDP growth amplifies GCC’s impact, as seen in SSP scenarios where higher GDP increases peak-hour and vulnerable-hour blackout risks. This integrated perspective provides deeper insights into the temporal and nonlinear dynamics shaping DG system reliability. The revision is shown in Figure R12.

**Interfered Effect of GCC’s combined impacts**

We further investigated how other factors, such as GDP growth and energy transitions, could interfere the chain effect
through which the GCC’s combined impacts change the DG blackout risk. These factors can interfere the chain effect by
altering demand-temperature sensitivity and grid capacity. For instance, GDP growth can affect electricity demand²⁶. As
demand rises, GDP growth may encourage grid upgrades. To model GDP growth scenarios, we use the Shared Socio-
economic Pathways (SSP) framework²⁷. For each GDP growth scenario, we consider three scenarios involving varying
demand, SOB, and GCC: (1) a benchmark scenario without GCC, (2) the RCP 4.5 scenario with moderate GCC, and (3)
the RCP 8.5 scenario with significant climate change. Using these scenarios, we apply the U-shaped curve to model load
changes and infrastructure upgrades, resulting in specific SOB outcomes in the TB3R framework.

The power-flow analysis enabled by the T3RB model framework reveals a profound interaction between the impacts of
GCC and GDP growth on DG blackout risk through the chain effect. When the progress of GCC and GDP intertwine, their
relative influence on power flow through the chain effect varies over time, altering whether GCC or GDP is the dominant
driver of DG blackout risk changes at different periods. AS shown in Figure 3(a) and (b), the GDP growth is the primary
driving force before 2030. Consequently, while the pure effect analysis in the previous section confirms the presence of
the GCC’s impact on power flow, it does not significantly exacerbate the average DG blackout risk beyond the GDP-driven
effects. Correspondingly, the grid upgrades spurred by GDP growth effectively mitigate the blackout risk increase caused
by temperature changes during this period. However, the GCC’s effect on power flow causes a significant extra risk increase
in addition to what GDP does. These GDP-driven grid upgrades are insufficient to counteract the risk increases caused by
temperature changes after 2030, as the GCC’s influence on DG blackout risk remains relatively moderate before this point.

A further examination on the mixed effects of GCC and GDP growth on the DG power flow clarifies the conditions when
the GCC’s effect contributes significant extra blackout risk in addition to the GDP growth’s effect. According to the pure
effect analysis, there exists a threshold temperature beyond which the GCC’s effect on DG’s blackout risk will be accelerated
when the non-linear effect of the power flow dynamic converts the temperature change to the blackout risk increase. In the
RCP 8.5 scenario, the local temperature will reach the threshold level around 2030. Consequently, the tipping point—where
the GCC’s impact on blackout risk starts to escalate—occurs around 2020–2030, as indicated by the pure effect shown in
Figure 2(c). After 2030, the rate of increase in blackout risk during peak-load hours and the number of vulnerable hours
accelerates more rapidly in the GCC scenarios compared to the benchmark scenario. Correspondingly, DG expansion to
accommodate GDP-driven demand growth will no longer counterbalance the effects of GCC. This divergence results in
distinctly different risk trajectories for DG blackout across the three GCC scenarios for the remainder of the century.

We discover that the GDP growth can non-linearly influence the magnitude of the GCC’s impact on the DG’s blackout
risk due to the complex interactions between their effects on power flow during the intermediate processes of the chain
effect. The GCC’s impact on the DG’s blackout risk varies significantly across different GDP scenarios examined in this
research. In the SSP 1 scenario, the summer-average blackout risk of Figure 3(a) by 2100 is projected to more than double
compared to 2000, increasing from 3.36% to 7.16% in the RCP 8.5 scenario. In the SSP 2 scenario, which assumes slower

Figure R12: Reorganized paper from the perspective of model integration: revision on the interfered effect.

- Insights into GCC Adaptation and DG Resilience:

In response to the reviewer’s suggestion, we have moved the results related to assessing adaptation options from the discussion section into this micro perspective analysis section to enhance the logical clarity of our work, which is shown in Figure R13. Using the TB3R framework, we examine how GCC reshapes SOB and power flows, revealing critical but often overlooked risk factors, such as the influence of topology type and locational vulnerabilities. For instance, comparative analyses of tree and mesh network structures demonstrate that tree systems are highly sensitive to temperature-induced load changes, particularly at remote nodes, leading to significantly higher blackout risks. In contrast, mesh systems exhibit enhanced resilience, maintaining consistently lower risks across all scenarios. These findings highlight the necessity of adopting mesh topologies or strategically deploying distributed energy technologies at critical nodes to enhance system resilience under climate change. By integrating these insights into a unified framework, the paper emphasizes the importance of rethinking DG planning and adaptation strategies to address the complex challenges posed by GCC effectively.

**Efficient Adaptation by Managing the Chain Effect**

With its ability to analyze the micro-dynamics of how the SOB and power flow are reshaped during the GCC, the TB3R
framework enables the identification of critical yet previously overlooked risk factors—such as the locations of temperature-
sensitive users and the configuration of network topology—that significantly influence DG blackout risk. Figure 4 illustrates
our experimental setup and results for evaluating the impact of network topology. For instance, we tested two types of
network structures: a tree structure (Figure 4(d)) and a mesh structure (Figure 4(h)). For each structure, we analyzed
vulnerable nodes under three scenarios: a benchmark scenario, a climate change scenario, and a climate change scenario
with consumer reallocation. The third scenario is specifically designed to assess locational vulnerability. Figure 4(a) shows
the joint distribution of loads P_2 and P_3 , while Figure 4(b) depicts the temperature-induced load increase for P_2 .

Figure R13: Reorganized paper from the perspective of model integration: revision on the GCC Adaptation.

3. Macro Perspective: Geographically Heterogeneous Impacts of Climate Change on Regional Distribution Services

At the macro level, our framework provides a powerful analytical tool to study how climate change impacts DG systems across different geographic regions. Using the integrated framework, we can compare the pressures faced by different regions in adapting to climate change.

The integrated framework enables the analysis of how climate change affects the stability of distribution services by assessing the difficulty of providing stable electricity supply under GCC impacts, and facilitates a standardized cross-country comparison. This allows us to compare the challenges faced by various regions or countries in maintaining reliable distribution services under changing climate conditions, based on power flow analysis. By integrating this information, we can better understand where the supply of stable distribution services is more difficult and where more significant adaptation strategies may be required.

We revised the terminology to better reflect the modeling and analysis orientation of the study and clearly explained how this framework is applied to analyze GCC's impacts on DG systems. We reorganized the study to emphasize the integration of key models (temperature-demand, temperature-facility, and DG power-flow models) and how this integrated framework captures the complex interactions between climate change, socio-economic factors, and DG system reliability. We also highlighted how the framework can be used to evaluate broad-based climate adaptation strategies with energy implications.

We believe these changes, based on your comment, provide a more precise and more comprehensive explanation of the framework and its application. If you have further suggestions or questions, we will revise the manuscript accordingly.

Comment R2.3: *Second, the discussion section needs to discuss the key findings, in the context of the existing literature and the key contributions the study makes recognizing the inherent limitations and key areas of improvement. Results on assessing adaptation options can be moved to preceding sections.*

Response to R2.3: Thanks for the suggestion to enhance the clarity and depth of our discussion. In response, we have revised the discussion section to explicitly frame our key findings in the context of existing literature, outline our study’s contributions, and acknowledge the inherent limitations and areas for future research. Per your recommendation, we have also moved the results on assessing adaptation options to the preceding sections. Below are the key adjustments we have made:

1. Key Findings in Context of Existing Literature:

In this study, we examined the impact of GCC on the blackout risk of DGs through a complex long-chain effect process, which differs significantly from direct infrastructure damage commonly studied in prior research. While existing literature focuses on catastrophic weather events such as storms or floods that destroy critical components like transformers, our analysis highlights the indirect mechanisms through which GCC influences DG systems. Specifically, GCC-induced local temperature changes affect power demand and the SOB in DG systems, which in turn alter power flow dynamics and ultimately influence blackout risk. Intricate physical laws govern these processes and manifest as a long-chain reaction.

Compared to static or single-node analyses in existing studies, our work contributes by addressing the dynamic interplay between temperature-driven demand changes, SOB variations, and power flow adjustments. This dynamic perspective allows us to identify distinct risk phases for DG systems under GCC—prolonged plateaus followed by abrupt risk escalations—which were not captured in prior studies. We have revised the text to elaborate on these findings and their distinction from existing literature, as shown in the figure below:

In this work, we examined how GCC’s impacts on DG’s different components such as demands and facilities combine
together to influence the risk of DG blackout through a complex chain effect. Unlike direct infrastructure damage, such
as extreme weather events that destroy critical grid components such as transformers, GCC’s combined impacts changes
DG’s blackout risk indirectly through intricate intermediate processes. For example, local temperature changes driven by
GCC alter power demand and DG line capacities, reshaping the SOB and power flow of DG, ultimately affecting the risk of
blackout. Accurately assessing the impact of GCC on the risk of DG blackout within this chain effect requires an integrated
modeling framework that captures the dynamics of each intermediate process.

Figure R14: The revised discussion: key findings in context of existing literature

2. Study Contributions and Novelty:

The study's primary contributions lie in its novel focus on intermediate processes, particularly the roles of SOB and power flow dynamics, within the long-chain effects of GCC. These contributions include:

- **Understanding Abrupt Changes in Blackout Risk:** By utilizing the TB3R framework, which integrates models for SOB and power flow dynamics, we demonstrated that power flow variations driven by GCC can trigger sudden shifts in DG blackout risk. This finding provides a novel explanation of how GCC-induced tipping points emerge and contributes to understanding the mechanisms behind nonlinear climate impacts.

The intermediate process of the chain effect are governed by complex physical laws. Therefore, these intermediate
processes are critical in determining when and how chain effects induced by GCC may lead to sudden changes in blackout
risk. Using TB3R framework, which integrates models for SOB changes and DG power flow dynamics, we confirmed that
power flow variations can amplify the GCC's chain effects, triggering abrupt transitions in DG blackout risk. Our analysis,
incorporating pure effect studies and SSP scenario evaluations, reveals that DG blackout risk evolves in distinct phases,
characterized by prolonged stability followed by sudden, nonlinear jumps.

Figure R15: The revised discussion: abrupt changes in blackout risk.

- **Policy-Relevant Tipping Points:** Our analysis identified critical tipping points at which DG reliability rapidly deteriorates. These tipping points can guide policymakers in optimizing the timing of investments for DG adaptation to climate change. Accurately assessing consumer demand thresholds near these tipping points enables more targeted, cost-effective interventions to mitigate risks and enhance long-term system resilience.

Understanding these abrupt shifts is crucial for informing policy and investment decisions related to adapting the DG to
climate change. As climate impacts intensify non-linearly, the reliability of the DG can rapidly deteriorate once it reaches
a critical tipping point. An economically efficient adaptation strategy requires strategic resource allocation around these
tipping points to mitigate risks and maintain system stability. Achieving this requires a precise assessment of consumer
demand thresholds to anticipate when a DG system will likely reach its tipping point, beyond which reliability declines
sharply. Identifying these thresholds enables timely, cost-effective interventions that enhance resilience and long-term
sustainability.

Figure R16: The revised discussion: identifying policy-relevant tipping points for adaptation.

- **Innovative Adaptation Strategies:** Through SOB and power flow analyses, we uncovered unique factors contributing to DG vulnerability and proposed several novel strategies:
 - Aligning temperature-sensitive electricity users with resilient grid nodes.

- Deploying distributed PV systems and energy storage at vulnerable nodes to enhance adaptation efficiency.
- Redesigning DG grid topologies tailored to local climate and socio-economic conditions to optimize resource allocation and reduce mismatch risks.

Analyzing the GCC’s chain effects provides new insights into DG vulnerability and novel strategies for adaptation.
 Power flow analysis reveals that a mismatch between temperature-sensitive electricity users and vulnerable grid nodes
 significantly increases a DG system’s susceptibility to the GCC-induced temperature changes. Consequently, effective
 adaptation strategies should focus on identifying vulnerable nodes and ensuring an optimal alignment between temperature-
 sensitive users and resilient grid infrastructure. Deploying distributed PV systems and energy storage at these critical
 nodes can significantly enhance the cost-efficiency of DG adaptation. Furthermore, power flow analysis indicates that DG
 vulnerability to GCC depends on grid topology. Thoughtfully designing and refining DG network topology, tailored to local
 climate conditions and socio-economic factors, presents a novel yet under-explored approach to enhancing DG resilience
 against the GCC impacts.

Figure R17: The revised discussion: novel strategies for GCC-driven DG vulnerability mitigation.

- **Spatial and Temporal Risk Polarization:** Our spatial and temporal analysis revealed a polarization of blackout risks across different regions, states, and countries in the US and EU. This underscores the importance of region-specific policies that account for varying local economic growth and energy transition scenarios. Our findings also highlight a previously unexplored interaction mechanism between GCC, economic growth, and energy transitions, suggesting that future evaluations of blackout risk should integrate both growth trends and gradual temperature changes rather than focusing solely on extreme weather events.

The TB3R framework reveals that the intermediate process of the chain effect is critical to determine whether a region’s
 electricity distribution reliability can be heavily impacted by GCC. Our macro analysis demonstrated that two regions that
 have similar the GCC-driven temperature changes can have divergent electricity-distribution sensitivity to GCC, due to
 their different social-economical conditions. Our spatial and temporal analysis of DG risks shows how the intermediate
 process of the chain effect can polarize the GCC sensitivity of electricity distribution service across different regions, states,
 and countries in the US and EU. Our micro discussion about DG’s topology and energy transition also demonstrated how
 engineering and technical features influence a region’s DG vulnerability to GCC. All these discoveries demonstrated the
 critical role of the intermediate process, which differs across regions. Therefore, all these findings in this work highlight the
 need for further academic discussion and policy practices for region-specific strategies tailored to local DG vulnerabilities.

Figure R18: The revised discussion: spatial and temporal polarization of blackout risks.

3. Recognizing Limitations and Areas for Improvement:

We have enhanced the limitations section to provide insights for future research, including:

- **Model Incompleteness:** While we did not assume a static demand-temperature relationship, we explored potential dynamic changes based on existing literature.

However, future studies should conduct a more comprehensive analysis of how socioeconomic dynamics influence grid vulnerabilities.

- **Limited Socioeconomic Scenarios:** Our study adopted commonly used scenarios from the literature, but there may be other significant, unexplored scenarios that lie beyond the scope of this research.
- **Uncertainty in Climate Projections:** Given the inherent assumptions and data quality issues in climate projection models, our results should be interpreted cautiously, considering the range of possible future variations.

4. Results on Adaptation Options Moved to the Preceding Sections:

As suggested, we have moved the discussion of adaptation options to the earlier sections on micro analysis. The adaptation discussion focuses on the “topology” and “micro-structural features” of the distribution network and its vulnerabilities.

**Efficient Adaptation by Managing the Chain Effect**

With its ability to analyze the micro-dynamics of how the SOB and power flow are reshaped during the GCC, the TB3R
framework enables the identification of critical yet previously overlooked risk factors—such as the locations of temperature-
sensitive users and the configuration of network topology—that significantly influence DG blackout risk. Figure 4 illustrates
our experimental setup and results for evaluating the impact of network topology. For instance, we tested two types of
network structures: a tree structure (Figure 4(d)) and a mesh structure (Figure 4(h)). For each structure, we analyzed
vulnerable nodes under three scenarios: a benchmark scenario, a climate change scenario, and a climate change scenario
with consumer reallocation. The third scenario is specifically designed to assess locational vulnerability. Figure 4(a) shows
the joint distribution of loads P_2 and P_3 , while Figure 4(b) depicts the temperature-induced load increase for P_2 .

Figure R19: Move the GCC adaptation to the earlier sections on micro analysis.

Comment R2.4: *Third, the methodological framework has inherent limitations, besides the three identified under the “Methods” section, most of which were pointed out in the previous review report. The manuscript needs to be explicit about these limitations and cautious when interpreting the findings. These and other key areas of improvement need to be highlighted in the discussion section.*

Response to R2.4:

Thank you for your feedback. In the revised manuscript, we acknowledged the **inherent limitations** of the methodological framework in the discussion section. These limitations include:

1. **Model Incompleteness:** The framework may not fully capture the dynamic relationship between electricity demand and temperature under evolving socioeconomic conditions. Factors such as changes in population behavior, energy efficiency improvements, and technological advancements can significantly influence this relationship. While we explored this under GDP growth scenarios based on existing literature, we emphasized the need for future research to develop more comprehensive models that better account for these dynamics.

1. The framework may not fully capture the dynamic relationship between electricity demand and temperature under
evolving socioeconomic conditions. Factors such as shifts in population behavior, improvements in energy efficiency,
and technological advancements can significantly influence this relationship. While we examined GDP growth sce-
narios based on existing literature, future research should develop more comprehensive models better to reflect the
evolving interplay between electricity demand and temperature.

Figure R20: The added limitation: Model Incompleteness.

2. **Limited Socioeconomic Scenarios:** Our study focuses on commonly used socioeconomic scenarios from the literature, providing a solid foundation for understanding GCC’ s impacts. However, these scenarios may not cover all potential variations in socioeconomic conditions. For example, we highlighted the importance of investigating alternative economic growth patterns, urbanization trends, and policy innovations in future studies to better assess distribution grid vulnerabilities. Expanding the scope of scenarios would enable a more holistic understanding of risks and adaptation strategies.

2. Our study primarily relies on widely used socioeconomic scenarios from the literature, providing a strong foundation
for understanding the GCC's impacts. However, these scenarios may not capture the full range of socioeconomic
variations. Future research should explore alternative economic growth trajectories, urbanization trends, and policy
innovations to assess how different pathways might alter distribution grid vulnerabilities. Expanding the range of
scenarios will enable a more comprehensive assessment of potential risks and adaptation strategies across diverse
regions.

Figure R21: The added limitation: Limited Socioeconomic Scenarios

3. **Uncertainty in Climate Projections:** We acknowledged that the findings rely on climate projection models, which inherently carry uncertainties arising from variations in emission scenarios, model assumptions, and data quality. These uncertainties can influence the accuracy of long-term predictions, particularly under extreme climate conditions. To address this, we emphasized the importance of cautious interpretation of the results and suggested employing probabilistic approaches or ensemble modeling in future work to improve robustness.

3. The study's findings are based on climate projection models, which inherently contain uncertainties due to variations
in emission scenarios, model assumptions, and data quality. These uncertainties can affect the accuracy of long-
term predictions, particularly under extreme climate conditions. While we recognize these limitations, our results
should be interpreted cautiously, considering the potential range of future outcomes. Future research could enhance
robustness by employing probabilistic approaches or ensemble modeling techniques to quantify better and mitigate
uncertainties in the GCC impact assessments.

Figure R22: The added limitation: Uncertainty in Climate Projections

Given these limitations, we provide caution in drawing conclusions. The above discussion serves to refine the scope of our findings, emphasizing that certain conclusions cannot be extended in directions constrained by these limitations. To clearly distinguish the discussion section's inherent limitations from methodological constraints, we have also revised the limitations presented in the methods section to explicitly highlight modeling restrictions. The revised "Limitation and potential future work on method" is shown in Figure R23. Specifically:

- **Simplification of the demand-side model** –Future improvements could incorporate refined household-level data to enhance the accuracy of electricity demand estimations.
- **Simplification of the energy transition model** –Our current approach does not account for user- or investor-driven transitions influenced by policy and environmental attitudes, which could be further explored in future studies.
- **Absence of technological progress considerations** –The potential impact of emerging technologies, such as advanced energy storage and grid management solutions, could be integrated to refine the framework's applicability.

**Limitation and potential future work on method**

We show the potential limitations of current modeling.

- • The first limitation comes from the simplified representation of the electricity demand model for the DG. Due to the
lack of the demographic details of household-level data, most electricity demand models in the current literature are
regional rather than household. Consequently, there are few models about how the household electricity demand
varies by household-level demographic features. Therefore, the electricity demand model integrated into the TB3R
framework in this paper is regional and makes it difficult to clarify how the results are influenced by different demo-
graphic and social scenarios that have various distributions of race, income, and marital status. Further, the lack of
household level model prevents us from applying the TB3R to analyze DGs directly connect to households. Future
studies can integrate an improved household-level demand-side model into the TB3R framework for further research.
- • The second limitation comes from the simplified discussion for energy transition. Energy transition in DG can be
self-motivated by the households or independent investors, who are influenced by the energy transition policies and
individual environmental attitudes. Because this paper focuses on the chain effect of the GCC's combined impacts,
it does not include related discussions to avoid distraction. Consequently, the conclusions in this work cannot reflect
how users and investors respond rationally to GCC. In future studies, the TB3R framework can be combined with
energy transition models to further discuss the influence of policy design and public attitudes on DG's vulnerability
and adaptation to GCC.
- • The third limitation is the simplification between socio-economic change and GCC's combined impacts. In this work,
we addressed whether the blackout risk increase driven by GCC's combined impacts can be mitigated by the expansion
of DG responding to the demand increase induced by GDP growth. However, the decisions on DG expansion and
renewal can also be influenced by other factors such as price designs for energy storage. The absence of those details
in the model disables the TB3R framework from comprehensively understanding socio-economic changes and the
DG blackout risk change due to GCC's combined impacts. For instance, before improving the DG expansion model,
the TB3R framework is hard to discuss the effect of the market designs for energy transition.

The above method limitations restrict this paper's estimation on the size of GCC's combined impacts on DG's blackout
risk in the discussed scenarios. However, these limitations do not affect the main conclusion of this paper, such as the
tipping points of the nonlinear trajectory of DG blackout risk and the novel insights for DG's vulnerability and adaptations
to the GCC's combined impacts through the chain effect. In future studies, the TB3R framework remains valuable due to
its careful modeling of the intermediate process of the chain effect. The improved socio-economic models, policy models,
and energy transition models can be integrated into TB3R to examine the corresponding change of SOB and power flow.

Figure R23: The revised “Limitation and potential future work on method”